# Continual Momentum Filtering on Parameter Space for Online Test-time Adaptation

**Jae-Hong Lee** [*]**, Joon-Hyuk Chang** [*]
Department of Electronic Engineering
Hanyang University
Seoul, Republic of Korea
{ljh93ljh,jchang}@hanyang.ac.kr

## Abstract

Deep neural networks (DNNs) have revolutionized tasks such as image classification and speech recognition but often falter when training and test data diverge in distribution. External factors, from weather effects on images to varied speech environments, can cause this discrepancy, compromising DNN performance. Online test-time adaptation (OTTA) methods present a promising solution, recalibrating models in real-time during the test stage without requiring historical data. However, the OTTA paradigm is imperfect, often falling prey to issues such as catastrophic forgetting due to its reliance on noisy, self-trained predictions. Although some contemporary strategies mitigate this by tying adaptations to the static source model, this restricts model flexibility. This paper introduces a continual momentum filtering (CMF) framework, leveraging the Kalman filter (KF) to strike a balance between model adaptability and information retention. The CMF intertwines optimization via stochastic gradient descent with a KF-based inference process. This methodology not only aids in averting catastrophic forgetting but also provides high adaptability to shifting data distributions. We validate our framework on various OTTA scenarios and real-world situations regarding covariate and label shifts, and the CMF consistently shows superior performance compared to state-of-the-art methods.

## 1 Introduction

Deep neural networks (DNNs) have been successfully applied to challenging tasks such as image classification (Krizhevsky et al., 2012; Simonyan & Zisserman, 2014; He et al., 2016b) and speech recognition (Hinton et al., 2012; Graves et al., 2013; Jelinek, 1997). The success of DNNs stems from the assumption that training and test data are drawn from the same distribution (Goodfellow et al., 2016; Murphy, 2023). However, this assumption is difficult to maintain in real-world environments owing to external factors (Hendrycks & Dietterich, 2019b; Koh et al., 2021). For example, images may be damaged due to weather changes or sensor degradation during the image classification tasks. Similarly, in speech recognition tasks, discrepancies arise from differences in the speaking environment or the frequency of words used by speakers compared with the source data. These distribution shifts lead to a significant performance degradation of the DNN models (Quinonero-Candela et al., 2008; Sun et al., 2017). To address these distributional discrepancies, there is growing interest in online test-time adaptation (OTTA) methodologies (Wang et al., 2020; Zhang et al., 2022). These innovative frameworks are equipped to recalibrate models on-the-fly during the test stage. Impressively, they achieved this without delving back into historical data, leaning exclusively from the knowledge gleaned from the pre-trained source model. This real-time adaptation bypasses many challenges associated with data storage, retrieval, and potential privacy infringement.

Nevertheless, the journey of the OTTA is with roadblocks. Rooted in a self-training paradigm, it habitually employs its own noisy predictions as training targets during adaptation. This recursive feedback can induce a series of complications, most notably catastrophic forgetting, in which

---

[*]Corresponding author.

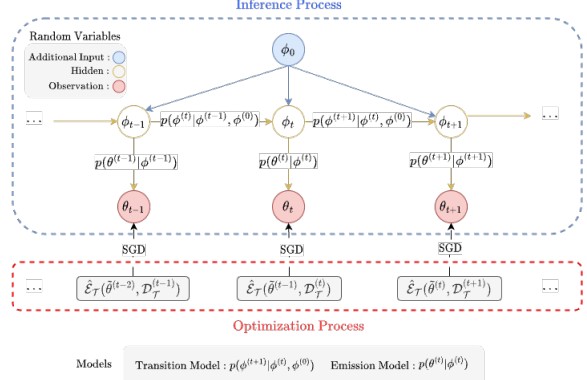

Figure 1: Illustration of the CMF framework. The optimization process draws the target model parameter $\theta^{(t)}$ by minimizing the target generalization error $\hat{\mathcal{E}}_{\mathcal{T}}$ using SGD on the target data $\mathcal{D}_{\mathcal{T}}^{(t)}$. The inference process derives $\tilde{\theta}^{(t+1)}$ with the refined source model's parameter (hidden) $\phi^{(t)}$ and the target model's parameters (observation) $\theta^{(t)}$ thorough the parameter ensemble method.

models inadvertently overwrite or discard previously assimilated information. Furthermore, there is a looming threat of mode collapse, where models disproportionately amplify the probabilities of specific classes at the expense of others (Boudiaf et al., 2022). Contemporary solutions, ranging from calibrating particular neural layers (Yang et al., 2022; Zhao et al., 2023; Hong et al., 2022) to tethering adaptations to static source model information, have been proposed (Wang et al., 2022; Niu et al., 2022; 2023; Marsden et al., 2023). SAR Niu et al. (2023) stochastically restores the target model parameters to those of the source model during adaptation. Similarly, ROID (Marsden et al., 2023) continuously ensembles the parameters of the source and target models to prevent loss of past information. These strategies regularize the target model to prevent it from diverging too far from the source model, thereby preventing catastrophic forgetting.

However, the continuous use of frozen source model information restricts the flexibility of the target model, making it difficult to adapt to distribution shifts. For more flexibility, we need to update the source model using noisy information from the target model; however, this carries the risk of catastrophic forgetting. The Kalman filter (KF) (Särkkä & Svensson, 2023) can emerge as a beacon in the quest for a harmonious balance between adaptability and information retention. The KF is an exact Bayesian filtering algorithm that accumulates past observations (i.e., parameters of the target model) along with denoising observations. The linear Gaussian state-space model (LG-SSM), which forms the basis of the KF, consists of three types of random variables: additional input, hidden, and observation. These random variables comprise two types of linear Gaussian models: *the transition model*, which models the evolution of the hidden variable over the adaptation time steps, and *the emission model*, which predicts the observations from the hidden variable.

In this paper, we propose a continual momentum filtering (CMF) framework that utilizes the KF algorithm to find a source model that is robust against catastrophic forgetting while maintaining high flexibility. The CMF alternates between an optimization process based on stochastic gradient descent (SGD) (Ruder, 2016) and an inference process based on the KF. During the optimization process, the target model is obtained by minimizing the target generalization error using an unsupervised loss function. For the inference process, the CMF conjugates the source model with the transition model to continuously preserve the information in the hidden variable (i.e., parameters of a refined source model) and an emission model to update the hidden variables by denoising the parameters of the target model. Finally, the refined source and target models are ensembled to perform the following optimization process. Through CMF, we utilize the information of the noisy target model to obtain a refined source model and use it for adaptation, thus narrowing the gap between adaptability and robustness and providing a new perspective on OTTA.

We validate the proposed framework in various scenarios used in existing OTTA methods (Niu et al., 2022; 2023; Marsden et al., 2023). These scenarios are broadly divided into covariate and label shifts and further classified based on the degree of time correlation of each input or label datum. For the covariate shifts, we experiment with scenarios in which the input data drawn from different domains are mixed or appear together. For the label shifts, we gradually adjust the degree to which the same labels simultaneously appear over time. We also validate our framework on a speech recognition task

for real-world streaming scenarios. The CMF demonstrates significant performance improvements in these wide ranges of scenarios compared to state-of-the-art methods.

## 2 BACKGROUND

This section elaborates on the OTTA problem and regularization methods for preventing catastrophic forgetting. First, we outline the setup of the problem (Section 2.1) and subsequently introduce a strategy of existing studies to mitigate catastrophic forgetting using the source model (Section 2.2).

### 2.1 TARGET GENERALIZATION ERROR FOR ONLINE TEST-TIME ADAPTATION

Let the labeled data from source distribution $p_{\mathcal{S}}$ be $\mathcal{D}_{\mathcal{S}} = \{(\mathbf{x}_n, \mathbf{y}_n) \sim p_{\mathcal{S}} : n = 1 : N_{\mathcal{S}}\}$ and the unlabeled data from target distribution $p_{\mathcal{T}}^{(t)}$ at each time step $t \in \{1, 2, \dots, T\}$ be $\mathcal{D}_{\mathcal{T}}^{(t)} = \{\mathbf{x}_n \sim p_{\mathcal{T}}^{(t)} : n = 1 : N_{\mathcal{T}}\}$ where $p_{\mathcal{S}} \neq p_{\mathcal{T}}^{(t)}$. The main objective is to minimize the generalization error on the target distribution $p_{\mathcal{T}}^{(t)}$, which can be calculated as

$$\mathcal{E}_{\mathcal{T}}(\Theta^{(0)}, p_{\mathcal{T}}^{(t)}) = \mathbb{E}_{p_{\mathcal{T}}^{(t)}}[\ell(f(\mathbf{x}_n; \Theta^{(0)}))], \tag{1}$$

where $f(.; \Theta^{(0)})$ represents the DNN pre-trained on $\mathcal{D}_{\mathcal{S}}$ and $\ell(.)$ is the unsupervised loss function. The target generalization error can be approximated empirically as follows:

$$\hat{\mathcal{E}}_{\mathcal{T}}(\Theta^{(0)}, \mathcal{D}_{\mathcal{T}}^{(t)}) = \frac{1}{N_{\mathcal{T}}} \sum_{\mathbf{x}_n \in \mathcal{D}_{\mathcal{T}}^{(t)}} \ell(f(\mathbf{x}_n; \Theta^{(0)})). \tag{2}$$

OTTA literature frequently utilizes entropy-based loss functions because the target model is adapted to unlabeled target data (Wang et al., 2020; Zhang et al., 2022; Chen et al., 2022). In particular, TENT (Wang et al., 2020) demonstrated the effectiveness of entropy-based loss in a self-training manner on a single domain; however, recent research has revealed that methods solely based on self-training often fail in more complex scenarios, such as multiple domains (Boudiaf et al., 2022; Gong et al., 2022; Niu et al., 2023).

### 2.2 PREVENTING CATASTROPHIC FORGETTING USING SOURCE MODEL

Minimizing the empirical generalization error causes the adapted target model to experience catastrophic forgetting. This results in a mode collapse problem, in which the probabilities assigned to certain classes become excessively large, causing significant degradation in performance (Niu et al., 2023; Marsden et al., 2023). A popular approach for mitigating this issue involves minimizing the empirical generalization error alongside the regularization loss that inhibits the target model from diverging too far from the source model Boudiaf et al. (2022). The target model parameter $\Theta^{(t+1)}$ at adaptation time step $t + 1$ can be found as follows:

$$\Theta^{(t+1)} = \underset{\Theta^{(t)}}{\arg\min} \, \hat{\mathcal{E}}_{\mathcal{T}}(\Theta^{(t)}, \mathcal{D}_{\mathcal{T}}^{(t+1)}) + \lambda \mathrm{d}(\Theta^{(0)}, \Theta^{(t)}), \tag{3}$$

where $\mathrm{d}(\Theta^{(0)}, \Theta^{(t)})$ represents the distance between the source and target models, and $\lambda$ is the regularization coefficient. Strategies that adapt only a subset of the source model parameters, $\theta^{(0)} \subset \Theta^{(0)}$, essentially perform a function to the regularization loss because the majority of the adapted parameters are identical to those of the source model. In particular, methods that calibrate only batch normalization (BN) layers (Ioffe & Szegedy, 2015; Li et al., 2018; Mancini et al., 2018; Yang et al., 2022; Zhao et al., 2023; Hong et al., 2022) have been widely recognized for their effectiveness in preventing catastrophic forgetting. However, these methods exhibit structural limitations when applied to models that employ layer normalization (LN) layers (Ba et al., 2016) because they alter the functionality of BN (Lin et al., 2022).

Recently, strategies that continuously transfer the information acquired from the source model to the target model (Wang et al., 2022; Niu et al., 2023; 2022; Marsden et al., 2023) have overcome these limitations, thereby alleviating the catastrophic forgetting problem across a broader range of model structures. For instance, EATA (Niu et al., 2022) mitigates catastrophic forgetting by utilizing the

---

**Algorithm 1** Continual Momentum Filtering

---
INPUT:
    Input data stream $\{\mathcal{D}_{\mathcal{T}}^{(1)} \dots \mathcal{D}_{\mathcal{T}}^{(T)}\}$, Source model $f(.; \theta^0)$, Number of updates $I$,
    Hyperparameter $(\alpha, q, r, \gamma)$, Initialization $\tilde{\theta}^{(0)} \leftarrow \theta^{(0)}$, $\mu_{0|0} \leftarrow \theta^{(0)}$, $\Sigma_{0|0} \leftarrow 0$
**for** $t = 1, \dots, T$ **do**
    **for** $i = 1, \dots, I$ **do**
        OPTIMIZATION PROCESS:
            $\theta^{(t)} = \arg\min_{\tilde{\theta}^{(t-1)}} \hat{\mathcal{E}}_{\mathcal{T}}(\tilde{\theta}^{(t-1)}, \mathcal{D}_{\mathcal{T}}^{(t)})$           ▷ Eq. (4)
        INFERENCE PROCESS:
        // Predict Step:
        $\mu_{t|t-1} = \text{Moments}(\mu_{t-1|t-1}, \phi^{(0)}, \alpha)$           ▷ Eq. (15)
        $\Sigma_{t|t-1} = \alpha^2 \Sigma_{t-1|t-1} + q$           ▷ Eq. (18)
        // Update Step:
        $\beta_t = r/(\Sigma_{t|t-1} + r)$           ▷ Eq. (19)
        $\mu_{t|t} = \text{Moments}(\mu_{t|t-1}, \theta^{(t)}, \beta_t)$           ▷ Eq. (16)
        $\Sigma_{t|t} = \beta_t \Sigma_{t|t-1}$           ▷ Eq. (20)
        // Parameter Ensemble:
        $\tilde{\theta}^{(t)} = \text{Moments}(\theta^{(t)}, \mu_{t|t}, \gamma)$           ▷ Eq. (17)
    **end for**
**end for**

---

Fisher information matrix computed from the source model and elastic weight consolidation loss (Kirkpatrick et al., 2017). In contrast, SAR (Niu et al., 2023) does not employ additive loss but instead accesses the parameters directly, taking the strategy of averaging the loss values over time and reverting to the source model when they exceed a predetermined threshold. Similarly, ROID (Marsden et al., 2023) continuously integrates the source model into the target model during adaptation to preserve past information using the parameter ensemble method (Wortsman et al., 2022; Rame et al., 2022). These approaches maintain the functionality of specific parameter subsets corresponding to layers without altering them, making them model-agnostic and demonstrating state-of-the-art performance. However, they impose constraints on the flexibility of the target model by utilizing a frozen source model.

## 3 METHODOLOGY

This section elucidates our methodology for constructing a flexible source model while preserving past information. The KF algorithm is employed to denoise the target model parameters, combining them with the source model. First, we outline the CMF that combines the optimization process with KF-based inference (Section 3.1). We then delve into parameterization of linear Gaussian models to thwart catastrophic forgetting (Section 3.2). Subsequently, the KF algorithm for our linear Gaussian modeling is explained (Section 3.3). A simplified version of the inference process is then presented to reduce the KF parameter count, ensuring the feasibility of the CMF (Section 3.4).

### 3.1 OVERALL PROCESS

The adaptation strategy alternates between SGD-based optimization and KF-based inference. A subset of the source model parameter $\theta^{(0)} \in \mathbb{R}^d$, where $d$ is the parameter dimension, is designated for adaptation following existing studies. At the outset, the parameter of the hidden model $\phi^{(t)} \in \mathbb{R}^d$ is initialized to $\theta^{(0)}$. The optimization initially minimizes the target generalization error using an SGD-optimizer on real-time target data $\mathcal{D}_{\mathcal{T}}^{(t+1)}$, resulting in the observation $\theta^{(t+1)}$ as follows:

$$\theta^{(t+1)} = \arg\min_{\tilde{\theta}^{(t)}} \hat{\mathcal{E}}_{\mathcal{T}}(\tilde{\theta}^{(t)}, \mathcal{D}_{\mathcal{T}}^{(t+1)}). \tag{4}$$

The smooth-target model parameter $\tilde{\theta}^{(t)} \in \mathbb{R}^d$, which is initialized as $\theta^{(0)}$, is derived recursively. Subsequently, the inference process computes the posterior distribution $p(\phi^{(t)}|\theta^{(1:t)}, \phi^{(0)})$. The posterior mean is then combined with the target model to obtain $\tilde{\theta}^{(t)}$ via the parameter ensemble

method. Algorithm 1 displays a step-by-step description of our framework. Relying solely on the optimization process results in the model being recursively adapted in a self-training manner. Consequently, the target model parameters can become noisy, thereby increasing the risk of catastrophic forgetting. The inference process mitigates this issue by denoising these parameters, updating the hidden model, and re-optimizing with the smooth-target model parameter. However, a general linear Gaussian model for the inference process faces the risk of forgetting the source model during adaptation. The subsequent section introduces countermeasures.

## 3.2 PARAMETERIZATION

The LG-SSM for the inference process comprises parameterized transition and emission models. We assume the distributions that $p(\phi^{(t-1)}|\phi^{(t)})$ and $p(\phi^{(t)}|\phi^{(0)})$ are both Gaussian distribution. Let the latter as the conjugate prior to the mean of former. We can compute the posterior of $\phi^{(t)}$ through the Bayesian inference. The mean of this posterior is a convex combination of $\phi^{(t)}$ and $\phi^0$ (See Appendix A.1 for further detail). Therefore, the transition model is directly parameterized as follows:

$$p(\phi^{(t)}|\phi^{(t-1)}, \phi^{(0)}) = \mathcal{N}(\phi^{(t)}|\mathrm{A}\phi^{(t-1)} + (1 - \mathrm{A})\phi^{(0)}, Q), \tag{5}$$

where $\mathrm{A} \in \mathbb{R}^{d \times d}$ and $Q \in \mathbb{R}^{d \times d}$ are assumed to be time-independent. This choice stabilizes the inference process (Murphy, 2023). For the emission model, we choose a simple linear Gaussian model as follows:

$$p(\theta^{(t)}|\phi^{(t)}) = \mathcal{N}(\theta^{(t)}|\mathrm{H}\phi^{(t)}, R), \tag{6}$$

where $\mathrm{H} \in \mathbb{R}^{d \times d}$ and $R \in \mathbb{R}^{d \times d}$ are static, as in the transition model. The source-conjugated transition model aids in persistently infusing source information into the hidden model, thereby ensuring the retention of prior knowledge.

## 3.3 INFERENCE PROCESS

We now introduce the KF algorithm tailored for the parameterized LG-SSM. The purpose of the algorithm is to recursively determine the posterior distribution $p(\phi^{(t)}|\theta^{(1:t)}, \phi^{(0)})$ when the previous step's posterior $p(\phi^{(t-1)}|\theta^{(1:t-1)}, \phi^{(0)}) = \mathcal{N}(\phi^{(t-1)}|\mu_{t-1|t-1}, \Sigma_{t-1|t-1})$ is given. By using the posterior from the previous step and Eq. (5), the joint predictive distribution of $\phi^{(t)}$ and $\phi^{(t-1)}$ is computed and subsequently marginalized over $\phi^{(t-1)}$. The one-step-ahead predictive distribution for the hidden variables is then given by

$$p(\phi^{(t)}|\theta^{(1:t-1)}, \phi^{(0)}) = \mathcal{N}(\phi^{(t)}|\mu_{t|t-1}, \Sigma_{t|t-1}) \tag{7}$$

$$\mu_{t|t-1} = \mathrm{A}\mu_{t-1|t-1} + (1 - \mathrm{A})\phi^{(0)}, \tag{8}$$

$$\Sigma_{t|t-1} = \mathrm{A}\Sigma_{t-1|t-1}\mathrm{A}^{\top} + Q. \tag{9}$$

This phase, called the *predict step*, estimates the conditional distribution of $\phi^{(t)}$ based on the past target model parameters (See Appendix A.2 for details). Next, given target model information of the current time step, the conditional distribution of $\phi^{(t)}$ is obtained from the joint distribution of $\phi^{(t)}$ and $\theta^{(t)}$ by using Eq. (6) and Eq. (7). This phase, called the *update step*, updates the posterior distribution (detailed in Appendix A.3). The posterior distribution is obtained as follows:

$$p(\phi^{(t)}|\theta^{(1:t)}, \phi^{(0)}) = \mathcal{N}(\phi^{(t)}|\mu_{t|t}, \Sigma_{t|t}), \tag{10}$$

$$\mathrm{K}_t = \Sigma_{t|t-1}\mathrm{H}^{\top}(\mathrm{H}\Sigma_{t|t-1}\mathrm{H}^{\top} + R)^{-1}, \tag{11}$$

$$\mu_{t|t} = \mu_{t|t-1} + \mathrm{K}_t(\theta^{(t)} - \mathrm{H}\mu_{t|t-1}), \tag{12}$$

$$\Sigma_{t|t} = \Sigma_{t|t-1} - \mathrm{K}_t\mathrm{H}\Sigma_{t|t-1}. \tag{13}$$

The Kalman gain $\mathrm{K}_t$ determines the extent to which the target model parameters are updated from the hidden model during the prediction step. For the optimization process of next time step, $\tilde{\theta}^{(t)}$ is derived through the parameter ensemble method of $\mu_{t|t}$ and $\theta^{(t)}$ as follows:

$$\tilde{\theta}^{(t)} = \Gamma\theta^{(t)} + (1 - \Gamma)\mu_{t|t}, \tag{14}$$

where the ensemble hyperparameter $\Gamma \in \mathbb{R}^{d \times d}$ is specified to be between $0$ and $1$. This phase allows for the transfer of information from the refined source model to the target model.

---

We denote the mean and covariance of the posterior distribution by $\mu_{t|t}$ and $\Sigma_{t|t}$, and the mean and covariance of the one-step-ahead predictive distribution by $\mu_{t|t-1}$ and $\Sigma_{t|t-1}$.

### 3.4 SIMPLIFIED INFERENCE PROCESS

The KF algorithm requires a high computational cost because the parameter dimension of LG-SSM $(A, Q, H, R, \Gamma)$ is $d^2$ where the parameter dimension $d$ of DNNs is typically large. We use scalar parameters for the LG-SSM to reduce the computational complexity, denoted as $(\alpha, q, \eta, r, \gamma)$. In particular, we opt $\eta$ for the scalar value 1, and then Eqs. (8), (12), and (14) are can be simplified as follows:

$$\mu_{t|t-1} = \text{Moments}(\mu_{t-1|t-1}, \phi^{(0)}, \alpha), \tag{15}$$

$$\mu_{t|t} = \text{Moments}(\mu_{t|t-1}, \theta^{(t)}, \beta_t), \tag{16}$$

$$\tilde{\theta}^{(t)} = \text{Moments}(\theta^{(t)}, \mu_{t|t}, \gamma), \tag{17}$$

where $\text{Moments}(x_1, x_2, a) = ax_1 + (1 - a)x_2$,

$$\Sigma_{t|t-1} = \alpha^2 \Sigma_{t-1|t-1} + q, \tag{18}$$

$$\beta_t = r/(\Sigma_{t|t-1} + r), \tag{19}$$

$$\Sigma_{t|t} = \beta_t \Sigma_{t|t-1}. \tag{20}$$

With these simplifications, the inference process becomes computationally efficient. Also, the implementation can be streamlined, because it primarily involves repeated applications of Moments (See Appendix A.4 for futher details). Next, we detail the rationale behind our choices of each KF parameter. 1) The initialization of $(\tilde{\theta}^{(0)}, \mu_{0|0}, \Sigma_{0|0})$ is $(\theta^{(0)}, \theta^{(0)}, 0)$. 2) We opt for the same values for $0 \le \alpha \le 1$ and $0 \le \gamma \le 1$ to equalize the degree to which the source and refined source model information are obtained. 3) The properties of the simplified process are predominantly changed by $\beta_t$ because it is $1 - K_t$. To determine this value, we consider the constraint $q + r = 1$ for variances $0 < q$ and $0 < r$, to satisfy $\beta_t \le 1$. As $q$ increases, the information in the target model is more reflected in the refined source model, and conversely, as $q$ decreases, the strength of the denoising decreases; in a specific case where $q = 0$, our framework is simplified to the parameter ensemble method. In summary, only two hyperparameters $(\alpha, q)$ require modification in the proposed framework.

## 4 EXPERIMENTS

We conducted all the experiments on four random seeds using PyTorch (Paszke et al., 2019) toolkits for image classification (Marsden & Döbler, 2022) and speech recognition (Ott et al., 2019). The additional implementation and scenario details are provided in Appendix B.

**Datasets and Metric** We relied on multiple datasets to comprehensively evaluate the domain shifts, including both corruptions and natural shifts. For image classification, guided by the benchmark of Marsden et al. (2023), we selected ImageNet-C (Hendrycks & Dietterich, 2019a), ImageNet-D109 (D109) (Marsden et al., 2023), ImageNet-R (Rendition) (Hendrycks et al., 2021), and ImageNet-Sketch (Sketch) (Wang et al., 2019). ImageNet-C comprises 15 corruptions applied to ImageNet (Deng et al., 2009) validation and test images across five severity levels. D109, rooted in Domain-Net (Peng et al., 2019), includes images corresponding to 109 overlapping classes with ImageNet, showcasing six domain shifts (Rusak et al., 2022; Marsden et al., 2023). Rendition features 30,000 images demonstrates various renderings across 200 ImageNet classes. Sketch consists of 50 sketches for each of the 1,000 ImageNet classes. For speech recognition, we used LibriSpeech (Panayotov et al., 2015), TEDLIUM3 (TED) (Hernandez et al., 2018), and CommonVoice (CV) Ardila et al. (2019). LibriSpeech, which includes audio recordings of speakers reading excerpts from Project Gutenberg e-books, serves as source data. TED offers a test dataset of 0.24 hours, representing a professional corpus of topical lectures. CV, a crowdsourcing project, comprises approximately 25 h of utterances recorded by volunteers reading Wikipedia sentences. We used the average error rate for image classification and the word error rate (WER) for speech recognition for performance evaluation.

**Scenarios** The primary goal was to evaluate the performance of the universal OTTA methods across various scenarios. Guided by previous benchmarks (Gong et al., 2022; Niu et al., 2022; Marsden et al., 2023; Marsden & Döbler, 2022), our scenarios delved into covariate shifts (where the input distribution changes) and label shifts (where the label distribution undergoes alterations). We initiated the testing of data from multiple domains that underwent covariate shifts. This setting involved

Table 1: Average error rates (%) and their corresponding standard deviations in the scenario of **CS**. Red fonts indicate performance degradation.

| Method | ImageNet-C | | | | D109 | | | |
|---|---|---|---|---|---|---|---|---|
| | ResNet-50 | ViT | Swin | D2V | ResNet-50 | ViT | Swin | D2V |
| Source | 82.0 | 60.2 | 64.0 | 51.8 | 58.8 | 53.6 | 51.4 | 48.0 |
| TENT | 85.7±0.95 | 55.1±0.08 | 62.6±0.18 | 50.5±0.06 | 55.4±0.08 | 76.8±0.36 | 61.5±0.41 | 57.9±0.42 |
| CoTTA | 82.0±0.08 | 59.6±0.02 | 63.9±0.01 | 51.2±0.02 | 55.3±0.04 | 53.3±0.04 | 51.2±0.03 | 47.8±0.01 |
| RoTTA | 79.5±0.10 | 58.7±0.04 | 62.9±0.03 | 51.3±0.03 | 54.8±0.04 | 50.9±0.05 | 48.6±0.05 | 46.8±0.03 |
| SAR | 79.6±0.68 | 52.3±0.12 | 60.5±1.04 | 50.7±0.07 | 53.6±0.07 | 61.2±0.36 | 53.9±0.08 | 48.1±0.08 |
| EATA | 72.5±1.44 | 51.8±0.14 | 56.2±0.29 | 76.2±20.23 | 53.1±0.09 | 48.5±0.11 | 48.8±0.12 | 46.2±0.05 |
| ROID | 69.5±0.13 | 50.7±0.08 | 55.0±0.26 | 47.4±0.08 | 50.9±0.04 | 46.9±0.02 | 47.2±0.07 | 45.0±0.01 |
| CMF (ours) | **67.6±0.20** | **49.0±0.10** | **52.1±0.12** | **45.7±0.03** | **49.4±0.21** | **44.5±0.08** | **44.8±0.04** | **42.8±0.05** |

Table 2: Average error rates (%) and their corresponding standard deviations in the scenario of **TC-CS**. Red fonts indicate performance degradation with respect to Source.

| Method | ImageNet-C | | | D109 | | | Rendition | | | Sketch | | |
|---|---|---|---|---|---|---|---|---|---|---|---|---|
| | ViT | Swin | D2V | ViT | Swin | D2V | ViT | Swin | D2V | ViT | Swin | D2V |
| Source | 60.2 | 64.0 | 51.8 | 53.6 | 51.4 | 48.0 | 56.0 | 54.2 | 46.6 | 70.6 | 68.4 | 60.4 |
| TENT | 54.5±0.04 | 64.0±0.14 | 51.9±0.09 | 83.3±0.13 | 66.4±0.33 | 62.9±0.21 | 53.3±0.09 | 53.8±0.38 | 46.0±0.03 | 70.8±1.12 | 68.7±0.22 | 60.3±0.06 |
| CoTTA | 60.4±0.02 | 64.2±0.01 | 51.7±0.02 | 53.3±0.03 | 51.2±0.01 | 47.8±0.02 | 55.6±0.03 | 54.1±0.02 | 46.4±0.01 | 70.6±0.01 | 68.3±0.02 | 60.3±0.01 |
| RoTTA | 59.1±0.05 | 63.4±0.01 | 51.3±0.01 | 51.4±0.03 | 49.1±0.03 | 47.2±0.03 | 54.8±0.04 | 53.5±0.03 | 46.5±0.02 | 69.3±0.03 | 67.3±0.03 | 60.1±0.03 |
| SAR | 51.7±0.14 | 65.9±1.27 | 51.0±0.12 | 57.3±0.41 | 53.5±1.05 | 48.5±0.10 | 48.5±0.21 | 53.7±2.78 | 45.9±0.05 | 70.5±1.21 | 73.4±1.31 | 60.2±0.07 |
| EATA | 49.9±0.06 | 52.9±0.25 | 64.4±15.84 | 47.2±0.10 | 47.4±0.18 | 45.8±0.06 | 49.0±0.20 | 49.9±0.33 | 45.0±0.08 | 59.8±0.19 | 60.6±0.26 | 78.3±17.08 |
| ROID | 45.0±0.09 | 47.0±0.26 | 44.8±0.01 | 45.0±0.04 | 45.1±0.10 | 44.2±0.06 | 44.2±0.13 | 46.0±0.10 | 41.8±0.11 | 58.6±0.04 | 58.9±0.11 | 56.2±0.05 |
| CMF (ours) | **44.8±0.12** | **46.6±0.12** | **43.5±0.04** | **43.4±0.07** | **43.6±0.12** | **42.3±0.11** | **42.7±0.20** | **44.1±0.24** | **40.0±0.06** | **57.0±0.08** | **56.7±0.13** | **53.9±0.03** |

the analysis of both temporally uncorrelated (CS) and temporally-correlated shifts (TC-CS) during streaming. In CS, test data from all domains are randomly shuffled before adaptation, whereas in TC-CS, consecutive test samples are likely to arise from the same domain. Further, we delved into scenarios where temporal correlated label shifts (TC-LS) occurred over CS or TC-CS by adjusting the Dirichlet distribution's $\delta$ (Gong et al., 2022). As $\delta$ increases, the label distribution at time step $t$ begins to approach uniformity; a reduced $\delta$ signifies a skewed distribution towards specific classes. These scenarios are illustrated in Fig. 3. In all vision scenarios for ImageNet-C, the domains comprised 15 corruptions, each encountered at the highest severity level of 5. For D109, the domains comprised five types (clipart, infograph, painting, real and sketch). Our last scenario aimed to test the OTTA methods for speech recognition over real-world streaming data.

**Source Model and Baseline** For image classification, we employed various of models including ResNet-50 (He et al., 2016a), VisionTransformer (ViT) (Dosovitskiy et al., 2020), SwinTransformer (Swin) (Liu et al., 2021), and data2vec-vision (D2V) (Baevski et al., 2022b), which were all pre-trained on ImageNet. These modes were suited for wild world (Niu et al., 2023). The *Base* models for ViT, Swin, and D2V were chosen to balance complexity and performance. For speech recognition, the choices were data2vec *Base* model (D2V-Libri) and the data2vec *Large* model (D2V-Vox). The former is pre-trained on a LibriSpeech, while the latter is pre-trained on the entire LibriVox (Kahn et al., 2020). We compared our approach with other OTTA methods that use arbitrary off-the-shelf pre-trained models. For images, we compared TENT, LAME (Boudiaf et al., 2022), CoTTA (Wang et al., 2022), RoTTA (Yuan et al., 2023), EATA, and SAR. For speech, we compared our framework with SUTA (Lin et al., 2022), which is a state-of-the-art test-time adaptation method for speech recognition tasks.

**Implementation Details** We offer implementations for both image classification and speech recognition tasks. *Image Classification:* Our approach largely adheres to the hyperparameter guidelines put forth in Marsden & Döbler (2022) to ensure robustness and reproducibility. We designated the parameter subset $\theta^{(0)}$ as the normalization layer of each model, inspired by Niu et al. (2023). Our chosen loss function is the diversity-weighted soft likelihood ratio loss (DW-SLR) from Marsden et al. (2023), augmented with a consistency loss rooted in symmetric cross entropy (SCE). Unless specified otherwise, the hyperparameters $(\alpha, q)$ for CMF were set to $(0.99, 0.005)$. We employ prior correction for post-processing, as detailed in (Royer & Lampert, 2015; Marsden et al., 2023). *Speech Recognition:* Our approach leverages the Bitfit method from Zaken et al. (2021), which exclusively amends the self-attention bias of the transformer for a parameter subset. This methodology was selected because the correction of the normalization layer does not considerably alter performance as noted in (Lin et al., 2022). For the CMF, the hyperparameters $(\alpha, q)$ are fixed at $(0.8, 0.005)$. The post-processing step involves greedy decoding, as prescribed in (Pratap et al., 2019), and is devoid of any language model.

Table 3: Average error rates (%) and their corresponding standard deviations in the scenario of **TC-LS over TC-CS**.

| $\delta$ | Model | ImageNet-C | | | | | D109 | | | | |
|---|---|---|---|---|---|---|---|---|---|---|---|
| | | LAME | SAR | EATA | ROID | CMF (ours) | LAME | SAR | EATA | ROID | CMF (ours) |
| 0.0 | ViT | 44.1±0.02 | 48.3±0.28 | 71.8±1.22 | 16.2±0.06 | **15.9±0.04** | 35.2±0.55 | 58.5±0.40 | 58.6±1.45 | 31.4±0.07 | **31.0±0.10** |
| | Swin | 47.1±0.09 | 60.1±0.74 | 72.7±0.67 | 18.1±0.03 | **16.7±0.10** | 30.1±0.16 | 55.4±0.17 | 54.2±0.99 | 30.3±0.25 | **29.6±0.21** |
| | D2V | 38.9±0.07 | 48.3±0.15 | 58.2±2.21 | 17.4±0.21 | **14.4±0.24** | 29.7±0.15 | 49.5±0.04 | 46.1±0.37 | 29.3±0.03 | **27.8±0.12** |
| 0.01 | ViT | 83.2±0.23 | 48.7±0.29 | 47.7±0.12 | 36.3±0.08 | **35.0±0.04** | 44.8±0.69 | 58.6±0.80 | 50.7±1.20 | 32.2±0.10 | **31.8±0.10** |
| | Swin | 84.7±0.12 | 58.4±0.86 | 50.0±0.35 | 37.2±0.06 | **35.1±0.16** | 39.9±0.77 | 53.7±0.53 | 49.6±0.41 | 31.1±0.11 | **30.3±0.24** |
| | D2V | 79.5±0.20 | 47.9±0.05 | 65.0±18.58 | 35.9±0.08 | **32.7±0.04** | 39.9±0.56 | 49.1±0.14 | 47.1±1.08 | 30.7±0.09 | **28.6±0.11** |
| 0.1 | ViT | 79.9±0.06 | 48.4±0.30 | 46.1±0.17 | 41.3±0.05 | **39.6±0.03** | 68.9±0.24 | 57.7±0.56 | 47.4±0.16 | 37.3±0.12 | **36.1±0.11** |
| | Swin | 84.5±0.09 | 58.4±0.75 | 48.3±0.09 | 42.1±0.04 | **39.6±0.02** | 64.6±0.25 | 53.4±0.70 | 47.4±0.21 | 36.9±0.11 | **35.0±0.05** |
| | D2V | 70.1±0.04 | 48.0±0.04 | 65.5±19.11 | 41.3±0.03 | **38.2±0.05** | 64.6±0.25 | 48.6±0.04 | 45.7±0.08 | 36.3±0.06 | **34.1±0.13** |
| 1.0 | ViT | 80.0±0.03 | 48.3±0.25 | 45.7±0.15 | 41.2±0.03 | **39.4±0.03** | 90.0±0.09 | 57.4±0.12 | 47.2±0.04 | 42.9±0.03 | **41.3±0.06** |
| | Swin | 84.6±0.06 | 58.5±0.41 | 47.4±0.39 | 41.9±0.03 | **39.4±0.11** | 86.9±0.24 | 54.5±0.68 | 47.4±0.10 | 43.0±0.06 | **41.3±0.04** |
| | D2V | 70.2±0.07 | 47.9±0.09 | 87.0±18.44 | 41.2±0.01 | **38.1±0.03** | 88.3±0.13 | 48.5±0.09 | 45.7±0.04 | 42.2±0.04 | **40.1±0.10** |
| 5.0 | ViT | 80.2±0.09 | 55.5±12.62 | 45.6±0.17 | 41.3±0.03 | **39.5±0.03** | 93.3±0.17 | 57.3±0.22 | 47.2±0.08 | 43.9±0.09 | **42.5±0.08** |
| | Swin | 84.9±0.04 | 59.2±0.68 | 47.6±0.25 | 41.9±0.03 | **39.4±0.08** | 90.6±0.23 | 54.0±0.72 | 47.3±0.05 | 44.1±0.06 | **42.5±0.07** |
| | D2V | 70.5±0.12 | 47.9±0.08 | 65.9±18.92 | 41.2±0.03 | **38.0±0.05** | 92.8±0.16 | 48.4±0.12 | 45.7±0.06 | 43.2±0.04 | **41.1±0.06** |

## 4.1 RESULTS

**Covariate shifts** Table 1 lists the average error rates for each OTTA method in the scenario of CS. TENT exhibited a performance dip compared to the source model for both ImageNet-C and D109. SAR in D109 and EATA in ImageNet-C also exhibited degraded performance. RoTTA, CoTTA, and ROID, on the other hand, demonstrated relatively consistent results, with ROID standing out as the top performer. Among all the methods, our CMF framework exhibited the lowest average error rate. In Table 2, where the error rates under the TC-CS conditions are detailed, TENT exhibited a substantial decline in performance. CoTTA and RoTTA, which were previously robust, also displayed performance setbacks. EATA held steady performance in all except for D2V, yet it could not surpass ROID's performance. CMF continued its trend by recording the lowest average error rate, even outperforming ROID.

**Covariate and Label shifts** Table 3 presents the average error rates for each OTTA method in the context of TC-LS superimposed on TC-CS, indicating that the labels are becoming temporally correlated. In instances with the strongest temporal correlation (i.e., $\delta = 0.0$), all methods barring LAME and ROID revealed inconsistent outcomes with performance degradation. The latter exhibited superior results between LAME and ROID, except for Swin. As the temporal correlation diminished, the performance of LAME decreased noticeably, but ROID showed robust performance. The CMF still managed to eclipse ROID and LAME. Table 4 highlights TC-LS under CS following Marsden & Döbler (2022) for ImageNet-C and D109 with $\delta$ values of 0.01 and 0.1, respectively. CMF maintained its superiority across all datasets and models. Thus, the CMF exhibited unwavering robustness, even in the TC-LS scenarios.

Table 4: Average error rates (%) and their corresponding standard deviations in the scenario of **TC-LS over CS**.

| Method | ImageNet-C | | | D109 | | |
|---|---|---|---|---|---|---|
| | ViT | Swin | D2V | ViT | Swin | D2V |
| LAME | 36.1±0.09 | 37.4±0.12 | 36.3±0.11 | 29.9±0.18 | 28.6±0.23 | 29.1±0.19 |
| SAR | 54.1±0.40 | 65.4±0.53 | 47.2±0.08 | 61.0±0.51 | 53.6±0.24 | 48.6±0.35 |
| EATA | 70.5±0.67 | 77.1±0.93 | 85.8±18.90 | 52.9±2.98 | 50.3±0.25 | 45.9±0.13 |
| ROID | 23.6±0.05 | 28.6±0.16 | 18.8±0.01 | 29.1±0.09 | 28.2±0.05 | 26.3±0.07 |
| CMF (ours) | **23.2±0.05** | **27.1±0.08** | **17.1±0.09** | **28.7±0.19** | **27.3±0.05** | **24.9±0.10** |

**Distribution shifts on real-world streaming data** Table 5 presents the OTTA outcomes for the real-world streaming datasets. Both TED and CV deviate from the source domain, LibriSpeech, in terms of the recording environment and vocabulary domains. This results in the simultaneous presence of covariate and label

Table 5: Average WERs (%) and their corresponding standard deviations in **real-world streaming** scenario.

| Method | TED | | CV | |
|---|---|---|---|---|
| | D2V-Libri | D2V-VOX | D2V-Libri | D2V-VOX |
| Source | 12.2 | 8.5 | 33.4 | 20.6 |
| SUTA-cont. | 67.7±1.70 | 66.1±0.36 | 120.89±4.03 | 130.3±1.88 |
| SUTA-episodic | 12.0±0.03 | 8.0±0.03 | 30.3±0.01 | 18.9±0.01 |
| CMF (ours) | **11.8±0.05** | **7.9±0.02** | **29.6±0.02** | **18.7±0.03** |

shifts. The leading OTTA method for speech recognition tasks, SUTA, employs test-time adaptation for individual utterances using an episodic strategy (SUTA-episodic). However, a sharp decrease in performance was observed when this strategy was applied in a continual setting (SUTA-cont.). In these continuous settings, the CMF not only thwarted catastrophic forgetting but also displayed

enhanced performance compared to SUTA-episodic. Our framework showed robustness against the intricate distribution shifts found in real-world contexts.

## 4.2 ABLATION STUDY

We now delve into the essential components of the CMF. Detailed discussions, such as diversity and computational cost, can be found in Appendix C.

**Effectiveness of leveraging the noisy target model** Table 6 lists the average error rates of the CMF in CS and TC-CS scenarios for both ImageNet-C and D109 datasets. We examined the performance variations with respect to the value of $q$, which modulates the information from the noisy target model. Using the entropy loss of TENT, the CMF demonstrated superior average performance compared to TENT alone. However, this performance gain was only evident for D109, which suffered markedly from catastrophic forgetting, as shown in Tables 1 and 2. When comparing the outcomes for $q = 0.000$ and $q = 0.005$ on ImageNet-C, our model recorded a higher average error rate without utilizing the target model (i.e., $q = 0.000$), a trend reversed for D109. This highlights the limitation of relying solely on the entropy loss (Nevertheless, the entropy version of CMF significantly outperforms TENT when adjusting for diversity, which is

Table 6: Average error rates (%) for $q = 0.000$ and $q = 0.005$ in the **CS** and **TC-CS** scenarios.

| Model | Method | | $q$ | ImagNet-C | | D109 | | AVG |
|---|---|---|---|---|---|---|---|---|
| | | | | CS | TC-CS | CS | TC-CS | |
| ViT | TENT | Entropy | - | 55.1 | 54.5 | 76.8 | 83.3 | 67.4 |
| | CMF | Entropy | 0.000 | 58.6 | 57.4 | 53.6 | 54.5 | 56.0 |
| | | | 0.005 | 57.1 | 56.0 | 57.3 | 60.9 | 57.8 |
| | | DW-SLR | 0.000 | 52.7 | 47.8 | 47.1 | 46.0 | 48.4 |
| | | | 0.005 | 51.2 | 47.1 | 45.2 | 44.4 | 47.0 |
| | | + SCE | 0.000 | 50.7 | 45.0 | 46.9 | 45.0 | 46.9 |
| | | | 0.005 | **48.9** | **44.8** | **44.5** | **43.4** | **45.4** |
| Swin | TENT | Entropy | - | 62.6 | 64.0 | 61.5 | 66.4 | 63.6 |
| | CMF | Entropy | 0.000 | 64.4 | 64.2 | 51.5 | 51.6 | 57.9 |
| | | | 0.005 | 64.3 | 64.0 | 51.6 | 52.5 | 58.1 |
| | | DW-SLR | 0.000 | 57.5 | 51.1 | 49.4 | 46.3 | 51.1 |
| | | | 0.005 | 57.5 | 48.0 | 45.9 | 44.5 | 49.0 |
| | | + SCE | 0.000 | 55.0 | 47.0 | 47.2 | 45.1 | 48.6 |
| | | | 0.005 | **52.1** | **46.6** | **44.8** | **43.6** | **46.8** |
| D2V | TENT | Entropy | - | 50.5 | 51.9 | 57.9 | 62.9 | 55.8 |
| | CMF | Entropy | 0.000 | 52.3 | 52.9 | 48.1 | 48.5 | 50.4 |
| | | | 0.005 | 51.8 | 52.7 | 48.9 | 50.6 | 51.0 |
| | | DW-SLR | 0.000 | 49.0 | 45.9 | 46.9 | 45.4 | 46.8 |
| | | | 0.005 | 47.2 | 45.4 | 43.7 | 43.3 | 44.9 |
| | | + SCE | 0.000 | 47.4 | 44.8 | 45.0 | 44.2 | 45.4 |
| | | | 0.005 | **45.6** | **43.5** | **42.8** | **42.3** | **43.6** |

detailed in Appendix C.2). This inconsistency was addressed by incorporating DW-SLR (Marsden et al., 2023), an entropy-based loss function that consider diversity. For both datasets, the CMF coupled with DW-SLR exhibited superior performance improvements when $q = 0.005$ compared to $q = 0.000$. Similarly, SCE loss, which modulated diversity via augmentation, further enhanced performance. These results were consistent with the findings that a convex combination of parameters (i.e., Moments) from out-of-domain can escalate the generalization error if the diversity is not moderate (Wortsman et al., 2022; Rame et al., 2022). In essence, the CMF consistently boosts performance when diversity is appropriately managed.

**Effectiveness of the source-conjugated transition model** Figure 2 shows the average error rates for each model contingent on variations in $\alpha$, which determine the sway of the source model in the source-conjugated transition model. A comparison of the error rates for $\alpha = 1$ (devoid of source model influence) and $\alpha = 0.99$ (factoring in the source model influence) revealed enhanced performance across all models. However, as $\alpha$ increased, the performance metrics for both Swin and D2V models exhibited a downward trajectory. These findings underscore the significance of the moderated influence of the source model.

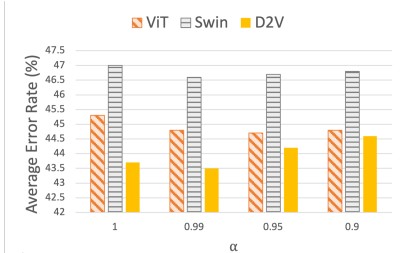

Figure 2: Average error rates (%) as decreasing $\alpha$ where $\gamma = 0.99$ in the scenarios of **TC-CS** on ImageNet-C.

## 5 CONCLUSION

We proposed the CMF, which a novel approach to bolster the OTTA methodology. This was achieved by deducing a refined source model through target model denoising by leveraging the KF. By streamlining the KF algorithm, the computational overheads were minimized, underpinning the pragmatic nature of the CMF. Our strategy withstood a rigorous evaluation across a spectrum of scenarios previously subjected to state-of-the-art methods, consistently demonstrating marked performance advancements. Its efficacy in real-time streaming contexts, such as speech recognition tasks, suggests its potential as an integral facet of established frameworks across diverse applications.

## REPRODUCIBILITY STATEMENT

In this paper, we conducted experiments based on the official GitHub code of the toolkits mentioned in Section 4 for image classification and speech recognition. Appendix B lists the download URLs of each toolkit, the adopted datasets, and the pre-trained source models and mentions more experimental details. The code is available at `https://github.com/j-pong/CMF`.

## ACKNOWLEDGEMENTS

This work was supported by the National Research Foundation of Korea(NRF) grant funded by the Korea government(MSIT) (No. RS-2023-00302424). This work was supported by Institute of Information & communications Technology Planning & Evaluation (IITP) grant funded by the Korea government(MSIT) (No.2021-0-00456, Development of Ultra-high Speech Quality Technology for Remote Multi-speaker Conference System).

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

# A    DERIVATIONS

For completeness, we included derivations for each component of the CMF inference process. These derivations follow Särkkä & Svensson (2023) and conform to the notations in Murphy (2023). All parameters $(S_1, S_2, A, Q, H, R)$ are in $\mathbb{R}^{d \times d}$.

## A.1    SOURCE-CONJUGATED TRANSITION MODEL

Given $S_1$ and $S_2$ are known constants, we assume that the likelihood for $\phi^{(t-1)}$ is

$$p(\phi^{(t-1)}|\phi^{(t)}) = \mathcal{N}(\phi^{(t-1)}|\phi^{(t)}, S_1), \tag{21}$$

and the conjugate prior of the likelihood is

$$p(\phi^{(t)}|\phi^{(0)}) = \mathcal{N}(\phi^{(t)}|\phi^{(0)}, S_2). \tag{22}$$

The conditional distribution of $\phi^{(t)}$ is obtained using Lemma A.3 from Särkkä & Svensson (2023)

$$p(\phi^{(t)}|\phi^{(t-1)}, \phi^{(0)}) = \mathcal{N}(\phi^{(t)}|A\phi^{(t-1)} + (1-A)\phi^{(0)}, Q). \tag{23}$$

where

$$\begin{aligned} A &= S_1(S_2 + S_1)^{-1}, \\ Q &= S_1 S_2 (S_2 + S_1)^{-1}. \end{aligned} \tag{24}$$

We directly parameterize the transition model with $(A, Q)$ rather than by using $(S_1, S_2)$.

## A.2    PREDICT STEP

The posterior of the previous step is given by

$$p(\phi^{(t-1)}|\theta^{(1:t-1)}, \phi^{(0)}) = \mathcal{N}(\phi^{(t-1)}|\mu_{t-1|t-1}, \Sigma_{t-1|t-1}). \tag{25}$$

According to Lemma A.2 from Särkkä & Svensson (2023), Eqs. (5) and (25), the joint distribution of $\phi^{(t)}$ and $\phi^{(t-1)}$ given $\theta^{(1:t-1)}, \phi^{(0)}$ is

$$\begin{aligned} p(\phi^{(t)}, \phi^{(t-1)}|\theta^{(1:t-1)}, \phi^{(0)}) &= p(\phi^{(t)}|\phi^{(t-1)}, \phi^{(0)})p(\phi^{(t-1)}|\theta^{(1:t-1)}) \\ &= \mathcal{N}(\phi^{(t)}|A\phi^{(t-1)} + (1-A)\phi^{(0)}, Q)\mathcal{N}(\phi^{(t-1)}|\mu_{t-1|t-1}, \Sigma_{t-1|t-1}) \\ &= \mathcal{N}\left(\begin{pmatrix} \phi^{(t-1)} \\ \phi^{(t)} \end{pmatrix} \Big| \mu', \Sigma'\right), \end{aligned} \tag{26}$$

where

$$\begin{aligned} \mu' &= \begin{pmatrix} \mu_{t-1|t-1} \\ A\mu_{t-1|t-1} + (1-A)\phi^0 \end{pmatrix}, \\ \Sigma' &= \begin{pmatrix} \Sigma_{t-1|t-1} & \Sigma_{t-1|t-1}A^\top \\ A\Sigma_{t-1|t-1} & A\Sigma_{t-1|t-1}A^\top + Q \end{pmatrix}, \end{aligned} \tag{27}$$

and the marginal distribution of $\phi^{(t)}$ is obtained using Lemma A.3 from Särkkä & Svensson (2023)

$$p(\phi^{(t)}|\theta^{(1:t-1)}, \phi^{(0)}) = \mathcal{N}(\phi^{(t)}|\mu_{t|t-1}, \Sigma_{t|t-1}), \tag{28}$$

where

$$\begin{aligned} \mu_{t|t-1} &= A\mu_{t-1|t-1} + (1-A)\phi^{(0)}, \\ \Sigma_{t|t-1} &= A\Sigma_{t-1|t-1}A^\top + Q. \end{aligned} \tag{29}$$

## A.3 UPDATE STEP

According to Lemma A.2 from Särkkä & Svensson (2023), Eqs. (23) and (28), the joint distribution of $\phi^{(t)}$ and $\theta^{(t)}$ is

$$
\begin{aligned}
p(\phi^{(t)}, \theta^{(t)} | \theta^{(1:t-1)}, \phi^{(0)}) &= p(\theta^{(t)} | \phi^{(t)}) p(\phi^{(t)} | \theta^{(1:t-1)}, \phi^{(0)}) \\
&= \mathcal{N}(\theta^{(t)} | \mathrm{H}\phi^{(t)}, R) \mathcal{N}(\phi^{(t)} | \mu_{t|t-1}, \Sigma_{t|t-1}) \\
&= \mathcal{N}\left( \begin{pmatrix} \phi^{(t)} \\ \theta^{(t)} \end{pmatrix} \Big| \mu'', \Sigma'' \right),
\end{aligned}
\tag{30}
$$

where

$$
\begin{aligned}
\mu'' &= \begin{pmatrix} \mu_{t-1|t-1} \\ \mathrm{H}\mu_{|t-1} \end{pmatrix}, \\
\Sigma'' &= \begin{pmatrix} \Sigma_{t|t-1} & \Sigma_{t|t-1}\mathrm{H}^\top \\ \mathrm{H}\Sigma_{t|t-1} & \mathrm{H}\Sigma_{t|t-1}\mathrm{H}^\top + R \end{pmatrix},
\end{aligned}
\tag{31}
$$

and the conditional distribution of $\phi^{(t)}$ is obtained using Lemma A.3 from Särkkä & Svensson (2023)

$$
p(\phi^{(t)} | \theta^{(1:t)}, \phi^{(0)}) = \mathcal{N}(\phi^{(t)} | \mu_{t|t}, \Sigma_{t|t}),
\tag{32}
$$

where

$$
\begin{aligned}
\mathrm{K}_t &= \Sigma_{t|t-1}\mathrm{H}^\top (\mathrm{H}\Sigma_{t|t-1}\mathrm{H}^\top + R)^{-1}, \\
\mu_{t|t} &= \mu_{t|t-1} + \mathrm{K}_t(\theta^{(t)} - \mathrm{H}\mu_{t|t-1}), \\
\Sigma_{t|t} &= \Sigma_{t|t-1} - \mathrm{K}_t\mathrm{H}\Sigma_{t|t-1}.
\end{aligned}
\tag{33}
$$

## A.4 SIMPLIFICATION

We chose $(\mathrm{A}, Q, \mathrm{H}, R)$ to scalar values $(\alpha, q, 1, 1-q)$. Thus, Eq. (29) is

$$
\begin{aligned}
\mu_{t|t-1} &= \alpha\mu_{t-1|t-1} + (1-\alpha)\phi^{(0)}, \\
\Sigma_{t|t-1} &= \alpha^2\Sigma_{t-1|t-1} + q,
\end{aligned}
\tag{34}
$$

and Eq. (33) is

$$
\begin{aligned}
\beta_t &= (1-q)/(\Sigma_{t|t-1} + 1 - q), \\
\mu_{t|t} &= \beta_t\mu_{t|t-1} + (1-\beta_t)\theta^{(0)}, \\
\Sigma_{t|t} &= \beta_t\Sigma_{t|t-1}.
\end{aligned}
\tag{35}
$$

where $\beta_t = 1 - \mathrm{K}_t$. Consequently, $\mu_{t|t-1}, \Sigma_{t|t-1}, \mu_{t|t}, \Sigma_{t|t}$ and $\beta_t$ are scalars.

# B DETAILS OF IMPLEMENTATION

## B.1 EXPERIMENTS OF IMAGE CLASSIFICATION

We focused on the numerous classes, potential corruption, and natural distribution shifts that can occur in the wild world (Niu et al., 2023). ImageNet-C is a standard TTA benchmark used to evaluate robustness against corruption. ImageNet contains 1,281,167/50,000 training/testing data respectively. ImageNet-C is a dataset applied to ImageNet according to 15 types of damage (gaussian noise, shot noise, impulse noise, defocus blur, glass blur, motion blur, zoom blur, snow, frost, fog, brightness, contrast, elastic transform, pixelate, and jpeg compression) at five severity levels. Each corruption was considered a domain, and severity level of 5 has been selected. D109 is a dataset concerning natural distribution shifts provided by Marsden et al. (2023). This dataset consists of five domains (clipart, infograph, painting, real, and sketch). D109 classes were based on DomainNet,

---

https://zenodo.org/record/2235448#.Yj2RO_co_mF
http://ai.bu.edu/M3SDA/

including 109 classes that overlapp with ImageNet. For other natural shifts, we used the Rendition and Sketch datasets. Each dataset consisted only of domains and was used in the TC-CS scenario. Rendition included 30,000 images with various artistic renderings of 200 ImageNet classes, primarily collected from Flickr and filtered by Amazon MTurk annotators. Sketch dataset consists of 50,000 images, with 50 images for each of the 1,000 ImageNet classes. This dataset is constructed from Google image queries with the standard class name "sketch of" and is only searched within the "black and white" color scheme.

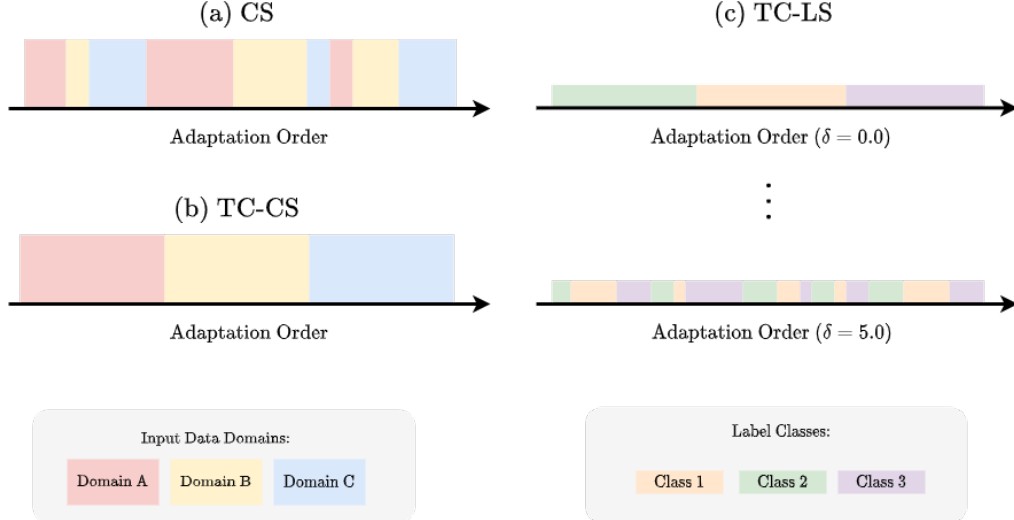

Figure 3: Illustration of the OTTA scenarios.

Four scenarios were considered: covariate shifts (CS), temporally-correlated covariate shifts (TC-CS), temporally-correlated label shifts (TC-LS) over CS, and TC-CS over TC-CS. CS involves previously defined domains streamed into the model, which were all mixed up. Hence, the likelihood of temporally adjacent input data being drawn from the same domain is low Niu et al. (2022). Conversely, in the TC-CS scenario, each domain was applied sequentially as previously described; therefore, temporally adjacent input data were likely to belong to the same domain. Meanwhile, TC-LS simulates the characteristics of real online data, where labels are temporally interrelated, and data of the same class appears multiple times (Gong et al., 2022). This methodology entails the creation of a non-i.i.d partition for a given number of tokens across a set number of classes. The Dirichlet distribution, driven by the concentration parameter $\delta > 0$, determines how data from each class were allocated to each token.

We implemented the CMF method based on the benchmark in Marsden & Döbler (2022) and used the associated default hyperparameters. For all the vision experiments, the learning rate was set to 0.00025 for ViT and Swin, 0.0002 for ResNet-50, and 0.00001 for D2V. Unless specifically mentioned, we used $(\alpha, q) = (0.99, 0.005)$ for CMF. Furthermore, because each model had a different number of learnable parameters, we adjusted it by multiplying the ratio of each model's number of learnable parameters to that of ViT by $q$. The SGD optimizer was used for model training in all scenarios. The iteration $I$ for the adaptation was set to 1.

---

https://github.com/hendrycks/imagenet-r
https://github.com/HaohanWang/ImageNet-Sketch
https://github.com/mariodoebler/test-time-adaptation
https://pytorch.org/vision/0.14/models/generated/torchvision.models.
vit_b_16.html#torchvision.models.vit_b_16
https://pytorch.org/vision/0.14/models/generated/torchvision.models.
swin_b.html#torchvision.models.swin_b
https://pytorch.org/vision/0.14/models/generated/torchvision.models.
resnet50.html#torchvision.models.resnet50
https://huggingface.co/facebook/data2vec-vision-base-ft1k

### B.2 Experiments of speech recognition

The LibriSpeech dataset, which is widely used for pre-training in speech recognition, contains audio recordings of speakers reading excerpts from Project Gutenberg e-books. It included 1,000 hours of utterances, with subsets of 100 h, 360 h, and 500 h for training. D2V-Libri was trained both self-supervised and supervised on the 960h subset. Libri-Light is a collection of English voice audios suitable for training speech recognition systems with limited or no supervision. It is derived from the open-source audiobooks of the LibriVox project, which contain over 60,000 h of audio. D2V-Vox was self-supervised trained on Libri-Light and supervised trained on LibriSpeech 960h.

We used TED and CV as the target data, each representing a different scenario. TED provide an official test dataset of lectures covering various topics and areas. This test dataset was divided into 11 datasets per speaker, each of which had a different domain. This simulated an actual situation where professionals speak continuously and at length. The average duration of utterances per speaker in the test dataset was 0.24 h. CV is a crowdsourcing project supported by volunteers who read Wikipedia sentences and record samples at 48kHz. The sampling rate is resampled to 16kHz to match the learning conditions of the source model. We used the Common Voice Corpus 5.1 (dated 7/14/2020). Unlike TED, the test dataset was not divided per speaker. This test dataset contained approximately 25 h of utterances, simulating an extremely long utterance environment.

A learning rate of $0.0006$ was selected for D2V-Libri and D2V-Vox. We set $(\alpha, q)$ to $(0.8, 0.005)$ to maintain the CMF approach used in the vision experiments. The Adam optimizer was used for model training in all speech recognition experiments. The number of iteration $I$ for the adaptation followed Lin et al. (2022) using 10. We also applied the masking augmentation (Hsu et al., 2021; Chen et al., 2021; Schneider et al., 2019; Baevski et al., 2019; 2020; 2022a) four times, considering diversity. For a fair comparison, both comparison targets, SUTA-cont. and SUTA-episodic, used the same learning rate and $I$, and the same augmentation was applied.

## C  Additional Ablation Study

### C.1  Efficiency of Methods

Table 7: Comparison of the computation efficiency for ResNet-50 on Rendition in the scenario of **TC-CS**.

| Method | Average Error Rate (%) | #Forwards | #Backwards | Train Param (%) |
|--------|------------------------|-----------|------------|-----------------|
| Source | 63.8 | 30,000 | - | - |
| LAME | 99.4 | 30,000 | - | - |
| TENT | 57.4 | 30,000 | 30,000 | 0.21 |
| CoTTA | 57.4 | 90,000 | 30,000 | 100 |
| RoTTA | 60.8 | 90,000 | 30,000 | 0.21 |
| SAR | 57.2 | 46,279 | 30,111 | 0.12 |
| EATA | 54.2 | 30,000 | 5,440 | 0.21 |
| ROID | 51.4 | 48,610 | 37,220 | 0.21 |
| CMF | 50.7 | 48,610 | 37,220 | 0.21 |

Table 7 provides an overview of the computational efficiency of the different methods, measured by the number of forward and backward passes required during the Table 7 provides an overview of the computational efficiency of the different methods, measured by the number of forward and backward passes required during the test-time adaptation. CoTTA and RoTTA, owing to their reliance on the teacher-student modeling approach (Hinton et al., 2015; Xie et al., 2020; Sohn et al., 2020; Berthelot et al., 2019), these methods require three forward passes: two for the teacher model and one for the

---

https://www.openslr.org/12
https://github.com/facebookresearch/libri-light
https://www.openslr.org/51/
https://commonvoice.mozilla.org/en/datasets
https://dl.fbaipublicfiles.com/fairseq/data2vec/audio_base_ls_960h.pt
https://dl.fbaipublicfiles.com/fairseq/data2vec/vox_960h.pt
https://github.com/DanielLin94144/Test-time-adaptation-ASR-SUTA

student model. Correspondingly, one backward pass was required for the teacher model. EATA selectively filtered the samples during learning. This filtering mechanism ensured the superiority of EATA in efficiency, demanding fewer backward passes than TENT. SAR requires more forward and backward passes than TENT because it incorporates a general forward and a forward with filtering. ROID adopts a filtering strategy (i.e., DW-SLR loss) akin to EATA while incorporating data augmentation driven by SCE loss, resulting in an augmented count of forward and backward passes relative to TENT. By leveraging both DW-SLR and SCE losses, the CMF mirrors ROID in terms of efficiency. We also investigate the memory cost for ROID/CMF: each method consumes 7841MiB/8073MiB for D2V and 10339MiB/10667MiB for Swin. The CMF requires slightly more memory than ROID because it uses hidden variables. However, it does not require much additional computation because inference process of the CMF does not perform backward, which dominates the memory cost. To summarize, the superior performance of CMF is noteworthy, even with similar computational and memory requirements as ROID.

### C.2 DIVERSITY DIAGNOSIS AND PRESCRIPTION

A vital factor of the CMF framework is the parameter diversity, which gauges how much target models deviate from the source model. A moderate level of diversity often indicates that our framework is likely to operate more effectively. To quantify this diversity, we employed cosine similarity metrics between the source and target models across different domains. The diversity was calculated as the inverse of this similarity, specifically as $1 -$ Average Cosine Similarity where Average Cosine Similarity is calculated for a domain in the dataset.

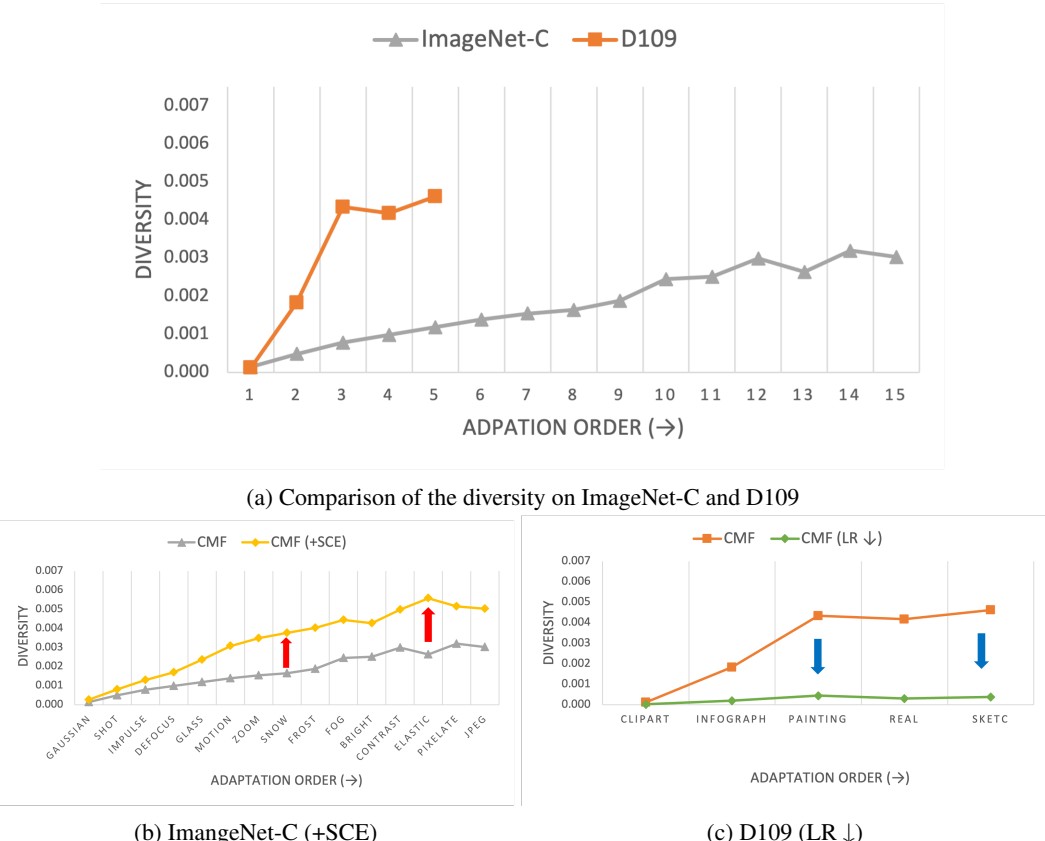

(a) Comparison of the diversity on ImageNet-C and D109

(b) ImangeNet-C (+SCE)

(c) D109 (LR ↓)

Figure 4: Comparison of the diversity for ViT in the scenario of **TC-CS**.

Figure 4 provided insights into the diversity variation across different data sets and scenarios, explicitly focusing on ViT model. We also explored how diversity changes when additional SCE loss functions (+SCE) are introduced or when the learning rate is decreased (LR ↓). When the "LR ↓" method was applied, the learning rate was set to $0.0001$ for the ViT/Swin model, down from an

initial value of 0.00025, and to 0.000005 for the D2V model, which started at 0.00001. Our analysis revealed that the D109 dataset exhibits significantly higher diversity than the ImageNet-C dataset, as shown in Figure 4 (a). When SCE loss was applied to the ImageNet-C dataset, which showed lower diversity, an increase in diversity was observed (Figure 4 (b)). Conversely, in the case of the D109 dataset, which presented a higher diversity level, reducing the learning rate decreased diversity (Figure 4 (c)).

Table 8: Comparison of average error rates (%) for each prescription in the scenario of **TC-CS**.

| Dataset | Method | Prescription | Average Error Rate (%) | | |
|---------|--------|--------------|------|------|------|
| | | | ViT | Swin | D2V |
| ImageNet-C | TENT | - | 54.5 | 64.0 | 51.9 |
| | CMF | - | 56.0 | 64.0 | 52.7 |
| | | + SCE | **53.6** | **60.1** | **51.5** |
| D109 | TENT | - | 83.3 | 66.4 | 62.9 |
| | CMF | - | 60.9 | 52.5 | 50.6 |
| | | LR ↓ | **54.1** | **51.7** | **48.6** |

This manipulation of diversity through adjustments, the introduction of SCE loss or the reduction in learning rates, has been shown to positively impact the performance of different models, as evidenced in Table 8. Thus, managing diversity was a crucial strategy for optimizing the functionality of the CMF approach. The pain of managing diversity was largely solved by using DW-SLR loss, as discussed in Section 4.2.

## C.3 LIMITATION OF FIXED SOURCE MODEL

Table 9: Comparison of average error rates (%) for WE and CMF in the scenario of **TC-CS**.

| Method | Model | ImageNet-C | D109 | Rendition | Sketch |
|--------|-------|------------|------|-----------|--------|
| DW-SLR+SCE | ViT | 47.6±0.06 | 48.3±0.92 | 44.6±0.25 | 58.4±0.05 |
| | Swin | 51.1±0.30 | 50.0±0.10 | 47.0±0.14 | 59.1±0.18 |
| | D2V | 46.3±0.15 | 47.4±0.41 | 42.2±0.13 | 55.6±0.07 |
| DW-SLR+SCE+WE | ViT | 45.0±0.09 | 45.0±0.04 | 44.2±0.13 | 58.6±0.04 |
| | Swin | 47.0±0.26 | 45.1±0.10 | 46.0±0.10 | 58.9±0.11 |
| | D2V | 44.8±0.01 | 44.2±0.06 | 41.8±0.11 | 56.2±0.05 |
| DW-SLR+SCE+CMF | ViT | **44.8±0.12** | **43.4±0.07** | **42.7±0.20** | **57.0±0.08** |
| | Swin | **46.6±0.12** | **43.6±0.12** | **44.1±0.24** | **56.7±0.13** |
| | D2V | **43.5±0.04** | **42.3±0.11** | **40.0±0.06** | **53.9±0.03** |

Conventional weight ensemble (WE) methods utilize a fixed source model. In contrast, the CMF approach updates the source model. To evaluate the performance of both WE and CMF, we set the loss to DW-SLR + SCE and measured their respective performances in TC-CS scenario.

When WE is added, significant performance improvements in datasets such as Sketch or Rendition are not readily observable. Conversely, substituting WE with CMF leads to substantial performance enhancements across all datasets, including the two aforementioned, and notably outperforms WE in the ImageNet-C and D109 datasets. These results highlight the limitations of the fixed source model and clearly delineate the contribution of CMF, which continually updates the hidden model with both the target and source model information.

## C.4 SENSITIVITY OF HYPERPARAMETERS IN CMF

Table 10: Comparison of average error rates (%) for sensitivity of $q$ in the scenario of **TC-CS**.

| Model | Source | TENT | CMF ($q$) | | | | |
|---|---|---|---|---|---|---|---|
| | | | 0.01 | 0.0075 | 0.005 | 0.0025 | 0.0001 |
| ViT | 60.2 | 54.5±0.04 | 44.9±0.09 | 44.9±0.09 | **44.8±0.12** | **44.8±0.14** | 44.9±0.05 |
| Swin | 64.0 | 64.0±0.14 | 46.7±0.25 | **46.5±0.05** | 46.6±0.12 | **46.5±0.18** | 46.9±0.18 |
| D2V | 51.8 | 51.9±0.09 | **43.4±0.08** | 43.4±0.09 | 43.5±0.04 | 43.7±0.15 | 44.5±0.02 |

Table 11: Comparison of average error rates (%) for sensitivity of $\alpha$ in the scenario of **TC-CS**.

| Model | Source | TENT | CMF ($\alpha$) | | | | |
|---|---|---|---|---|---|---|---|
| | | | 0.999 | 0.99 | 0.95 | 0.9 | 0.8 |
| ViT | 60.2 | 54.5±0.04 | 45.2±0.03 | 44.8±0.12 | **44.7±0.06** | 44.9±0.09 | 44.9±0.06 |
| Swin | 64.0 | 64.0±0.14 | 46.9±0.12 | **46.6±0.12** | 46.7±0.13 | 46.8±0.24 | 47.1±0.20 |
| D2V | 51.8 | 51.9±0.09 | 43.6±0.17 | **43.5±0.04** | 44.2±0.11 | 44.6±0.04 | 44.8±0.04 |

Table 12: Comparison of average error rates (%) for sensitivity of $\gamma$ in the scenario of **TC-CS**.

| Model | Source | TENT | CMF ($\gamma$) | | | | |
|---|---|---|---|---|---|---|---|
| | | | 0.999 | 0.99 | 0.95 | 0.9 | 0.8 |
| ViT | 60.2 | 54.5±0.04 | 45.2±0.11 | **44.8±0.12** | 45.5±0.09 | 46.7±0.07 | 49.2±0.04 |
| Swin | 64.0 | 64.0±0.14 | 47.1±0.21 | **46.6±0.12** | 47.2±0.14 | 48.5±0.20 | 50.7±0.28 |
| D2V | 51.8 | 51.9±0.09 | **43.2±0.22** | 43.5±0.04 | 45.3±0.04 | 46.8±0.06 | 48.6±0.11 |

An analysis was conducted to understand the impact of adjusting hyperparameters $(q, \alpha, \gamma)$ in CMF, as outlined in Algorithm 1. The default experimental settings used were $(q = 0.005, \alpha = 0.99, \gamma = 0.99)$. By varying one parameter while keeping the others constant, the experiments, as shown in Tables 10, 11, and 12, revealed that the impact of $\gamma$, which directly influences the target model, was the most significant. In contrast, the effects of $q$ and $alpha$ were comparatively lesser. Despite these hyperparameter variations, CMF still demonstrated superior performance compared to Source and TENT.

## D LIMITATIONS

Our proposed CMF relies on the KF that uses a linear Gaussian model. In the context of CMF, this linear Gaussian model employs a linear system for the transition and emission of DNN's parameters. While such linear systems have been frequently used in previous studies (Izmailov et al., 2018; Garipov et al., 2018; Guo et al., 2023) and have shown superior performance in various applications as demonstrated in our paper, it is challenging to guarantee that all applications will depend on this linear system. According to the sensitivity analysis introduced in Appendix C.4, we observed that our model is relatively more influenced by the hyperparameter $\gamma$. This result implies that the linear system may not function in certain environments. To overcome this limitation, we plan future work to identify applications where the application of such linear systems is challenging and to apply simple non-linear functions (for example, confidence-based parameter selection). This approach aims to enhance the adaptability and effectiveness of CMF in a broader range of applications.

# E DETAILS OF RESULTS

This section provides detailed experimental results from 4.1.

Table 13: Average error rate (%) and their corresponding standard deviations on **ImageNet-C** in the scenario of **CS**.

| Method | Model | gaussian | shot | impulse | defocus | glass | motion | zoom | snow | frost | fog | bright | contrast | elastic | pixelate | jpeg | Avg. |
|---|---|---|---|---|---|---|---|---|---|---|---|---|---|---|---|---|---|
| TENT | ResNet-50 | 99.1 | 98.3 | 98.8 | 89.4 | 94.4 | 89.6 | 83.6 | 85.6 | 82.8 | 85.3 | 46.0 | 97.6 | 85.8 | 76.9 | 71.7 | 85.7±0.95 |
| | ViT | 60.6 | 60.5 | 59.6 | 63.6 | 67.9 | 57.2 | 61.2 | 55.1 | 48.8 | 47.4 | 28.6 | 66.6 | 53.9 | 50.6 | 44.4 | 55.1±0.08 |
| | Swin | 65.8 | 64.1 | 68.2 | 73.4 | 75.4 | 59.1 | 64.6 | 60.1 | 57.8 | 49.2 | 28.7 | 61.7 | 72.2 | 81.8 | 56.8 | 62.6±0.18 |
| | D2V | 44.5 | 43.4 | 43.5 | 67.9 | 74.9 | 57.0 | 66.4 | 40.8 | 41.9 | 39.5 | 26.0 | 45.8 | 65.5 | 58.5 | 42.7 | 50.5±0.06 |
| CoTTA | ResNet-50 | 92.9 | 91.2 | 92.6 | 87.3 | 90.0 | 86.6 | 81.4 | 81.3 | 81.2 | 79.0 | 46.6 | 92.4 | 83.0 | 75.0 | 69.6 | 82.0±0.08 |
| | ViT | 65.2 | 66.5 | 64.7 | 67.9 | 73.3 | 63.1 | 65.7 | 56.3 | 44.6 | 48.0 | 29.0 | 83.8 | 56.4 | 59.7 | 49.3 | 59.6±0.02 |
| | Swin | 71.0 | 69.7 | 75.2 | 72.8 | 81.6 | 64.0 | 68.2 | 57.5 | 51.4 | 40.4 | 28.6 | 59.9 | 72.1 | 85.8 | 59.5 | 63.9±0.01 |
| | D2V | 43.7 | 43.0 | 43.2 | 69.6 | 77.3 | 58.8 | 68.5 | 39.9 | 43.3 | 36.3 | 26.2 | 48.7 | 67.2 | 60.0 | 43.0 | 51.2±0.02 |
| RoTTA | ResNet-50 | 90.4 | 89.7 | 90.3 | 84.8 | 89.9 | 86.9 | 80.9 | 81.3 | 78.8 | 76.9 | 38.3 | 90.6 | 80.8 | 71.1 | 61.5 | 79.5±0.10 |
| | ViT | 65.1 | 66.3 | 64.3 | 68.3 | 71.7 | 60.8 | 64.7 | 54.0 | 44.0 | 47.4 | 28.2 | 80.6 | 54.9 | 60.3 | 50.5 | 58.7±0.04 |
| | Swin | 67.9 | 66.5 | 70.9 | 73.5 | 79.4 | 62.7 | 67.6 | 55.6 | 49.3 | 42.6 | 28.8 | 59.5 | 70.7 | 87.5 | 60.3 | 62.9±0.03 |
| | D2V | 43.5 | 42.9 | 42.7 | 69.8 | 77.8 | 59.4 | 68.7 | 39.7 | 42.8 | 36.0 | 26.3 | 49.9 | 67.1 | 60.4 | 43.2 | 51.3±0.03 |
| SAR | ResNet-50 | 97.9 | 96.6 | 97.5 | 84.4 | 87.9 | 83.2 | 77.8 | 78.0 | 76.6 | 73.2 | 43.0 | 95.1 | 78.9 | 62.4 | 60.8 | 79.6±0.68 |
| | ViT | 58.8 | 57.6 | 57.4 | 59.3 | 63.7 | 53.1 | 58.4 | 52.4 | 47.3 | 45.6 | 28.4 | 61.4 | 51.5 | 47.4 | 41.9 | 52.3±0.12 |
| | Swin | 63.9 | 62.2 | 64.3 | 72.2 | 71.2 | 57.8 | 62.6 | 58.7 | 55.6 | 51.3 | 29.1 | 60.1 | 67.1 | 77.4 | 53.9 | 60.5±1.04 |
| | D2V | 44.6 | 43.7 | 43.8 | 69.0 | 76.3 | 57.1 | 66.8 | 40.9 | 41.1 | 38.8 | 25.9 | 46.2 | 65.5 | 58.6 | 42.7 | 50.7±0.07 |
| EATA | ResNet-50 | 90.7 | 88.9 | 90.8 | 76.4 | 81.4 | 74.3 | 69.1 | 71.2 | 69.4 | 63.6 | 41.6 | 93.6 | 69.9 | 52.4 | 54.8 | 72.5±1.44 |
| | ViT | 59.2 | 57.7 | 57.9 | 58.9 | 63.2 | 52.7 | 58.2 | 51.3 | 46.7 | 43.9 | 28.7 | 58.1 | 51.0 | 47.1 | 41.8 | 51.8±0.14 |
| | Swin | 61.7 | 60.3 | 61.4 | 65.6 | 68.8 | 52.7 | 58.3 | 53.4 | 50.5 | 45.4 | 26.9 | 50.9 | 62.9 | 72.0 | 51.7 | 56.2±0.29 |
| | D2V | 72.6 | 71.8 | 71.5 | 84.4 | 89.0 | 79.9 | 83.9 | 68.2 | 71.1 | 71.2 | 62.2 | 82.5 | 83.5 | 79.9 | 71.2 | 76.2±20.23 |
| ROID | ResNet-50 | 76.4 | 75.3 | 76.1 | 77.9 | 81.7 | 75.1 | 69.9 | 70.9 | 68.4 | 64.3 | 42.5 | 85.4 | 69.8 | 53.0 | 55.6 | 69.5±0.13 |
| | ViT | 58.3 | 57.2 | 57.3 | 57.4 | 61.6 | 52.1 | 58.3 | 49.7 | 44.1 | 42.1 | 27.2 | 55.8 | 50.6 | 47.0 | 41.5 | 50.7±0.08 |
| | Swin | 61.1 | 59.6 | 60.8 | 66.4 | 67.3 | 53.4 | 57.3 | 51.0 | 45.1 | 43.1 | 26.2 | 52.6 | 59.6 | 71.1 | 50.9 | 55.0±0.26 |
| | D2V | 42.9 | 42.3 | 42.0 | 64.6 | 70.3 | 54.3 | 62.3 | 38.3 | 37.1 | 34.1 | 24.5 | 42.7 | 59.7 | 55.0 | 40.3 | 47.4±0.08 |
| cmf | ResNet-50 | 72.8 | 71.7 | 72.4 | 76.0 | 78.3 | 71.9 | 67.4 | 69.5 | 67.9 | 62.7 | 45.1 | 86.2 | 66.1 | 51.8 | 55.0 | **67.6±0.20** |
| | ViT | 57.1 | 55.6 | 55.8 | 56.4 | 58.2 | 48.9 | 55.5 | 47.5 | 44.5 | 43.1 | 27.9 | 52.9 | 47.4 | 44.0 | 40.6 | **49.0±0.10** |
| | Swin | 59.5 | 57.6 | 58.9 | 64.7 | 63.9 | 50.9 | 54.7 | 46.9 | 43.3 | 42.9 | 26.7 | 48.7 | 54.0 | 60.7 | 47.8 | **52.1±0.12** |
| | D2V | 42.5 | 41.6 | 41.6 | 61.5 | 66.0 | 51.6 | 60.2 | 37.1 | 36.1 | 33.8 | 23.9 | 41.6 | 55.8 | 52.3 | 39.2 | **45.7±0.03** |

Table 14: Average error rate (%) and their corresponding standard deviations on **ImageNet-C** in the scenario of **TC-CS**. We choose $q = 0.00025$ for ResNet-50.

| Method | Model | Adaptation Order ($\rightarrow$) | | | | | | | | | | | | | | | Avg. |
|---|---|---|---|---|---|---|---|---|---|---|---|---|---|---|---|---|---|
| | | gaussian | shot | impulse | defocus | glass | motion | zoom | snow | frost | fog | bright | contrast | elastic | pixelate | jpeg | |
| TENT | ResNet-50 | 82.4 | 75.9 | 74.0 | 78.2 | 74.5 | 66.2 | 55.7 | 61.3 | 62.7 | 50.3 | 37.4 | 71.7 | 50.3 | 47.0 | 52.7 | 62.7±0.08 |
| | ViT | 63.6 | 59.9 | 58.8 | 68.2 | 68.2 | 58.0 | 61.4 | 53.9 | 45.4 | 47.9 | 28.2 | 63.1 | 53.5 | 50.7 | 42.4 | 54.5±0.04 |
| | Swin | 66.9 | 62.0 | 63.5 | 79.2 | 78.6 | 65.4 | 67.4 | 59.0 | 55.8 | 51.6 | 32.2 | 62.5 | 74.0 | 82.8 | 59.3 | 64.0±0.14 |
| | D2V | 43.9 | 43.1 | 43.3 | 69.4 | 76.4 | 57.9 | 67.1 | 40.5 | 44.7 | 51.3 | 27.3 | 47.0 | 64.5 | 58.6 | 43.2 | 51.9±0.09 |
| CoTTA | ResNet-50 | 84.8 | 83.7 | 84.3 | 84.6 | 83.8 | 72.4 | 60.3 | 64.9 | 66.9 | 50.6 | 34.6 | 80.4 | 54.2 | 49.3 | 57.4 | 67.5±0.08 |
| | ViT | 65.8 | 67.1 | 65.2 | 68.6 | 74.0 | 63.9 | 66.2 | 56.6 | 44.6 | 46.0 | 28.9 | 92.9 | 56.4 | 60.3 | 49.0 | 60.4±0.02 |
| | Swin | 71.1 | 69.9 | 75.4 | 72.8 | 81.6 | 63.9 | 68.2 | 57.9 | 51.8 | 43.1 | 28.8 | 61.7 | 71.6 | 86.1 | 59.0 | 64.2±0.01 |
| | D2V | 43.9 | 43.3 | 43.4 | 69.7 | 78.0 | 59.3 | 68.9 | 40.1 | 44.2 | 38.2 | 26.2 | 49.3 | 67.1 | 60.7 | 43.0 | 51.7±0.02 |
| RoTTA | ResNet-50 | 88.4 | 83.2 | 83.3 | 92.2 | 85.7 | 74.7 | 60.6 | 66.9 | 66.0 | 53.9 | 35.5 | 77.2 | 56.0 | 50.5 | 55.4 | 68.6±0.16 |
| | ViT | 65.8 | 67.1 | 64.9 | 68.9 | 73.3 | 62.8 | 65.2 | 55.6 | 44.1 | 45.7 | 27.9 | 80.3 | 54.5 | 60.0 | 49.8 | 591±0.05 |
| | Swin | 71.0 | 69.3 | 73.8 | 73.2 | 80.4 | 62.7 | 67.2 | 56.9 | 48.7 | 42.9 | 29.1 | 59.0 | 69.2 | 88.7 | 59.0 | 63.4±0.01 |
| | D2V | 43.9 | 43.3 | 43.3 | 69.7 | 77.8 | 59.4 | 68.7 | 39.8 | 42.5 | 35.8 | 26.2 | 49.7 | 66.6 | 60.1 | 43.3 | 51.3±0.01 |
| SAR | ResNet-50 | 82.6 | 75.8 | 72.9 | 77.7 | 74.2 | 65.5 | 55.7 | 61.4 | 62.4 | 50.3 | 36.7 | 69.3 | 49.4 | 45.8 | 51.4 | 62.1±0.18 |
| | ViT | 61.3 | 55.7 | 54.4 | 62.0 | 61.4 | 53.8 | 57.0 | 53.9 | 45.1 | 45.9 | 29.1 | 55.1 | 51.5 | 49.2 | 40.3 | 51.7±0.14 |
| | Swin | 63.5 | 57.4 | 58.0 | 77.1 | 73.8 | 68.0 | 71.7 | 65.5 | 67.8 | 63.3 | 32.0 | 70.2 | 71.8 | 84.8 | 63.0 | 65.9±1.27 |
| | D2V | 44.2 | 43.8 | 43.7 | 69.7 | 77.5 | 57.1 | 66.8 | 41.2 | 41.4 | 41.9 | 26.3 | 48.2 | 64.3 | 57.1 | 41.9 | 51.0±0.12 |
| EATA | ResNet-50 | 77.1 | 67.2 | 65.6 | 73.6 | 69.1 | 62.1 | 53.6 | 58.8 | 59.6 | 48.2 | 35.8 | 63.5 | 47.6 | 44.0 | 47.7 | 58.2±0.15 |
| | ViT | 61.6 | 55.3 | 53.8 | 60.0 | 58.8 | 52.7 | 54.9 | 51.3 | 43.5 | 42.9 | 29.1 | 49.2 | 48.7 | 46.4 | 39.8 | 49.9±0.06 |
| | Swin | 63.2 | 55.6 | 54.9 | 67.6 | 64.3 | 54.2 | 54.3 | 51.9 | 47.2 | 43.5 | 26.3 | 54.6 | 61.5 | | 47.3 | 52.9±0.25 |
| | D2V | 47.2 | 48.1 | 59.2 | 78.0 | 84.6 | 71.6 | 77.3 | 55.2 | 59.0 | 57.9 | 46.9 | 73.9 | 76.7 | 71.6 | 59.2 | 64.4±15.84 |
| ROID | ResNet-50 | 73.0 | 62.8 | 62.7 | 69.6 | 66.8 | 57.2 | 49.1 | 52.2 | 57.7 | 43.5 | 33.3 | 59.3 | 45.3 | 41.7 | 45.9 | **54.7±0.04** |
| | ViT | 57.6 | 51.5 | 52.2 | 55.1 | 52.4 | 46.5 | 47.2 | 45.6 | 39.5 | 36.0 | 26.0 | 45.0 | 43.8 | 39.7 | 36.3 | 45.0±0.09 |
| | Swin | 58.0 | 51.6 | 51.4 | 62.9 | 57.6 | 49.9 | 47.5 | 44.2 | 39.9 | 36.2 | 24.2 | 43.9 | 44.5 | 50.4 | 42.5 | 47.0±0.26 |
| | D2V | 42.8 | 40.5 | 40.1 | 64.0 | 64.6 | 50.6 | 57.6 | 37.0 | 36.6 | 31.7 | 24.5 | 39.7 | 57.3 | 47.8 | 37.1 | 44.8±0.01 |
| CMF | ResNet-50 | 73.0 | 62.6 | 61.8 | 69.9 | 66.1 | 57.8 | 50.0 | 53.5 | 57.6 | 44.6 | 34.4 | 58.2 | 45.7 | 42.0 | 45.6 | **54.9±0.16** |
| | ViT | 57.6 | 50.3 | 50.3 | | 51.0 | 46.7 | 46.9 | 46.1 | 40.6 | 37.9 | 26.2 | 44.3 | 41.3 | 39.6 | 36.1 | **44.8±0.12** |
| | Swin | 58.4 | 50.8 | 49.9 | 64.7 | 58.1 | 50.7 | 47.1 | 44.3 | 39.4 | 36.7 | 24.6 | 42.4 | 43.2 | 47.9 | 40.3 | **46.6±0.12** |
| | D2V | 42.9 | 40.3 | 39.8 | 64.0 | 63.4 | 49.5 | 55.1 | 36.7 | 34.9 | 32.4 | 23.4 | 38.4 | 52.0 | 44.9 | 35.6 | **43.5±0.04** |

Table 15: Average error rate (%) and their corresponding standard deviations on **ImageNet-C** in the scenario of **TC-LS** ($\delta = 0.0$) **over TC-CS**. We choose $q = 0.00025$.

| Method | Model | Adaptation Order ($\rightarrow$) | | | | | | | | | | | | | | | Avg. |
|---|---|---|---|---|---|---|---|---|---|---|---|---|---|---|---|---|---|
| | | gaussian | shot | impulse | defocus | glass | motion | zoom | snow | frost | fog | bright | contrast | elastic | pixelate | jpeg | |
| TENT | ViT | 58.7 | 53.9 | 54.3 | 58.4 | 58.7 | 52.7 | 74.0 | 99.4 | 99.8 | 99.9 | 99.9 | 99.9 | 99.9 | 99.9 | 99.9 | 80.6±0.07 |
| | Swin | 61.4 | 57.5 | 59.8 | 76.4 | 75.5 | 74.9 | 88.0 | 91.0 | 98.7 | 99.8 | 98.8 | 99.8 | 99.8 | 99.8 | 99.8 | 85.4±1.26 |
| | D2V | 43.6 | 41.4 | 40.6 | 65.4 | 70.6 | 54.5 | 62.6 | 38.8 | 42.8 | 39.3 | 25.1 | 42.9 | 58.6 | 51.9 | 39.4 | 47.8±0.06 |
| CoTTA | ViT | 65.8 | 65.5 | 64.1 | 66.6 | 71.4 | 62.6 | 66.9 | 58.6 | 45.0 | 42.0 | 30.5 | 90.5 | 56.9 | 60.5 | 49.8 | 59.8±0.16 |
| | Swin | 70.3 | 69.3 | 74.4 | 72.8 | 81.0 | 65.6 | 70.2 | 59.5 | 51.2 | 52.9 | 30.7 | 62.3 | 73.0 | 89.6 | 59.8 | 65.5±0.07 |
| | D2V | 43.8 | 42.3 | 42.0 | 69.5 | 77.2 | 57.4 | 66.8 | 40.5 | 41.5 | 44.0 | 25.2 | 45.6 | 65.6 | 59.5 | 41.4 | 50.8±0.04 |
| RoTTA | ViT | 66.6 | 68.1 | 71.0 | 69.6 | 71.0 | 58.7 | 64.8 | 59.3 | 52.2 | 50.8 | 33.2 | 83.9 | 60.0 | 84.9 | 76.5 | 64.7±0.16 |
| | Swin | 69.8 | 67.9 | 72.0 | 77.6 | 77.9 | 70.2 | 72.9 | 59.6 | 55.8 | 66.9 | 42.1 | 83.3 | 75.6 | 97.4 | 88.5 | 71.8±0.15 |
| | D2V | 43.8 | 42.0 | 42.0 | 69.8 | 74.5 | 59.3 | 67.4 | 40.2 | 39.5 | 40.1 | 29.0 | 74.1 | 72.3 | 72.8 | 51.4 | 54.5±0.03 |
| SAR | ViT | 55.9 | 51.9 | 56.2 | 56.4 | 56.4 | 56.0 | 56.7 | 61.8 | 43.0 | 35.8 | 26.4 | 43.9 | 50.6 | 37.7 | 36.3 | 48.3±0.28 |
| | Swin | 62.3 | 61.1 | 62.1 | 78.9 | 78.5 | 66.9 | 67.6 | 66.9 | 60.1 | 49.7 | 27.4 | 46.9 | 68.3 | 57.6 | 47.3 | 60.1±0.74 |
| | D2V | 44.2 | 41.8 | 41.0 | 67.8 | 72.0 | 54.8 | 63.6 | 39.2 | 39.1 | 38.3 | 25.6 | 43.7 | 63.6 | 51.2 | 38.0 | 48.3±0.15 |
| EATA | ViT | 59.5 | 64.0 | 70.7 | 77.1 | 73.7 | 75.2 | 72.7 | 71.8 | 67.6 | 73.5 | 56.6 | 98.9 | 77.4 | 71.9 | 66.6 | 71.8±1.22 |
| | Swin | 66.4 | 72.8 | 76.6 | 81.8 | 79.3 | 75.9 | 73.5 | 72.3 | 68.2 | 69.3 | 55.9 | 81.4 | 68.8 | 78.3 | 69.8 | 72.7±0.67 |
| | D2V | 52.0 | 49.2 | 49.6 | 72.1 | 81.3 | 64.7 | 71.6 | 42.5 | 50.4 | 51.2 | 32.8 | 73.8 | 70.8 | 63.6 | 47.6 | 58.2±2.21 |
| LAME | ViT | 40.1 | 39.0 | 39.2 | 48.6 | 58.6 | 43.2 | 48.4 | 39.6 | 34.1 | 42.2 | 23.8 | 84.7 | 44.5 | 40.2 | 35.8 | 44.1±0.02 |
| | Swin | 43.1 | 42.1 | 46.2 | 52.8 | 69.8 | 43.4 | 51.2 | 44.6 | 36.5 | 33.9 | 20.4 | 41.3 | 64.5 | 72.7 | 43.9 | 47.1±0.09 |
| | D2V | 30.5 | 29.8 | 30.2 | 49.4 | 62.4 | 39.9 | 50.3 | 31.3 | 34.3 | 31.4 | 22.7 | 39.9 | 55.9 | 41.5 | 34.5 | 38.9±0.07 |
| ROID | ViT | 25.7 | 22.9 | 23.8 | 24.0 | 22.0 | 15.8 | 18.2 | 14.9 | 12.3 | 10.3 | 6.4 | 14.9 | 12.2 | 9.9 | 10.1 | 16.2±0.06 |
| | Swin | 25.8 | 22.2 | 22.0 | 33.0 | 29.7 | 18.0 | 20.5 | 13.9 | 12.2 | 10.9 | 6.8 | 14.7 | 13.7 | 14.3 | 14.3 | 18.1±0.03 |
| | D2V | 12.2 | 11.8 | 11.5 | 33.6 | 35.7 | 18.3 | 30.2 | 12.6 | 11.6 | 9.6 | 7.3 | 11.8 | 26.4 | 15.8 | 12.9 | 17.4±0.21 |
| CMF | ViT | 25.1 | 21.9 | 23.3 | 24.6 | 21.2 | 15.5 | 16.0 | 15.0 | 12.8 | 10.3 | 6.5 | 14.6 | 11.4 | 9.8 | 10.0 | **15.9±0.04** |
| | Swin | 24.9 | 20.6 | 20.8 | 33.2 | 27.7 | 17.1 | 17.1 | 12.1 | 11.4 | 10.0 | 6.4 | 13.6 | 11.8 | 12.4 | 12.0 | **16.7±0.10** |
| | D2V | 12.4 | 12.0 | 11.9 | 28.8 | 23.9 | 15.6 | 22.4 | 11.2 | 10.2 | 8.9 | 6.3 | 11.3 | 17.9 | 13.0 | 9.7 | **14.4±0.24** |

Table 16: Average error rate (%) and their corresponding standard deviations on **ImageNet-C** in the scenario of **TC-LS** ($\delta = 0.01$) **over TC-CS**. We choose $q = 0.00025$.

| Method | Model | Adaptation Order ($\rightarrow$) | | | | | | | | | | | | | | | Avg. |
|---|---|---|---|---|---|---|---|---|---|---|---|---|---|---|---|---|---|
| | | gaussian | shot | impulse | defocus | glass | motion | zoom | snow | frost | fog | bright | contrast | elastic | pixelate | jpeg | |
| TENT | ViT | 58.2 | 53.7 | 54.4 | 59.0 | 59.0 | 52.4 | 66.6 | 98.4 | 99.8 | 99.9 | 99.9 | 99.9 | 99.9 | 99.9 | 99.9 | 80.1±0.11 |
| | Swin | 60.7 | 57.3 | 58.8 | 76.5 | 77.2 | 71.3 | 83.6 | 93.6 | 99.6 | 99.8 | 99.0 | 99.9 | 99.8 | 99.9 | 99.8 | 85.1±1.25 |
| | D2V | 43.3 | 41.4 | 40.6 | 65.8 | 70.3 | 54.5 | 62.7 | 38.8 | 41.7 | 39.7 | 25.0 | 42.7 | 58.3 | 52.3 | 39.3 | 47.8±0.13 |
| CoTTA | ViT | 65.6 | 65.2 | 63.6 | 66.7 | 71.4 | 62.8 | 67.0 | 58.7 | 45.2 | 41.7 | 30.8 | 91.2 | 56.5 | 61.8 | 50.8 | 59.9±0.17 |
| | Swin | 70.3 | 69.3 | 74.5 | 72.8 | 81.1 | 65.6 | 70.0 | 59.4 | 51.1 | 53.6 | 30.5 | 61.2 | 73.1 | 90.6 | 60.2 | 65.5±0.43 |
| | D2V | 43.7 | 42.2 | 42.0 | 69.4 | 77.0 | 57.4 | 66.7 | 40.6 | 41.3 | 45.0 | 25.2 | 46.9 | 65.6 | 59.9 | 41.6 | 51.0±0.06 |
| RoTTA | ViT | 65.3 | 64.7 | 65.6 | 67.7 | 68.5 | 56.0 | 61.5 | 55.2 | 48.6 | 46.7 | 30.6 | 74.5 | 54.7 | 76.4 | 66.3 | 60.2±0.17 |
| | Swin | 68.3 | 64.8 | 66.6 | 77.7 | 75.7 | 67.7 | 70.1 | 56.3 | 52.3 | 61.9 | 38.2 | 77.8 | 71.3 | 96.2 | 84.3 | 68.6±0.15 |
| | D2V | 43.5 | 41.3 | 41.0 | 68.7 | 71.6 | 56.7 | 65.0 | 39.3 | 38.4 | 38.7 | 28.3 | 66.7 | 67.9 | 68.0 | 49.2 | 52.3±0.04 |
| SAR | ViT | 55.0 | 50.9 | 53.6 | 57.7 | 64.6 | 53.7 | 54.4 | 60.7 | 40.9 | 35.5 | 26.8 | 44.3 | 54.9 | 40.9 | 36.0 | 48.7±0.29 |
| | Swin | 57.0 | 56.3 | 57.0 | 78.2 | 75.8 | 64.3 | 63.1 | 69.5 | 59.4 | 51.7 | 26.3 | 44.6 | 72.1 | 54.5 | 46.0 | 58.4±0.86 |
| | D2V | 43.9 | 41.7 | 40.9 | 68.2 | 71.6 | 54.9 | 63.5 | 39.3 | 39.1 | 38.5 | 25.2 | 44.5 | 58.2 | 50.0 | 39.3 | 47.9±0.05 |
| EATA | ViT | 54.6 | 50.4 | 52.5 | 57.2 | 53.6 | 49.9 | 49.8 | 50.3 | 45.8 | 43.4 | 30.3 | 51.1 | 44.5 | 41.0 | 40.7 | 47.7±0.12 |
| | Swin | 54.1 | 50.5 | 54.6 | 65.2 | 61.3 | 53.2 | 49.6 | 48.9 | 46.1 | 42.2 | 27.8 | 50.6 | 47.6 | 52.9 | 45.7 | 50.0±0.35 |
| | D2V | 52.2 | 59.0 | 59.3 | 77.2 | 84.5 | 70.3 | 76.7 | 55.3 | 59.3 | 57.0 | 45.7 | 73.9 | 76.5 | 71.0 | 57.7 | 65.0±18.58 |
| LAME | ViT | 88.0 | 80.9 | 90.0 | 81.3 | 89.9 | 91.4 | 90.3 | 89.4 | 71.1 | 98.4 | 51.4 | 99.6 | 82.5 | 76.1 | 68.2 | 83.2±0.23 |
| | Swin | 84.6 | 80.0 | 86.8 | 86.8 | 95.4 | 84.1 | 84.3 | 96.2 | 92.7 | 84.6 | 45.6 | 84.1 | 97.4 | 93.6 | 73.8 | 84.7±0.12 |
| | D2V | 70.0 | 68.2 | 68.4 | 81.2 | 88.9 | 76.8 | 88.0 | 79.7 | 86.3 | 89.5 | 64.9 | 89.6 | 91.3 | 78.1 | 71.7 | 79.5±0.20 |
| ROID | ViT | 49.2 | 46.0 | 47.3 | 46.3 | 43.7 | 37.2 | 38.0 | 34.6 | 32.2 | 27.3 | 19.3 | 35.9 | 30.2 | 28.2 | 28.8 | 36.3±0.08 |
| | Swin | 48.5 | 44.7 | 45.2 | 52.5 | 47.0 | 39.3 | 39.4 | 32.6 | 29.8 | 26.5 | 18.5 | 34.8 | 31.6 | 34.6 | 32.9 | 37.2±0.06 |
| | D2V | 34.8 | 33.2 | 33.1 | 52.2 | 52.1 | 41.4 | 50.1 | 30.0 | 29.4 | 24.4 | 18.1 | 29.6 | 44.7 | 36.1 | 29.2 | 35.9±0.08 |
| CMF | ViT | 47.4 | 43.3 | 44.6 | 45.6 | 41.9 | 35.6 | 34.5 | 33.8 | 32.7 | 26.8 | 19.8 | 34.9 | 28.5 | 27.2 | 28.1 | **35.0±0.04** |
| | Swin | 46.6 | 41.7 | 42.4 | 52.1 | 45.4 | 37.4 | 35.1 | 30.0 | 29.3 | 25.2 | 18.2 | 32.8 | 29.2 | 31.5 | 29.4 | **35.1±0.16** |
| | D2V | 34.5 | 32.6 | 32.6 | 48.5 | 45.1 | 36.6 | 41.5 | 28.2 | 27.1 | 23.2 | 16.9 | 28.7 | 37.2 | 32.6 | 25.5 | **32.7±0.04** |

Table 17: Average error rate (%) and their corresponding standard deviations on **ImageNet-C** in the scenario of **TC-LS** ($\delta = 0.1$) **over TC-CS**.

| Method | Model | Adaptation Order ($\rightarrow$) | | | | | | | | | | | | | | | Avg. |
|---|---|---|---|---|---|---|---|---|---|---|---|---|---|---|---|---|---|
| | | gaussian | shot | impulse | defocus | glass | motion | zoom | snow | frost | fog | bright | contrast | elastic | pixelate | jpeg | |
| TENT | ViT | 58.2 | 53.7 | 54.6 | 59.1 | 58.9 | 52.4 | 66.6 | 98.5 | 99.8 | 99.9 | 99.9 | 99.9 | 99.9 | 99.9 | 99.9 | 80.1±0.14 |
| | Swin | 60.8 | 57.0 | 58.6 | 76.4 | 76.4 | 68.9 | 76.5 | 89.1 | 99.3 | 99.7 | 98.8 | 99.9 | 99.8 | 99.9 | 99.8 | 84.1±0.72 |
| | D2V | 43.3 | 41.4 | 40.5 | 65.6 | 70.3 | 54.5 | 62.8 | 38.7 | 41.3 | 39.2 | 25.0 | 42.8 | 58.2 | 52.0 | 39.4 | 47.7±0.03 |
| CoTTA | ViT | 65.6 | 65.2 | 63.6 | 66.6 | 71.4 | 62.6 | 66.9 | 58.3 | 45.2 | 41.5 | 30.8 | 91.4 | 56.3 | 62.1 | 51.0 | 59.9±0.09 |
| | Swin | 70.3 | 69.3 | 74.6 | 72.8 | 81.1 | 65.5 | 69.9 | 59.4 | 51.2 | 54.8 | 30.7 | 62.7 | 73.3 | 89.7 | 60.6 | 65.7±0.25 |
| | D2V | 43.7 | 42.2 | 42.0 | 69.4 | 77.0 | 57.4 | 66.7 | 40.6 | 41.4 | 44.1 | 25.3 | 47.2 | 65.6 | 59.8 | 41.6 | 50.9±0.11 |
| RoTTA | ViT | 65.1 | 64.0 | 64.4 | 67.5 | 68.1 | 55.2 | 60.6 | 54.7 | 47.9 | 45.4 | 30.1 | 72.9 | 54.0 | 74.5 | 64.1 | 59.2±0.07 |
| | Swin | 68.0 | 64.0 | 65.2 | 77.9 | 75.0 | 66.9 | 69.5 | 56.1 | 51.7 | 60.9 | 37.4 | 76.4 | 70.8 | 96.0 | 83.6 | 68.0±0.02 |
| | D2V | 43.5 | 41.2 | 40.9 | 68.4 | 71.1 | 56.4 | 64.5 | 39.1 | 38.3 | 38.5 | 28.3 | 65.5 | 67.5 | 67.2 | 49.1 | 52.0±0.01 |
| SAR | ViT | 54.9 | 50.5 | 53.4 | 57.6 | 60.8 | 52.4 | 58.8 | 59.7 | 43.0 | 35.5 | 26.7 | 44.2 | 53.0 | 40.1 | 36.0 | 48.4±0.30 |
| | Swin | 56.7 | 55.8 | 56.5 | 77.4 | 76.4 | 66.7 | 64.7 | 67.2 | 59.0 | 47.5 | 26.3 | 45.1 | 75.1 | 54.6 | 47.5 | 58.4±0.75 |
| | D2V | 43.9 | 41.7 | 40.9 | 68.7 | 71.7 | 54.8 | 63.3 | 39.3 | 39.4 | 38.8 | 25.3 | 44.7 | 58.6 | 49.8 | 39.2 | 48.0±0.04 |
| EATA | ViT | 54.3 | 49.9 | 51.5 | 56.2 | 52.2 | 48.5 | 47.8 | 48.5 | 43.6 | 40.1 | 29.0 | 49.0 | 41.8 | 39.6 | 39.2 | 46.1±0.17 |
| | Swin | 53.8 | 49.4 | 52.6 | 64.0 | 58.9 | 51.9 | 48.0 | 47.1 | 43.6 | 39.7 | 27.3 | 48.7 | 45.2 | 50.2 | 43.5 | 48.3±0.09 |
| | D2V | 57.6 | 59.0 | 59.4 | 77.4 | 84.6 | 70.6 | 76.9 | 55.3 | 59.3 | 57.0 | 46.0 | 73.4 | 76.5 | 71.1 | 57.9 | 65.5±19.11 |
| LAME | ViT | 93.4 | 79.4 | 97.2 | 72.4 | 90.9 | 96.8 | 93.8 | 96.5 | 49.2 | 99.9 | 28.9 | 99.9 | 84.4 | 66.0 | 50.4 | 79.9±0.06 |
| | Swin | 86.5 | 76.6 | 87.8 | 84.8 | 97.6 | 84.4 | 81.9 | 99.8 | 99.3 | 96.1 | 29.4 | 87.8 | 99.8 | 95.3 | 60.8 | 84.5±0.09 |
| | D2V | 45.7 | 43.7 | 45.2 | 72.4 | 88.0 | 60.2 | 87.5 | 89.4 | 95.0 | 99.7 | 27.1 | 95.9 | 95.0 | 63.2 | 44.2 | 70.1±0.04 |
| ROID | ViT | 53.1 | 50.3 | 51.5 | 50.7 | 48.3 | 42.4 | 43.0 | 39.9 | 37.7 | 32.8 | 24.9 | 41.0 | 35.5 | 33.9 | 34.5 | 41.3±0.05 |
| | Swin | 52.6 | 49.5 | 50.0 | 55.9 | 50.6 | 44.3 | 43.9 | 37.7 | 35.1 | 31.9 | 23.8 | 39.6 | 37.0 | 40.3 | 38.7 | 42.1±0.04 |
| | D2V | 40.8 | 39.3 | 39.4 | 55.8 | 56.1 | 46.9 | 54.3 | 35.8 | 35.1 | 30.1 | 23.6 | 35.7 | 49.1 | 42.2 | 35.1 | 41.3±0.03 |
| CMF | ViT | 51.4 | 47.4 | 48.7 | 49.5 | 45.8 | 40.3 | 39.2 | 38.6 | 37.5 | 31.9 | 24.9 | 39.5 | 33.5 | 32.4 | 33.4 | **39.6±0.03** |
| | Swin | 50.5 | 46.3 | 46.9 | 54.7 | 48.8 | 41.9 | 39.3 | 34.9 | 34.6 | 30.1 | 23.1 | 37.8 | 34.0 | 36.5 | 34.8 | **39.6±0.02** |
| | D2V | 40.3 | 38.5 | 38.4 | 52.7 | 49.9 | 42.5 | 46.4 | 34.0 | 32.8 | 28.6 | 22.2 | 34.4 | 42.1 | 38.6 | 31.6 | **38.2±0.05** |

Table 18: Average error rate (%) and their corresponding standard deviations on **ImageNet-C** in the scenario of **TC-LS** ($\delta = 1.0$) **over TC-CS**.

| Method | Model | Adaptation Order ($\rightarrow$) | | | | | | | | | | | | | | | Avg. |
|---|---|---|---|---|---|---|---|---|---|---|---|---|---|---|---|---|---|
| | | gaussian | shot | impulse | defocus | glass | motion | zoom | snow | frost | fog | bright | contrast | elastic | pixelate | jpeg | |
| TENT | ViT | 58.2 | 53.7 | 55.8 | 59.1 | 67.4 | 64.2 | 73.7 | 95.7 | 99.8 | 99.9 | 99.8 | 99.9 | 99.9 | 99.9 | 99.9 | 81.8±3.68 |
| | Swin | 60.8 | 57.4 | 60.3 | 77.3 | 79.9 | 84.8 | 91.6 | 93.7 | 99.2 | 99.8 | 98.8 | 99.8 | 99.8 | 99.9 | 99.7 | 86.9±2.27 |
| | D2V | 43.3 | 41.4 | 40.9 | 65.6 | 70.4 | 54.6 | 62.8 | 38.8 | 41.4 | 41.1 | 39.5 | 42.6 | 58.1 | 52.2 | 39.3 | 47.7±0.27 |
| CoTTA | ViT | 65.6 | 65.2 | 63.5 | 66.6 | 71.5 | 62.6 | 66.9 | 58.4 | 45.2 | 41.6 | 30.8 | 91.3 | 56.9 | 62.1 | 50.9 | 59.9±0.17 |
| | Swin | 70.3 | 69.3 | 74.6 | 72.8 | 81.2 | 65.5 | 70.0 | 59.4 | 51.0 | 55.0 | 30.8 | 63.3 | 73.4 | 90.1 | 60.8 | 65.8±0.24 |
| | D2V | 43.7 | 42.2 | 42.0 | 69.4 | 77.0 | 57.4 | 66.7 | 40.6 | 41.4 | 44.1 | 25.3 | 46.9 | 65.6 | 59.8 | 41.6 | 50.9±0.08 |
| RoTTA | ViT | 65.0 | 63.9 | 64.1 | 67.2 | 67.9 | 55.1 | 60.7 | 54.7 | 47.8 | 45.4 | 30.0 | 72.9 | 53.9 | 74.7 | 64.0 | 59.2±0.11 |
| | Swin | 68.1 | 64.2 | 65.4 | 77.8 | 75.1 | 67.1 | 69.6 | 55.9 | 51.7 | 60.6 | 37.4 | 76.1 | 70.6 | 96.0 | 83.8 | 67.9±0.10 |
| | D2V | 43.5 | 41.2 | 40.8 | 68.3 | 71.1 | 56.4 | 64.5 | 39.1 | 38.3 | 38.5 | 28.3 | 65.5 | 67.2 | 67.2 | 49.1 | 51.9±0.04 |
| SAR | ViT | 54.8 | 50.6 | 53.0 | 58.9 | 61.9 | 53.7 | 54.3 | 61.0 | 39.8 | 35.6 | 26.7 | 44.2 | 51.4 | 42.8 | 35.9 | 48.3±0.25 |
| | Swin | 56.5 | 56.5 | 57.5 | 78.9 | 75.7 | 63.1 | 69.3 | 69.1 | 59.9 | 45.0 | 26.5 | 46.4 | 71.8 | 55.0 | 47.0 | 58.5±0.41 |
| | D2V | 43.9 | 41.7 | 40.9 | 68.3 | 71.7 | 54.8 | 63.4 | 39.3 | 39.3 | 38.7 | 25.2 | 44.6 | 58.1 | 49.6 | 39.1 | 47.9±0.09 |
| EATA | ViT | 54.2 | 49.7 | 51.1 | 55.6 | 51.9 | 48.1 | 47.3 | 48.1 | 43.3 | 40.3 | 28.7 | 48.1 | 41.5 | 39.1 | 38.7 | 45.7±0.15 |
| | Swin | 53.3 | 49.0 | 52.1 | 63.0 | 58.0 | 51.0 | 47.1 | 46.7 | 42.3 | 38.3 | 26.7 | 48.0 | 43.7 | 48.8 | 43.5 | 47.4±0.39 |
| | D2V | 78.0 | 84.6 | 84.8 | 91.8 | 94.6 | 89.7 | 91.4 | 83.0 | 83.6 | 84.4 | 80.7 | 91.3 | 91.7 | 89.6 | 85.1 | 87.0±18.44 |
| LAME | ViT | 93.8 | 79.1 | 97.1 | 72.6 | 91.1 | 96.8 | 94.0 | 96.1 | 49.2 | 99.9 | 29.2 | 99.9 | 84.7 | 65.9 | 50.7 | 80.0±0.03 |
| | Swin | 86.1 | 76.7 | 87.8 | 84.9 | 97.9 | 85.1 | 82.2 | 99.8 | 99.3 | 96.0 | 29.9 | 87.7 | 99.8 | 95.4 | 60.9 | 84.6±0.06 |
| | D2V | 45.9 | 43.8 | 45.3 | 72.2 | 88.2 | 60.2 | 87.4 | 90.2 | 95.2 | 99.7 | 26.9 | 96.0 | 95.5 | 62.8 | 44.0 | 70.2±0.07 |
| ROID | ViT | 53.2 | 50.3 | 51.4 | 50.6 | 48.2 | 42.3 | 42.8 | 39.7 | 37.7 | 32.8 | 25.0 | 41.0 | 35.4 | 33.8 | 34.4 | 41.2±0.03 |
| | Swin | 52.6 | 49.5 | 50.0 | 55.8 | 50.1 | 44.2 | 43.9 | 37.6 | 34.7 | 31.7 | 23.8 | 39.3 | 36.8 | 40.0 | 38.4 | 41.9±0.03 |
| | D2V | 40.8 | 39.4 | 39.4 | 55.8 | 55.7 | 46.8 | 54.2 | 35.8 | 34.9 | 30.1 | 23.5 | 35.5 | 48.7 | 41.9 | 35.0 | 41.2±0.01 |
| CMF | ViT | 51.3 | 47.2 | 48.3 | 49.3 | 45.5 | 40.1 | 38.8 | 38.6 | 37.6 | 31.7 | 24.9 | 39.4 | 33.3 | 32.1 | 33.3 | **39.4±0.03** |
| | Swin | 50.3 | 45.9 | 46.4 | 54.9 | 48.8 | 41.7 | 39.3 | 34.8 | 34.1 | 29.7 | 22.9 | 37.3 | 33.5 | 36.5 | 34.5 | **39.4±0.11** |
| | D2V | 40.3 | 38.4 | 38.3 | 52.9 | 50.2 | 42.3 | 46.2 | 33.9 | 32.7 | 28.5 | 22.2 | 34.5 | 41.8 | 38.2 | 31.4 | **38.1±0.03** |

Table 19: Average error rate (%) and their corresponding standard deviations on **ImageNet-C** in the scenario of **TC-LS** ($\delta = 5.0$) **over TC-CS**.

| Method | Model | Adaptation Order ($\rightarrow$) | | | | | | | | | | | | | | | Avg. |
|---|---|---|---|---|---|---|---|---|---|---|---|---|---|---|---|---|---|
| | | gaussian | shot | impulse | defocus | glass | motion | zoom | snow | frost | fog | bright | contrast | elastic | pixelate | jpeg | |
| TENT | ViT | 58.2 | 53.7 | 54.4 | 59.1 | 58.9 | 52.2 | 68.2 | 98.2 | 99.8 | 99.9 | 99.9 | 99.9 | 99.9 | 99.9 | 99.9 | 80.2±0.31 |
| | Swin | 61.0 | 57.7 | 59.2 | 76.7 | 77.0 | 74.0 | 83.6 | 91.9 | 99.1 | 99.8 | 99.1 | 99.9 | 99.8 | 99.9 | 99.8 | 85.2±1.19 |
| | D2V | 43.4 | 41.3 | 40.5 | 65.6 | 70.4 | 54.5 | 62.8 | 38.7 | 41.6 | 39.6 | 25.0 | 42.7 | 58.3 | 52.3 | 39.5 | 47.8±0.14 |
| CoTTA | ViT | 65.6 | 65.2 | 63.6 | 66.5 | 71.4 | 62.5 | 66.8 | 58.5 | 45.3 | 41.4 | 30.8 | 91.6 | 56.7 | 61.9 | 50.5 | 59.9±0.17 |
| | Swin | 70.3 | 69.3 | 74.6 | 72.7 | 81.1 | 65.5 | 70.0 | 59.4 | 51.3 | 55.7 | 30.7 | 62.2 | 73.4 | 90.0 | 60.9 | 65.8±0.17 |
| | D2V | 43.8 | 42.2 | 42.0 | 69.4 | 77.0 | 57.4 | 66.7 | 40.6 | 41.4 | 44.1 | 25.3 | 46.8 | 65.7 | 59.8 | 41.6 | 50.9±0.10 |
| RoTTA | ViT | 65.1 | 63.9 | 64.2 | 67.2 | 67.9 | 55.2 | 60.7 | 54.5 | 47.5 | 45.2 | 30.0 | 72.7 | 53.8 | 73.7 | 63.4 | 59.0±0.04 |
| | Swin | 68.1 | 64.1 | 65.1 | 77.9 | 75.2 | 67.1 | 69.6 | 56.0 | 51.8 | 61.2 | 37.5 | 76.1 | 70.7 | 96.0 | 83.5 | 68.0±0.16 |
| | D2V | 43.5 | 41.2 | 40.8 | 68.3 | 71.1 | 56.3 | 64.5 | 39.1 | 38.2 | 38.5 | 28.3 | 65.1 | 67.3 | 67.1 | 49.0 | 51.9±0.05 |
| SAR | ViT | 55.0 | 50.7 | 54.9 | 56.6 | 60.1 | 48.1 | 61.9 | 63.1 | 57.1 | 51.5 | 45.0 | 57.9 | 64.2 | 54.1 | 52.0 | 55.5±12.62 |
| | Swin | 56.6 | 55.9 | 56.9 | 80.7 | 73.7 | 65.3 | 69.0 | 70.9 | 60.6 | 54.2 | 25.8 | 44.7 | 73.6 | 55.3 | 45.2 | 59.2±0.68 |
| | D2V | 44.0 | 41.7 | 40.9 | 68.4 | 71.7 | 54.8 | 63.5 | 39.2 | 39.1 | 38.6 | 25.2 | 44.3 | 58.1 | 50.1 | 39.3 | 47.9±0.08 |
| EATA | ViT | 54.2 | 49.7 | 51.1 | 55.6 | 51.5 | 47.9 | 47.2 | 47.8 | 43.3 | 40.0 | 28.7 | 47.5 | 41.8 | 39.3 | 38.8 | 45.6±0.17 |
| | Swin | 53.5 | 49.5 | 52.0 | 63.7 | 58.2 | 51.5 | 46.9 | 45.8 | 42.5 | 38.6 | 26.6 | 48.0 | 44.4 | 49.3 | 43.5 | 47.6±0.25 |
| | D2V | 58.9 | 59.3 | 59.7 | 77.4 | 84.8 | 70.8 | 77.1 | 55.4 | 59.8 | 58.1 | 46.3 | 75.3 | 76.7 | 71.2 | 58.1 | 65.9±18.92 |
| LAME | ViT | 93.4 | 79.6 | 97.2 | 73.6 | 91.1 | 96.5 | 93.9 | 96.6 | 49.3 | 99.9 | 29.3 | 99.9 | 85.1 | 66.2 | 51.2 | 80.2±0.09 |
| | Swin | 86.3 | 77.3 | 88.0 | 85.5 | 97.8 | 85.0 | 82.2 | 99.6 | 99.5 | 96.4 | 30.1 | 88.1 | 99.8 | 95.6 | 61.8 | 84.9±0.04 |
| | D2V | 45.7 | 44.0 | 45.7 | 73.3 | 88.7 | 60.4 | 87.7 | 89.9 | 95.1 | 99.8 | 27.0 | 96.0 | 95.9 | 63.5 | 44.8 | 70.5±0.12 |
| ROID | ViT | 53.2 | 50.2 | 51.5 | 50.5 | 48.1 | 42.3 | 42.9 | 39.7 | 37.8 | 32.7 | 25.0 | 41.2 | 35.3 | 33.9 | 34.5 | 41.3±0.03 |
| | Swin | 52.6 | 49.4 | 49.9 | 55.9 | 50.6 | 44.2 | 43.9 | 37.7 | 34.9 | 31.8 | 23.7 | 39.5 | 36.6 | 39.9 | 38.4 | 41.9±0.03 |
| | D2V | 40.7 | 39.3 | 39.4 | 55.9 | 55.9 | 46.8 | 54.2 | 35.7 | 34.9 | 29.9 | 23.5 | 35.6 | 48.6 | 41.9 | 35.0 | 41.2±0.03 |
| CMF | ViT | 51.4 | 47.3 | 48.4 | 49.4 | 45.7 | 40.3 | 38.8 | 38.6 | 37.5 | 31.8 | 24.9 | 39.2 | 33.4 | 32.2 | 33.3 | **39.5±0.03** |
| | Swin | 50.5 | 46.0 | 46.7 | 54.7 | 48.8 | 41.5 | 39.2 | 34.9 | 34.3 | 29.9 | 23.0 | 37.2 | 33.5 | 36.3 | 34.3 | **39.4±0.08** |
| | D2V | 40.3 | 38.4 | 38.3 | 52.9 | 50.0 | 42.5 | 46.1 | 33.8 | 32.5 | 28.4 | 22.1 | 34.3 | 41.7 | 38.1 | 31.3 | **38.0±0.05** |

Table 20: Average error rate (%) and their corresponding standard deviations on **ImageNet-C** in the scenario of **TC-LS** ($\delta = 0.01$) **over CS**. We choose $q = 0.00025$.

| Method | Model | Adaptation Order ($\rightarrow$) | | | | | | | | | | | | | | | Avg. |
|---|---|---|---|---|---|---|---|---|---|---|---|---|---|---|---|---|---|
| | | gaussian | shot | impulse | defocus | glass | motion | zoom | snow | frost | fog | bright | contrast | elastic | pixelate | jpeg | |
| TENT | ViT | 86.3 | 85.4 | 85.8 | 83.3 | 85.5 | 80.1 | 83.3 | 81.1 | 79.3 | 78.0 | 70.1 | 82.7 | 81.5 | 79.5 | 75.8 | 81.2±2.69 |
| | Swin | 90.3 | 89.5 | 90.5 | 90.4 | 90.7 | 84.7 | 85.9 | 84.9 | 84.9 | 82.8 | 72.6 | 85.6 | 90.6 | 92.8 | 82.2 | 86.6±2.62 |
| | D2V | 43.1 | 42.0 | 42.1 | 61.4 | 67.2 | 52.1 | 62.1 | 39.5 | 42.0 | 36.8 | 24.4 | 42.2 | 56.0 | 52.9 | 40.5 | 47.1±0.09 |
| CoTTA | ViT | 60.7 | 60.9 | 59.6 | 72.5 | 73.7 | 64.2 | 68.1 | 58.6 | 47.5 | 46.3 | 32.5 | 95.3 | 57.6 | 61.5 | 54.8 | 60.9±0.24 |
| | Swin | 65.9 | 65.6 | 70.6 | 77.5 | 81.4 | 69.7 | 72.9 | 59.1 | 52.3 | 47.0 | 32.4 | 64.8 | 75.8 | 87.7 | 65.8 | 65.9±0.09 |
| | D2V | 42.6 | 41.6 | 41.8 | 66.3 | 73.4 | 55.3 | 65.6 | 39.8 | 40.3 | 33.9 | 24.4 | 44.3 | 64.7 | 55.1 | 41.2 | 48.7±0.05 |
| RoTTA | ViT | 86.1 | 85.6 | 86.0 | 75.3 | 75.0 | 66.3 | 71.5 | 58.6 | 55.2 | 64.7 | 33.4 | 92.8 | 62.1 | 88.4 | 80.5 | 72.1±0.14 |
| | Swin | 81.3 | 80.3 | 83.3 | 84.6 | 83.1 | 78.9 | 81.7 | 63.9 | 65.9 | 73.6 | 49.6 | 84.6 | 82.6 | 96.4 | 87.4 | 78.5±0.22 |
| | D2V | 44.7 | 43.8 | 44.0 | 77.7 | 78.6 | 64.7 | 72.1 | 39.8 | 41.9 | 38.5 | 27.8 | 68.9 | 68.5 | 67.6 | 44.8 | 54.9±0.07 |
| SAR | ViT | 64.2 | 62.4 | 62.9 | 59.0 | 64.0 | 52.7 | 60.8 | 54.9 | 49.5 | 47.3 | 30.4 | 58.9 | 53.3 | 48.7 | 42.8 | 54.1±0.40 |
| | Swin | 71.8 | 70.0 | 72.2 | 74.0 | 76.6 | 61.4 | 66.1 | 66.7 | 61.6 | 57.8 | 32.6 | 62.3 | 73.0 | 79.8 | 55.6 | 65.4±0.53 |
| | D2V | 43.2 | 42.0 | 42.3 | 62.7 | 68.7 | 52.4 | 62.5 | 39.3 | 39.3 | 35.5 | 24.3 | 43.0 | 60.2 | 52.4 | 40.5 | 47.2±0.08 |
| EATA | ViT | 79.5 | 78.6 | 79.1 | 79.8 | 78.1 | 73.9 | 75.4 | 68.1 | 62.1 | 65.7 | 40.7 | 90.5 | 63.7 | 68.5 | 54.3 | 70.5±0.67 |
| | Swin | 84.6 | 83.1 | 85.4 | 86.7 | 85.8 | 76.7 | 76.0 | 72.1 | 71.4 | 69.8 | 43.2 | 86.1 | 74.3 | 89.2 | 71.5 | 77.1±0.93 |
| | D2V | 83.2 | 82.9 | 82.9 | 90.9 | 93.8 | 88.3 | 90.2 | 81.4 | 81.5 | 81.7 | 78.5 | 88.8 | 90.7 | 88.5 | 83.4 | 85.8±18.90 |
| LAME | ViT | 36.3 | 36.1 | 36.2 | 36.3 | 36.4 | 36.2 | 36.4 | 36.1 | 35.9 | 36.2 | 35.6 | 36.7 | 36.0 | 36.1 | 35.8 | 36.1±0.09 |
| | Swin | 37.5 | 37.3 | 37.5 | 37.5 | 37.8 | 37.4 | 37.7 | 37.2 | 37.1 | 37.3 | 36.6 | 37.5 | 37.5 | 37.9 | 37.3 | 37.4±0.12 |
| | D2V | 36.1 | 35.9 | 35.9 | 36.5 | 36.6 | 36.4 | 36.6 | 36.1 | 36.3 | 36.3 | 35.9 | 36.5 | 36.4 | 36.3 | 36.1 | 36.3±0.11 |
| ROID | ViT | 27.0 | 26.2 | 26.1 | 26.1 | 32.0 | 23.4 | 29.2 | 23.4 | 19.4 | 18.6 | 10.8 | 26.6 | 25.5 | 21.5 | 17.7 | 23.6±0.05 |
| | Swin | 30.9 | 30.1 | 31.0 | 34.8 | 37.2 | 26.4 | 31.1 | 26.8 | 22.8 | 23.2 | 12.3 | 27.0 | 33.6 | 36.4 | 25.2 | 28.6±0.16 |
| | D2V | 14.7 | 14.4 | 14.4 | 27.0 | 31.7 | 20.4 | 26.9 | 14.5 | 14.4 | 13.8 | 9.0 | 16.4 | 27.2 | 21.2 | 15.2 | 18.8±0.01 |
| CMF | ViT | 27.1 | 26.2 | 26.2 | 25.6 | 30.5 | 22.6 | 28.9 | 22.7 | 19.5 | 18.7 | 11.0 | 25.3 | 24.4 | 21.1 | 17.5 | **23.2±0.05** |
| | Swin | 29.2 | 28.3 | 29.0 | 35.7 | 35.5 | 25.3 | 30.0 | 24.7 | 22.2 | 22.8 | 12.4 | 25.0 | 31.4 | 31.2 | 24.4 | **27.1±0.08** |
| | D2V | 14.4 | 14.0 | 14.1 | 24.3 | 26.0 | 18.2 | 25.3 | 13.4 | 13.4 | 13.2 | 8.6 | 15.7 | 22.7 | 18.3 | 14.4 | **17.1±0.09** |

Table 21: Average error rate (%) and their corresponding standard deviations on **D109** in the scenario of **CS**.

| Method | Model | clipart | infograph | painting | real | sketch | Avg. |
|---|---|---|---|---|---|---|---|
| TENT | ResNet-50 | 56.0 | 80.8 | 50.2 | 25.4 | 64.4 | 55.6±0.08 |
| | ViT | 81.8 | 90.9 | 74.2 | 48.7 | 88.4 | 76.8±0.36 |
| | Swin | 66.1 | 83.9 | 55.8 | 25.4 | 76.5 | 61.5±0.41 |
| | D2V | 60.6 | 83.0 | 52.3 | 25.5 | 68.0 | 57.9±0.42 |
| CoTTA | ResNet-50 | 56.4 | 78.3 | 50.8 | 25.4 | 65.4 | 55.3±0.04 |
| | ViT | 56.6 | 76.4 | 44.6 | 21.9 | 67.1 | 53.3±0.04 |
| | Swin | 52.5 | 73.8 | 44.1 | 20.5 | 64.9 | 51.2±0.03 |
| | D2V | 48.6 | 73.0 | 40.8 | 20.2 | 56.6 | 47.8±0.01 |
| RoTTA | ResNet-50 | 57.2 | 78.1 | 48.5 | 24.1 | 65.9 | 54.8±0.04 |
| | ViT | 53.2 | 73.4 | 42.9 | 21.6 | 63.2 | 50.9±0.05 |
| | Swin | 49.1 | 71.4 | 41.4 | 20.1 | 60.9 | 48.6±0.05 |
| | D2V | 46.8 | 71.9 | 40.3 | 20.1 | 55.1 | 46.8±0.03 |
| SAR | ResNet-50 | 54.4 | 77.3 | 49.2 | 24.9 | 62.3 | 53.6±0.07 |
| | ViT | 66.2 | 82.9 | 54.1 | 26.1 | 76.4 | 61.2±0.36 |
| | Swin | 55.8 | 78.4 | 46.7 | 21.6 | 67.3 | 53.9±0.08 |
| | D2V | 48.8 | 74.0 | 40.6 | 20.1 | 56.9 | 48.1±0.08 |
| EATA | ResNet-50 | 53.0 | 78.0 | 49.4 | 24.3 | 60.8 | 53.1±0.09 |
| | ViT | 50.1 | 71.5 | 41.7 | 20.7 | 58.4 | 48.5±0.11 |
| | Swin | 49.3 | 71.8 | 42.0 | 19.9 | 61.1 | 48.8±0.12 |
| | D2V | 46.3 | 71.6 | 39.4 | 19.4 | 54.5 | 46.2±0.05 |
| ROID | ResNet-50 | 51.0 | 75.8 | 46.7 | 23.7 | 57.3 | 50.9±0.04 |
| | ViT | 48.6 | 69.7 | 40.6 | 20.5 | 55.2 | 46.9±0.02 |
| | Swin | 48.2 | 69.9 | 40.6 | 19.6 | 57.7 | 47.2±0.07 |
| | D2V | 44.6 | 70.0 | 38.0 | 19.3 | 53.2 | 45.0±0.01 |
| CMF | ResNet-50 | 47.6 | 75.8 | 46.3 | 24.0 | 53.5 | **49.4±0.21** |
| | ViT | 44.0 | 67.8 | 39.6 | 20.6 | 50.2 | **44.5±0.08** |
| | Swin | 43.9 | 67.8 | 39.3 | 19.7 | 53.3 | **44.8±0.04** |
| | D2V | 41.7 | 67.4 | 36.4 | 18.7 | 49.8 | **42.8±0.05** |

Table 22: Average error rate (%) and their corresponding standard deviations on **D109** in the scenario of **TC-CS**. We choose $q = 0.00025$ for ResNet-50.

| Method | Model | Adaptation Order (→) | | | | | Avg. |
|--------|-------|---------|----------|----------|------|--------|------|
| | | clipart | infograph | painting | real | sketch | |
| TENT | ResNet-50 | 53.0 | 78.1 | 48.0 | 24.7 | 60.2 | 52.8±0.04 |
| | ViT | 57.3 | 86.1 | 80.9 | 93.1 | 99.2 | 83.3±0.13 |
| | Swin | 52.6 | 80.1 | 59.9 | 43.7 | 95.6 | 66.4±0.33 |
| | D2V | 49.1 | 78.8 | 56.6 | 40.4 | 89.5 | 62.9±0.21 |
| CoTTA | ResNet-50 | 54.7 | 79.7 | 49.5 | 24.6 | 62.5 | 54.2±0.07 |
| | ViT | 56.8 | 76.0 | 45.0 | 21.8 | 66.8 | 53.3±0.03 |
| | Swin | 52.6 | 73.7 | 44.2 | 20.5 | 64.8 | 51.2±0.01 |
| | D2V | 48.7 | 72.9 | 41.0 | 20.1 | 56.5 | 47.8±0.02 |
| RoTTA | ResNet-50 | 55.1 | 78.3 | 48.5 | 24.0 | 59.3 | 53.0±0.03 |
| | ViT | 56.3 | 74.6 | 43.5 | 21.5 | 61.4 | 51.4±0.03 |
| | Swin | 52.1 | 72.8 | 42.1 | 20.0 | 58.5 | 49.1±0.03 |
| | D2V | 48.6 | 72.6 | 40.7 | 20.0 | 53.9 | 47.2±0.03 |
| SAR | ResNet-50 | 53.3 | 77.9 | 47.5 | 24.5 | 58.7 | 52.6±0.01 |
| | ViT | 55.7 | 82.6 | 53.0 | 21.5 | 73.5 | 57.3±0.41 |
| | Swin | 51.2 | 78.4 | 48.6 | 20.8 | 68.6 | 53.5±1.05 |
| | D2V | 48.3 | 74.4 | 42.9 | 20.3 | 56.5 | 48.5±0.10 |
| EATA | ResNet-50 | 51.1 | 76.6 | 47.1 | 24.0 | 57.6 | 51.3±0.25 |
| | ViT | 52.4 | 70.1 | 40.8 | 20.6 | 52.4 | 47.2±0.10 |
| | Swin | 50.3 | 70.3 | 41.2 | 19.5 | 55.4 | 47.4±0.18 |
| | D2V | 47.3 | 71.0 | 39.2 | 19.2 | 52.2 | 45.8±0.06 |
| ROID | ResNet-50 | 45.6 | 74.3 | 44.8 | 23.1 | 52.7 | 48.1±0.07 |
| | ViT | 46.2 | 68.2 | 39.9 | 20.5 | 50.2 | 45.0±0.04 |
| | Swin | 46.1 | 67.7 | 39.8 | 19.7 | 52.2 | 45.1±0.10 |
| | D2V | 44.0 | 69.0 | 37.7 | 19.3 | 51.2 | 44.2±0.06 |
| CMF | ResNet-50 | 45.2 | 74.3 | 44.9 | 22.9 | 51.9 | **47.8±0.06** |
| | ViT | 43.9 | 66.4 | 39.4 | 20.0 | 47.4 | **43.4±0.07** |
| | Swin | 44.2 | 66.6 | 39.0 | 19.2 | 48.8 | **43.6±0.12** |
| | D2V | 43.0 | 66.5 | 36.3 | 18.5 | 47.3 | **42.3±0.11** |

Table 23: Average error rate (%) and their corresponding standard deviations on **D109** in the scenario of **TC-LS ($\delta = 0.0$) over TC-CS**. We choose $q = 0.00025$.

| Method | Model | Adaptation Order ($\rightarrow$) | | | | | Avg. |
| | | clipart | infograph | painting | real | sketc | |
|--------|-------|---------|-----------|----------|------|-------|------|
| TENT | ViT | 58.2 | 86.1 | 82.2 | 94.2 | 99.2 | 84.0±0.01 |
| | Swin | 52.9 | 79.9 | 60.5 | 51.0 | 98.4 | 68.5±0.17 |
| | D2V | 49.2 | 78.3 | 56.3 | 40.7 | 88.3 | 62.6±0.01 |
| CoTTA | ViT | 56.8 | 76.1 | 45.0 | 22.1 | 66.8 | 53.3±0.04 |
| | Swin | 52.6 | 73.7 | 44.2 | 20.6 | 64.9 | 51.2±0.01 |
| | D2V | 48.7 | 72.9 | 41.0 | 20.1 | 56.6 | 47.8±0.01 |
| RoTTA | ViT | 56.7 | 75.5 | 44.9 | 22.7 | 68.3 | 53.6±0.08 |
| | Swin | 52.5 | 73.3 | 43.2 | 20.3 | 63.7 | 50.6±0.03 |
| | D2V | 48.7 | 72.8 | 41.1 | 20.5 | 56.6 | 48.0±0.02 |
| SAR | ViT | 64.5 | 80.9 | 51.8 | 21.8 | 73.5 | 58.5±0.40 |
| | Swin | 60.3 | 77.7 | 50.0 | 20.7 | 68.3 | 55.4±0.17 |
| | D2V | 48.7 | 75.4 | 46.8 | 20.1 | 56.6 | 49.5±0.04 |
| EATA | ViT | 53.5 | 71.1 | 52.8 | 34.6 | 80.8 | 58.6±1.45 |
| | Swin | 51.2 | 71.3 | 45.7 | 29.8 | 73.2 | 54.2±0.99 |
| | D2V | 47.3 | 70.5 | 40.3 | 20.0 | 52.6 | 46.1±0.37 |
| LAME | ViT | 29.5 | 75.6 | 18.7 | 9.7 | 42.6 | 35.2±0.55 |
| | Swin | 24.7 | 62.9 | 18.6 | 8.3 | 36.2 | 30.1±0.16 |
| | D2V | 26.0 | 68.8 | 19.2 | 8.0 | 26.7 | 29.7±0.15 |
| ROID | ViT | 28.9 | 55.8 | 24.9 | 11.1 | 36.3 | 31.4±0.07 |
| | Swin | 26.6 | 54.0 | 24.2 | 10.6 | 36.2 | 30.3±0.25 |
| | D2V | 25.4 | 56.0 | 21.2 | 10.5 | 33.3 | 29.3±0.03 |
| CMF | ViT | 28.5 | 54.8 | 25.5 | 11.0 | 35.4 | **31.0±0.10** |
| | Swin | 26.6 | 52.8 | 23.8 | 10.5 | 34.5 | **29.6±0.21** |
| | D2V | 24.9 | 52.8 | 20.4 | 10.2 | 30.9 | **27.8±0.12** |

Table 24: Average error rate (%) and their corresponding standard deviations on **D109** in the scenario of **TC-LS ($\delta = 0.01$) over TC-CS**. We choose $q = 0.00025$.

| Method | Model | Adaptation Order ($\rightarrow$) | | | | | Avg. |
| | | clipart | infograph | painting | real | sketc | |
|---|---|---|---|---|---|---|---|
| TENT | ViT | 57.5 | 86.1 | 80.8 | 93.5 | 99.2 | 83.4±0.30 |
| | Swin | 52.7 | 79.9 | 59.5 | 48.6 | 96.8 | 67.5±1.86 |
| | D2V | 49.1 | 78.7 | 56.5 | 40.9 | 89.3 | 62.9±0.35 |
| CoTTA | ViT | 56.8 | 76.1 | 45.0 | 22.0 | 66.9 | 53.4±0.01 |
| | Swin | 52.6 | 73.7 | 44.2 | 20.6 | 64.9 | 51.2±0.02 |
| | D2V | 48.7 | 72.9 | 41.0 | 20.1 | 56.5 | 47.9±0.00 |
| RoTTA | ViT | 56.7 | 75.4 | 44.6 | 22.5 | 67.3 | 53.3±0.08 |
| | Swin | 52.5 | 73.3 | 43.1 | 20.3 | 62.8 | 50.4±0.03 |
| | D2V | 48.7 | 72.8 | 41.1 | 20.4 | 56.2 | 47.8±0.00 |
| SAR | ViT | 62.8 | 82.2 | 51.1 | 21.8 | 74.8 | 58.6±0.80 |
| | Swin | 52.5 | 78.9 | 46.8 | 20.7 | 69.5 | 53.7±0.53 |
| | D2V | 48.3 | 75.2 | 45.1 | 20.4 | 56.4 | 49.1±0.14 |
| EATA | ViT | 53.3 | 70.3 | 42.1 | 26.7 | 61.3 | 50.7±1.20 |
| | Swin | 50.5 | 70.6 | 43.4 | 22.1 | 61.6 | 49.6±0.41 |
| | D2V | 47.3 | 70.9 | 41.5 | 22.7 | 52.9 | 47.1±1.08 |
| LAME | ViT | 43.2 | 82.9 | 29.4 | 14.7 | 54.0 | 44.8±0.69 |
| | Swin | 38.8 | 73.4 | 29.2 | 11.4 | 46.8 | 39.9±0.77 |
| | D2V | 39.8 | 79.2 | 29.6 | 12.5 | 38.1 | 39.9±0.56 |
| ROID | ViT | 31.0 | 56.9 | 25.5 | 11.5 | 35.8 | 32.2±0.10 |
| | Swin | 28.6 | 55.0 | 24.5 | 11.2 | 36.4 | 31.1±0.11 |
| | D2V | 27.9 | 57.6 | 22.3 | 10.9 | 34.7 | 30.7±0.09 |
| CMF | ViT | 30.7 | 56.1 | 25.5 | 11.4 | 35.2 | **31.8±0.10** |
| | Swin | 28.3 | 53.6 | 23.8 | 11.0 | 35.0 | **30.3±0.24** |
| | D2V | 27.0 | 53.7 | 21.2 | 10.4 | 30.5 | **28.6±0.11** |

Table 25: Average error rate (%) and their corresponding standard deviations on **D109** in the scenario of **TC-LS ($\delta = 0.1$) over TC-CS**.

| Method | Model | Adaptation Order ($\rightarrow$) | | | | | Avg. |
| | | clipart | infograph | painting | real | sketc | |
|---|---|---|---|---|---|---|---|
| TENT | ViT | 57.3 | 86.3 | 81.0 | 93.9 | 99.2 | 83.5±0.08 |
| | Swin | 52.7 | 80.0 | 59.4 | 44.6 | 95.3 | 66.4±0.38 |
| | D2V | 49.1 | 78.9 | 56.5 | 40.5 | 89.3 | 62.9±0.29 |
| CoTTA | ViT | 56.8 | 76.0 | 45.0 | 21.8 | 66.8 | 53.3±0.04 |
| | Swin | 52.6 | 73.7 | 44.2 | 20.5 | 64.8 | 51.2±0.03 |
| | D2V | 48.7 | 72.9 | 41.0 | 20.1 | 56.5 | 47.9±0.00 |
| RoTTA | ViT | 56.5 | 75.1 | 44.1 | 22.1 | 65.1 | 52.6±0.06 |
| | Swin | 52.3 | 73.0 | 42.6 | 20.2 | 61.1 | 49.8±0.05 |
| | D2V | 48.7 | 72.7 | 40.9 | 20.2 | 55.3 | 47.6±0.01 |
| SAR | ViT | 59.0 | 83.2 | 50.5 | 21.6 | 74.4 | 57.7±0.56 |
| | Swin | 51.6 | 78.8 | 47.8 | 20.8 | 68.0 | 53.4±0.70 |
| | D2V | 48.4 | 74.6 | 43.5 | 20.3 | 56.4 | 48.6±0.04 |
| EATA | ViT | 52.2 | 69.9 | 41.2 | 21.0 | 52.7 | 47.4±0.16 |
| | Swin | 50.2 | 70.0 | 41.2 | 19.8 | 55.9 | 47.4±0.21 |
| | D2V | 47.3 | 70.7 | 39.2 | 19.4 | 51.9 | 45.7±0.08 |
| LAME | ViT | 74.0 | 95.3 | 57.6 | 37.2 | 80.5 | 68.9±0.24 |
| | Swin | 70.3 | 92.2 | 57.7 | 27.9 | 74.9 | 64.6±0.25 |
| | D2V | 69.8 | 94.5 | 57.6 | 32.8 | 68.2 | 64.6±0.25 |
| ROID | ViT | 37.9 | 63.2 | 30.9 | 13.6 | 41.3 | 37.3±0.12 |
| | Swin | 36.8 | 62.1 | 30.2 | 13.2 | 42.3 | 36.9±0.11 |
| | D2V | 35.5 | 63.7 | 28.0 | 12.8 | 41.7 | 36.3±0.06 |
| CMF | ViT | 36.6 | 61.2 | 30.5 | 13.7 | 38.4 | **36.1±0.11** |
| | Swin | 35.7 | 59.8 | 28.8 | 12.5 | 38.4 | **35.0±0.05** |
| | D2V | 34.4 | 60.3 | 26.6 | 12.1 | 36.8 | **34.1±0.13** |

Table 26: Average error rate (%) and their corresponding standard deviations on **D109** in the scenario of **TC-LS ($\delta = 1.0$) over TC-CS**.

| Method | Model | Adaptation Order ($\rightarrow$) | | | | | Avg. |
|--------|-------|---------|----------|----------|------|-------|------|
| | | clipart | infograph | painting | real | sketc | |
| TENT | ViT | 57.2 | 86.1 | 81.1 | 93.6 | 99.2 | 83.4±0.07 |
| | Swin | 52.6 | 79.9 | 59.3 | 43.2 | 95.4 | 66.1±0.23 |
| | D2V | 49.1 | 78.8 | 56.5 | 40.0 | 89.3 | 62.7±0.17 |
| CoTTA | ViT | 56.8 | 76.0 | 45.0 | 21.8 | 66.7 | 53.3±0.01 |
| | Swin | 52.6 | 73.7 | 44.2 | 20.5 | 64.9 | 51.2±0.02 |
| | D2V | 48.7 | 72.9 | 41.0 | 20.1 | 56.5 | 47.8±0.01 |
| RoTTA | ViT | 56.4 | 74.8 | 43.7 | 21.7 | 62.9 | 51.9±0.06 |
| | Swin | 52.1 | 72.8 | 42.1 | 20.0 | 59.3 | 49.3±0.04 |
| | D2V | 48.6 | 72.7 | 40.7 | 20.0 | 54.4 | 47.3±0.02 |
| SAR | ViT | 54.4 | 82.6 | 53.7 | 21.5 | 74.6 | 57.4±0.12 |
| | Swin | 51.6 | 78.6 | 52.0 | 20.7 | 69.6 | 54.5±0.68 |
| | D2V | 48.3 | 74.4 | 43.1 | 20.3 | 56.4 | 48.5±0.09 |
| EATA | ViT | 52.5 | 70.1 | 40.6 | 20.5 | 52.3 | 47.2±0.04 |
| | Swin | 50.4 | 70.1 | 41.0 | 19.5 | 55.8 | 47.4±0.10 |
| | D2V | 47.3 | 71.1 | 39.1 | 19.2 | 52.0 | 45.7±0.04 |
| LAME | ViT | 96.6 | 99.6 | 85.1 | 71.1 | 97.8 | 90.0±0.09 |
| | Swin | 94.1 | 99.5 | 85.0 | 59.1 | 96.8 | 86.9±0.24 |
| | D2V | 94.8 | 99.6 | 84.7 | 68.5 | 94.1 | 88.3±0.13 |
| ROID | ViT | 44.1 | 67.5 | 37.6 | 17.0 | 48.2 | 42.9±0.03 |
| | Swin | 44.1 | 67.0 | 37.3 | 16.5 | 50.3 | 43.0±0.06 |
| | D2V | 42.4 | 68.2 | 35.0 | 15.8 | 49.3 | 42.2±0.04 |
| CMF | ViT | 42.4 | 65.7 | 36.9 | 16.3 | 45.4 | **41.3±0.06** |
| | Swin | 42.8 | 65.4 | 35.9 | 15.5 | 46.7 | **41.3±0.04** |
| | D2V | 41.3 | 65.5 | 33.5 | 15.0 | 45.1 | **40.1±0.10** |

Table 27: Average error rate (%) and their corresponding standard deviations on **D109** in the scenario of **TC-LS ($\delta = 5.0$) over TC-CS**.

| Method | Model | Adaptation Order ($\rightarrow$) | | | | | Avg. |
|---|---|---|---|---|---|---|---|
| | | clipart | infograph | painting | real | sketc | |
| TENT | ViT | 57.3 | 86.0 | 80.5 | 92.8 | 99.2 | 83.1±0.26 |
| | Swin | 52.6 | 79.9 | 59.3 | 43.0 | 95.3 | 66.0±0.07 |
| | D2V | 49.1 | 78.8 | 56.5 | 40.1 | 89.6 | 62.8±0.06 |
| CoTTA | ViT | 56.8 | 76.0 | 44.9 | 21.8 | 66.8 | 53.3±0.02 |
| | Swin | 52.6 | 73.7 | 44.2 | 20.5 | 64.8 | 51.2±0.01 |
| | D2V | 48.7 | 72.8 | 41.0 | 20.1 | 56.5 | 47.8±0.01 |
| RoTTA | ViT | 56.3 | 74.7 | 43.6 | 21.7 | 62.5 | 51.7±0.05 |
| | Swin | 52.1 | 72.7 | 42.0 | 20.0 | 59.1 | 49.2±0.03 |
| | D2V | 48.6 | 72.6 | 40.7 | 20.0 | 54.2 | 47.2±0.01 |
| SAR | ViT | 55.2 | 82.8 | 53.7 | 21.5 | 73.6 | 57.3±0.22 |
| | Swin | 51.8 | 78.5 | 49.2 | 20.7 | 69.7 | 54.0±0.72 |
| | D2V | 48.3 | 74.3 | 43.0 | 20.3 | 56.4 | 48.4±0.12 |
| EATA | ViT | 52.3 | 70.0 | 40.7 | 20.5 | 52.4 | 47.2±0.08 |
| | Swin | 50.6 | 70.4 | 41.1 | 19.5 | 55.1 | 47.3±0.05 |
| | D2V | 47.3 | 71.0 | 39.0 | 19.2 | 51.9 | 45.7±0.06 |
| LAME | ViT | 98.8 | 99.6 | 92.1 | 76.9 | 99.1 | 93.3±0.17 |
| | Swin | 98.0 | 99.6 | 91.9 | 64.3 | 99.3 | 90.6±0.23 |
| | D2V | 98.4 | 99.6 | 91.7 | 75.7 | 98.3 | 92.8±0.16 |
| ROID | ViT | 45.3 | 67.8 | 38.9 | 18.3 | 49.2 | 43.9±0.09 |
| | Swin | 45.0 | 67.8 | 38.9 | 17.6 | 51.3 | 44.1±0.06 |
| | D2V | 43.5 | 68.7 | 36.4 | 17.0 | 50.3 | 43.2±0.04 |
| CMF | ViT | 43.6 | 66.1 | 38.5 | 17.6 | 46.5 | **42.5±0.08** |
| | Swin | 43.8 | 66.3 | 37.9 | 16.7 | 47.7 | **42.5±0.07** |
| | D2V | 42.4 | 66.0 | 34.9 | 16.1 | 46.0 | **41.1±0.06** |

Table 28: Average error rate (%) and their corresponding standard deviations on **D109** in the scenario of **TC-LS ($\delta = 0.1$) over CS**. We choose $q = 0.00025$.

| Method | Model | Adaptation Order ($\rightarrow$) | | | | | Avg. |
|---|---|---|---|---|---|---|---|
| | | clipart | infograph | painting | real | sketc | |
| TENT | ViT | 82.1 | 91.1 | 73.5 | 47.4 | 88.6 | 76.5±0.52 |
| | Swin | 66.5 | 84.0 | 55.6 | 25.1 | 76.5 | 61.5±0.31 |
| | D2V | 60.4 | 83.0 | 51.8 | 25.2 | 67.7 | 57.6±0.46 |
| CoTTA | ViT | 56.8 | 76.4 | 44.7 | 21.9 | 67.3 | 53.4±0.02 |
| | Swin | 52.4 | 73.9 | 44.1 | 20.5 | 65.0 | 51.2±0.03 |
| | D2V | 48.6 | 72.9 | 40.8 | 20.2 | 56.5 | 47.8±0.03 |
| RoTTA | ViT | 56.8 | 75.2 | 44.9 | 22.5 | 67.5 | 53.4±0.05 |
| | Swin | 51.5 | 72.8 | 42.9 | 20.4 | 64.1 | 50.4±0.07 |
| | D2V | 48.5 | 72.8 | 41.1 | 20.5 | 56.6 | 47.9±0.05 |
| SAR | ViT | 66.1 | 82.5 | 54.0 | 26.0 | 76.4 | 61.0±0.51 |
| | Swin | 55.6 | 77.7 | 46.4 | 21.5 | 67.1 | 53.6±0.24 |
| | D2V | 49.6 | 74.7 | 41.1 | 20.2 | 57.5 | 48.6±0.35 |
| EATA | ViT | 55.3 | 73.6 | 45.9 | 24.9 | 64.6 | 52.9±2.98 |
| | Swin | 51.0 | 71.9 | 44.3 | 21.1 | 63.2 | 50.3±0.25 |
| | D2V | 45.5 | 70.7 | 39.4 | 19.7 | 54.1 | 45.9±0.13 |
| LAME | ViT | 28.7 | 27.9 | 32.2 | 30.5 | 30.2 | 29.9±0.18 |
| | Swin | 27.4 | 26.6 | 31.0 | 29.0 | 28.8 | 28.6±0.23 |
| | D2V | 27.7 | 27.2 | 31.6 | 29.6 | 29.1 | 29.1±0.19 |
| ROID | ViT | 27.1 | 47.4 | 24.8 | 13.1 | 33.3 | 29.1±0.09 |
| | Swin | 24.7 | 47.0 | 24.0 | 12.7 | 32.4 | 28.2±0.05 |
| | D2V | 22.3 | 46.0 | 21.9 | 12.3 | 29.1 | 26.3±0.07 |
| CMF | ViT | 26.7 | 47.1 | 24.4 | 13.0 | 32.3 | **28.7±0.19** |
| | Swin | 23.8 | 46.4 | 23.4 | 12.6 | 30.7 | **27.3±0.05** |
| | D2V | 20.1 | 45.1 | 20.9 | 12.0 | 26.4 | **24.9±0.10** |

