# OpenReview forum: "Continual Momentum Filtering on Parameter Space for Online Test-time Adaptation"
_ICLR.cc/2024/Conference — ICLR 2024 poster_

### Official Review · Reviewer_aPnP · 2023-10-30

**Soundness:** 3 good
**Presentation:** 3 good
**Contribution:** 3 good
**Rating:** 8
**Confidence:** 4

**Summary:**

The paper focuses on online test time adaptation and introduces a Kalman Filter (KF) based approach on momentum filtering of DNNs.  The proposed method, referred to as CMF,  integrates a SGD-based optimization process with a KF-based inference (filtering) process. Some simplification techniques, such as momentum smoothing, have been employed to reduce time complexity. The experimental results demonstrate the competitiveness of the proposed method across multiple datasets.

**Strengths:**

The paper studies a practical and important problem: online test-time adaptation, focusing on mitigating catastrophic forgetting during adaptation.

The proposed method, which employs a KF-based approach to filter the momentum parameter of DNNs, provides a sound solution to mitigate catastrophic forgetting in online test-time adaptation.

Overall, the paper is well-structured and effectively communicates the details of the proposed method.

Furthermore, in the experimental evaluation, the proposed method demonstrates superior performance compared to previous approaches, as reported in the paper.

**Weaknesses:**

1) This paper could benefit from a comparison with related works that utilize the Kalman filter for online adaptation tasks, as seen in [1,2].  Such a comparison or discussion regarding the difference between the proposed method and [1,2] would be beneficial.

2) It is very common to use replay-based approaches to overcome catastrophic forgetting, such as ER [3] and A-GEM [4].  An analysis of the advantages and disadvantages of the proposed approach in contrast to replay-based methods would provide a more comprehensive understanding of how it addresses the issue of catastrophic forgetting.

3) In the alation study on “Effectiveness of the source-conjugated transition model”, do the hyperparameters alpha and gamma remain the same? It is better to explore the effects of hyperparameters alpha and gamma individually. Since the roles of alpha and gamma are distinct and independent, isolating their impact would provide a more detailed understanding.

4) The paper simplifies the inference process by using scalar parameters to replace matrix parameters in the KF process. It would be helpful to conduct an experiment on a smaller model to compare the performance of the simplified version with the original matrix version of the method.

[1] A. Abuduweili, et al, “Robust online model adaptation by extended kalman filter with exponential moving average and dynamic multi-epoch strategy”, L4DC 2020.
[2] Y. Cheng, et al. "Human motion prediction using semi-adaptable neural networks." ACC, 2019.
[3] A. Chaudhry, et al “On tiny episodic memories in continual learning”, arxiv 2019.
[4] A. Chaudhry, et al “Efficient lifelong learning with a-gem”, ICLR 2018.

**Questions:**

1) How does the proposed method compare to related Kalman filter-based techniques for adaptation, as discussed in [1,2]?

2) What are the advantages and disadvantages of the proposed approach in contrast to replay-based methods [3,4] for mitigating catastrophic forgetting?

3) In the alation study on “Effectiveness of the source-conjugated transition model”, do the hyperparameters alpha and gamma remain the same? It is better to explore the effects of hyperparameters alpha and gamma individually.

4) What is the performance and computational cost comparison between the simplified scalar version of the KF process and the original matrix version of the KF process?

---

> ### Author Response · Authors · 2023-11-16
> **Response to Reviewer aPnP [1/2]**
>
> ### **Overall**
>
> We have thoroughly reviewed and considered the insightful questions posed by reviewer **aPnP**. Our efforts have been directed towards enhancing the quality and clarity of our paper through this review process.
>
> **By adding a section on related works in Appendix D.1 and D.2,** we have addressed the papers mentioned in your questions and clarified the distinctions between CMF and other studies that have applied Bayesian filters to different adaptation tasks.
>
> Our focus is on the methods of Online Test-Time Adaptation (OTTA), as discussed in Section 2.1, which are characterized as unsupervised source-free domain adaptations. These methods require immediate adaptation to samples of real-time streaming data, where there are no labels for the target data, and the use of source data is not feasible. Additionally, multiple epochs are not allowed, necessitating immediate adaptation to each sample and subsequent prediction.
>
> In this context, we have endeavored to comprehensively answer the questions raised by the reviewer.
>
> ### **Response**
>
> **Q1. How does the proposed method compare to related Kalman filter-based techniques for adaptation, as discussed in [1,2]?**
>
> **A1 (other Kalman filter).** We have not found studies applying Bayesian filters like KF or EKF to OTTA. **The studies [1, 2]** focus on multi-epoch adaptation problems with **observations being natural data (e.g., images) or labels (e.g., labels for human motion)**. However, **CMF** focuses on OTTA problems with **DNN’s parameter $\theta_t$ as the observation.**
>
> → [1] designs the emission model $\mathcal{N}(Y|f(\theta_{t-1}, X_{t-1}), I)$ with a non-linear function $f$ since natural data or labels as input $X$ have a non-linear relationship. In contrast, CMF’s emission model $p(\theta^{(t)}|\phi^{(t)})=\mathcal{N} (\theta^{(t)}|H\phi^{(t)}, R)$ targets DNN’s parameters, and **the relationship is often set as a linear system in previous studies [a, b, c].**
>
> → EKF requires linear approximation of non-linear functions, often leading to instability and reduced accuracy.
>
> → CMF, on the other hand, **does not require linear approximation with EKF**, ensuring stability. Furthermore, the inverse matrix operation of KF is removed in the simplified inference process (Section 3.4), allowing for stable operations. **Thus, CMF is numerically stable, simple, and has a relatively low computation cost. Additionally, unlike [1, 2], CMF introduces KF to refine noisy parameters rather than noisy labels.**
>
> [a] Izmailov, Pavel et al. “Averaging Weights Leads to Wider Optima and Better Generalization.” *Conference on Uncertainty in Artificial Intelligence* (2018).
>
> [b] Garipov, Timur, et al. "Loss surfaces, mode connectivity, and fast ensembling of dnns." *Advances in neural information processing systems* 31 (2018).
>
> [c] Guo, Hao, Jiyong Jin, and Bin Liu. "Stochastic weight averaging revisited." *Applied Sciences* 13.5 (2023): 2935.
>
> **Q2. What are the advantages and disadvantages of the proposed approach in contrast to replay-based methods [3,4] for mitigating catastrophic forgetting?**
>
> **A2 (overcoming catastrophic forgetting).**  We have not identified studies actively using methods like A-GEM or ER in OTTA. Fundamentally, OTTA cannot use source domain data, creating ambiguity in defining 'past tasks' for ER or A-GEM approaches.
>
> → In OTTA, information about past tasks is typically derived from gradients or data from the source domain, which is generally not permissible in OTTA scenarios. An alternative approach would be to identify past tasks within streaming data for applying A-GEM or ER, but this requires additional domain-aware methods, complicating the application.
>
> → However, there are OTTA methods like [d] that use memory similar to ER, storing part of the streaming data during adaptation and balancing classes among samples to mitigate catastrophic forgetting. Yet, this approach has downsides in terms of privacy, as mentioned in Section 1.
>
> → **For these reasons, as mentioned in Section 2.2, most OTTA methods continuously inject source model parameters into target model parameters during the adaptation process. In contrast, our proposed CMF method computes a new source model $\phi_t$ (i.e., the hidden model), inferred from both the target and source models, to mitigate the issue of static source models overly relying on past tasks.**
>
> → Nonetheless, actively introducing ER or A-GEM into OTTA could be a valuable follow-up study, as it allows explicit control over past information.
>
> [d] Gong, Taesik, et al. "NOTE: Robust continual test-time adaptation against temporal correlation." *Advances in Neural Information Processing Systems* 35 (2022): 27253-27266.

---

> ### Author Response · Authors · 2023-11-16
> **Response to Reviewer aPnP [2/2]**
>
> **Q3. In the ablation study on 'Effectiveness of the source-conjugated transition model', do the hyperparameters alpha and gamma remain the same? It is better to explore the effects of hyperparameters alpha and gamma individually.**
>
> **A3.** In the ablation study, as indicated in the caption of Figure 2, we initially fixed $\gamma=0.99$ and varied $\alpha$ to assess its impact. Following your suggestion, we recognized the value in exploring the effects of varying $\gamma$ while keeping $\alpha$ constant.
>
> → Consequently, we conducted additional experiments with $\alpha=0.99$ fixed and varying $\gamma$. The results are as follows:
>
> | Model | Source | TENT | 0.999 | 0.99 | 0.95 | 0.9 | 0.8 |
> | --- | --- | --- | --- | --- | --- | --- | --- |
> | ViT | 60.2 | 54.5±0.04 | 45.2±0.11 | 44.8±0.12 | 45.5±0.09 | 46.7±0.07 | 49.18±0.04 |
> | Swin | 64.0 | 64.0±0.14 | 47.1±0.21 | 46.6±0.12 | 47.2±0.14 | 48.5±0.20 | 50.68±0.28 |
> | D2V | 51.8 | 51.9±0.09 | 43.2±0.22 | 43.5±0.04 | 45.3±0.04 | 46.8±0.06 | 48.64±0.11 |
>
> → The results of these experiments, which provide insights into the individual effects of these hyperparameters, have been incorporated into Appendix C.5.
>
> **Q4. What is the performance and computational cost comparison between the simplified scalar version of the KF process and the original matrix version of the KF process?**
>
> **A4.** Implementing the matrix version of CMF requires handling a matrix of size $d^2$, where $d$ represents the dimension of the target parameter, which in our experiments ranged from $128$ to $1024$. The KF process necessitates matrix inversion for all target DNN's parameters to calculate the Kalman gain $K$. For example, for the ViT model, the calculation is required for $50$ matrices. However, due to the substantial size of this matrix, computations using the PyTorch toolkit are not only exceedingly slow but also prone to errors, leading to numerical instability.
>
> → Unfortunately, these limitations within the toolkit render conducting experiments with the original matrix version of KF currently unfeasible. To circumvent this, CMF introduces a simplified inference process in Section 3.4. Our extensive experiments have demonstrated that this scalar setting significantly enhances performance with only a minimal increase in computational cost. The results of these experiments are detailed in Table 1 - 5 and Appendix C.1.
>
> -> To illustrate the computational efficiency of the scalar version, we also provide actual execution times (in seconds) for each model and OTTA method in the TC-CS scenario on the ImageNet-C dataset On average, CMF consumed a very small 0.55% additional computation compared to the highest performing recorded ROID.
>
> | Method | Model |  |  |
> | --- | --- | --- | --- |
> |  | ViT | Swin | D2V |
> | CMF | 159.4 | 255.2 | 162.0 |
> | ROID | 158.6 | 254.4 | 160.6 |
> | SAR | 181.2 | 294.1 | 180.8 |
> | EATA | 112.6 | 176.8 | 64.0 |
> | TENT | 102.5 | 153.6 | 101.3 |
>
>
> → Furthermore, it is important to note that even in its scalar form, CMF continues to perform exact Bayesian filtering for the linear Gaussian model, ensuring the accuracy and reliability of the inference processes.

---

> ### Author Response · Authors · 2023-11-22
> **A gentle reminder for discussion**
>
> Dear Reviewer aPnP,
>
> Thank you again for your insightful review of our submission. We have consistently endeavored to provide detailed responses to address the concerns you have raised along your commnets. Your valuable comments have also helped us to make our study more robust and clear.
>
>
> There is now one day left in the author-reviewer discussion period, during which we hope to have the reviewer review our response and revised submission. If you have any additional comments, we will do our best to answer any questions.
>
>
> We sincerely appreciate your efforts and time in reviewing our submission.
>
>
> Author(s)

---

> > ### Comment · Reviewer_aPnP · 2023-11-22
> > **Response to the Rebuttal**
> >
> > Thank you for your comprehensive response to my inquiries.
> > You have addressed most of my questions/concerns. After reading the rebuttal and other reviews, I am inclined to adjust my initial score from 6 to 8.

---

> ### Author Response · Authors · 2023-11-23
> **Thanks for your positive feedback with score updating**
>
> Thank you for your valuable feedback and positive support.
>
> We're glad we were able to address your concerns.

---

### Official Review · Reviewer_gT9R · 2023-10-31

**Soundness:** 3 good
**Presentation:** 3 good
**Contribution:** 3 good
**Rating:** 6
**Confidence:** 4

**Summary:**

The paper introduces a novel framework called Continual Momentum Filtering (CMF) that leverages the Kalman Filter to strike a balance between model adaptability and information retention. The CMF framework alleviates the catastrophic forgetting issue and provides high adaptability to shifting data distributions. The paper provides examples of real-world situations where the CMF framework has been validated, including scenarios involving covariate and label shifts in speech recognition tasks and ImageNet-C. The results show that the CMF consistently outperforms state-of-the-art methods.

**Strengths:**

The investigated problem, namely adaptability and anti-forgetting tradeoff, is practical for the real-world deployment of TTA methods. The resulting CMF framework is simple yet effective.

Experimental evaluations on various models, datasets and scenarios are thorough and demonstrate the effectiveness of the proposed framework.

**Weaknesses:**

The performance gains compared with ROID are a bit marginal.

**Questions:**

Are there any sensitivity analyses regarding the parameter $I$ in algorithm 1?

I am curious about the performance of replacing the “weight ensemble” in ROID with CMF？ (namely ROID with CMF).

How about the performance of “DW-SLR + SCE” without CMF?

How about the in-distribution performance of compared methods after the adaptation of out-of-distribution data? Please refer to the comparison manner proposed in EATA.

Could the authors provide a computational complexity (wall-clock GPU time) comparison regarding the proposed CMF?

Could CMF help MEMO work stably in the online setting?

More motivation/explanations from the high level about why CMF could achieve a better tradeoff between adaptability and information retention are preferred.

Could the authors provide implementation details (hyperparameters) of baselines on different models and datasets?

---

> ### Author Response · Authors · 2023-11-16
> **Response to Reviewer gT9R [1/2]**
>
> ### **Overall**
>
> We have carefully reviewed the detailed questions the reviewer **gT9R** provided regarding the robustness of our experiments and have made every effort to respond to these questions. We have added several experimental results to our paper to enhance its robustness, as the reviewer suggested.
>
> ### **Response**
>
> **Q1. Are there any sensitivity analyses regarding the parameter in algorithm 1?**
>
> → Thank you for suggesting sensitivity analyses. We expanded the experiments in Section 4's ablation study and added the details in Appendix C.5. These experiments in the TC-CS scenario on ImageNet-C involved varying one hyperparameter at a time for parameters in algorithm 1 $(q, \alpha, \gamma)$.
>
> Sensitivity Analysis for $q$:
> | Model | Source | TENT | CMF ($q$) |  |  |  |  |
> | --- | --- | --- | --- | --- | --- | --- | --- |
> |  |  |  | 0.01 | 0.0075 | 0.005 | 0.0025 | 0.0001 |
> | ViT | 60.2 | 54.5±0.04 | 44.9±0.09 | 44.9±0.09 | 44.8±0.12 | 44.8±0.14 | 44.9±0.05 |
> | Swin | 64.0 | 64.0±0.14 | 46.7±0.25 | 46.5±0.05 | 46.6±0.12 | 46.5±0.18 | 46.9±0.18 |
> | D2V | 51.8 | 51.9±0.09 | 43.4±0.08 | 43.4±0.09 | 43.5±0.04 | 43.7±0.15 | 44.5±0.02 |
>
> Sensitivity Analysis for $\alpha$:
> | Model | Source | TENT | CMF ($\alpha$) |  |  |  |  |
> | --- | --- | --- | --- | --- | --- | --- | --- |
> |  |  |  | 0.999 | 0.99 | 0.95 | 0.9 | 0.8 |
> | ViT | 60.2 | 54.5±0.04 | 45.2±0.03 | 44.8±0.12 | 44.7±0.06 | 44.9±0.09 | 44.9±0.06 |
> | Swin | 64.0 | 64.0±0.14 | 46.9±0.12 | 46.6±0.12 | 46.7±0.13 | 46.8±0.24 | 47.1±0.20 |
> | D2V | 51.8 | 51.9±0.09 | 43.6±0.17 | 43.5±0.04 | 44.2±0.11 | 44.6±0.04 | 44.8±0.04 |
>
> Sensitivity Analysis for $\gamma$:
> | Model | Source | TENT | CMF ($\gamma$) |  |  |  |  |
> | --- | --- | --- | --- | --- | --- | --- | --- |
> |  |  |  | 0.999 | 0.99 | 0.95 | 0.9 | 0.8 |
> | ViT | 60.2 | 54.5±0.04 | 45.2±0.11 | 44.8±0.12 | 45.5±0.09 | 46.7±0.07 | 49.18±0.04 |
> | Swin | 64.0 | 64.0±0.14 | 47.1±0.21 | 46.6±0.12 | 47.2±0.14 | 48.5±0.20 | 50.68±0.28 |
> | D2V | 51.8 | 51.9±0.09 | 43.2±0.22 | 43.5±0.04 | 45.3±0.04 | 46.8±0.06 | 48.64±0.11 |
>
> → The results showed that $\gamma$ had the most significant impact. These results are added in Appendix C.5.
>
> **Q2. I am curious about the performance of replacing the “weight ensemble” in ROID with CMF (namely ROID with CMF).**
>
> → As mentioned in Section 4, CMF currently uses the same loss as ROID. Thus, the performance of CMF is equivalent to that of ROID with the weight ensemble replaced by CMF. This is listed in Table 1 - 4.
>
> **Q3. How about the performance of “DW-SLR + SCE” without CMF?**
>
> → The experiment you mentioned was conducted in the TC-CS scenario on the ImageNet-C dataset. The results are as follows:
>
> |  |  | ImageNet-C | D109 | Rendition | Sketch |
> | --- | --- | --- | --- | --- | --- |
> | DW-SLR + SCE (w/o CMF) | ViT | 47.6±0.06 | 48.3±0.92 | 44.6±0.25 | 58.4±0.05 |
> |  | Swin | 51.1±0.30 | 50.0±0.10 | 47.0±0.14 | 59.1±0.18 |
> |  | D2V | 46.3±0.15 | 47.4±0.41 | 42.2±0.13 | 55.6±0.07 |
> | DW-SLR + SCE + WE | ViT | 45.0±0.09 | 45.0±0.04 | 44.2±0.13 | 58.6±0.04 |
> |  | Swin | 47.0±0.26 | 45.1±0.10 | 46.0±0.10 | 58.9±0.11 |
> |  | D2V | 44.8±0.01 | 44.2±0.06 | 41.8±0.11 | 56.2±0.05 |
> | DW-SLR + SCE + CMF | ViT | 44.8±0.12 | 43.4±0.07 | 42.7±0.20 | 57.0±0.08 |
> |  | Swin | 46.6±0.12 | 43.6±0.12 | 44.1±0.24 | 56.7±0.13 |
> |  | D2V | 43.5±0.04 | 42.3±0.11 | 40.0±0.06 | 53.9±0.03 |
>
> → **The weight ensemble (WE) method, using a fixed source model, did not show significant performance improvements on the Rendition and Sketch datasets**. This empirically demonstrates the problem we raised in Section 2.2 about dependency on past tasks due to the fixed source model.
>
> → **Replacing WE with CMF** allows for a **balance between preserving past information and adapting to the current task**, thus securing superior performance across all datasets.
>
> → **These experimental details are added in Appendix C.3.**
>
> **Q4. How about the in-distribution performance of compared methods after the adaptation of out-of-distribution data? Please refer to the comparison manner proposed in EATA.**
>
> → We conducted experiments on both in-distribution (Clean) and out-of-distribution (Corrupt) data. In the TC-CS scenario, we used the ImageNet test dataset for Clean and the entire ImageNet-C for Corrupt. After completing a specific domain in the OTTA process, we measured performance on both Clean and Corrupt datasets.
>
> → **The results are added in Appendix C.4, Figure 5**. For Clean data, EATA, ROID, and CMF all showed a **tendency to converge to a similar average error rate**. However, for Corrupt data, both EATA and ROID lost past information and converged to a certain average error rate, failing to effectively utilize past information. In contrast, **CMF continuously reduced the average error rate**.
>
> → In conclusion, CMF demonstrated the ability to continuously accumulate knowledge through balancing past and current information.

---

> ### Author Response · Authors · 2023-11-16
> **Response to Reviewer gT9R [2/2]**
>
> **Q5. Could the authors provide a computational complexity (wall-clock GPU time) comparison regarding the proposed CMF?**
>
> → This is the computational complexity (in second) for each model and OTTA method in the TC-CS scenario on the ImageNet-C dataset.
>
> | Method | Model |  |  |
> | --- | --- | --- | --- |
> |  | ViT | Swin | D2V |
> | CMF | 159.4 | 255.2 | 162.0 |
> | ROID | 158.6 | 254.4 | 160.6 |
> | SAR | 181.2 | 294.1 | 180.8 |
> | EATA | 112.6 | 176.8 | 64.0 |
> | TENT | 102.5 | 153.6 | 101.3 |
>
> → Despite using the same code as the official GitHub, SAR reported high computational complexity. On average, CMF consumed a very small 0.55% additional computation compared to the highest performing recorded ROID.
>
> **Q6. Could CMF help MEMO work stably in the online setting?**
>
> **A6.** Thank you for raising this interesting question regarding the application of CMF in an online setting for MEMO. In response to your query, we conducted experiments applying CMF to MEMO in an online (i.e., non-episodic) setting. These experiments were carried out using the ViT model on the ImageNet-C dataset under the TC-CS scenario.
>
> → For these experiments, we adhered to the learning rate of $0.00001$ as originally used in MEMO, with a batch size of $1$. We employed AugMix for augmentation, increasing the batch size to $64$. Other experimental conditions followed the basic hyperparameters mentioned in Section 4. The results of these experiments are as follows:
>
> | Method | gaussian | shot | impulse | defocus | glass | motion | zoom | snow | frost | fog | bright | contrast | elastic | pixelate | jpeg | AVG. |
> | --- | --- | --- | --- | --- | --- | --- | --- | --- | --- | --- | --- | --- | --- | --- | --- | --- |
> | MEMO | 71.1 | 96.7 | 99.0 | 91.5 | 99.5 | 99.3 | 99.7 | 99.8 | 99.9 | 99.9 | 99.6 | 99.9 | 99.9 | 99.9 | 99.9 | 97.0±0.11 |
> | MEMO + CMF | 65.0 | 64.8 | 62.1 | 68.5 | 73.0 | 62.8 | 65.4 | 57.3 | 46.5 | 50.4 | 29.0 | 73.4 | 57.0 | 57.1 | 48.3 | 58.7±0.01 |
>
> → The results indicate that integrating CMF with MEMO in an online setting effectively mitigates catastrophic forgetting. This finding is significant as it demonstrates the potential of CMF to enhance the stability and performance of MEMO in continuous learning environments. We believe these results contribute valuable insights into the adaptability and robustness of CMF when applied to different learning scenarios.
>
>
> **Q7. More motivation/explanations from the high level about why CMF could achieve a better tradeoff between adaptability and information retention are preferred.**
>
> → For a high-level explanation, we added **comparison figures and related works in Appendix D.1** to enhance the clarity of the tradeoff we claim. Thank you for the excellent advice.
>
> **Q8. Could the authors provide implementation details (hyperparameters) of baselines on different models and datasets?**
>
> → For fairness, we followed the hyperparameters of the existing benchmarks mentioned in Section 4, and all hyperparameters are disclosed in the supplemental materials `cfg/*.yaml` and `conf.py`. These values are fixed for datasets and vary only in learning rate for models. As disclosed in Appendix B.1 and B.2, “… the learning rate was set to $0.00025$ for ViT and Swin, $0.0002$ for ResNet-50, and $0.0001$ for D2V.”

---

> ### Author Response · Authors · 2023-11-22
> **A gentle reminder for discussion**
>
> Dear Reviewer gT9R ,
>
> Thank you again for your insightful review of our submission. We have consistently endeavored to provide detailed responses to address the concerns you have raised along your commnets. Your valuable comments have also helped us to make our study more robust and clear.
>
>
> There is now one day left in the author-reviewer discussion period, during which we hope to have the reviewer review our response and revised submission. If you have any additional comments, we will do our best to answer any questions.
>
>
> We sincerely appreciate your efforts and time in reviewing our submission.
>
>
> Author(s)

---

### Official Review · Reviewer_iWLp · 2023-10-31

**Soundness:** 1 poor
**Presentation:** 2 fair
**Contribution:** 2 fair
**Rating:** 3
**Confidence:** 4

**Summary:**

This paper addresses the  Online Test-time Adaptation (OTTA) by applying the Kalman filter to infer the test-time "model parameters". The proposed method is evaluated over various datasets.

**Strengths:**

- Casting the OTTA challenge as a Bayesian filtering issue presents an interesting approach. The incorporation of the Kalman filter (KF) to facilitate this is noteworthy, as it offers a closed-form solution for posterior inference. However, the inherent linear assumption of the KF may not align well with many real-world scenarios.

- The research commendably evaluates the proposed method across a spectrum of image classification and speech recognition tasks, and the results indicate reasonable enhancements.

**Weaknesses:**

- Utilizing the Kalman filter for sequential model parameter inference might not be optimal given its inherent assumptions. The Kalman filter operates under the presumption of linear system dynamics and posits a Gaussian distribution for the posterior. This is often misaligned with real-world scenarios where state posteriors frequently exhibit multimodal distributions.

- While the exploration of the Kalman filter and other Bayesian filtering techniques for model adaptation/TTA is not entirely novel, it is crucial to delineate this work from previous contributions. It's recommended to rigorously compare, both theoretically and empirically, with established works like EKF[1] and PFDE[2]. Such a comparative analysis can better spotlight this paper's unique technical contributions.

- In Section 3.4, the paper discusses simplifying Bayesian filtering computations. An analysis evaluating the accuracy of these approximations is warranted.

[1] Abuduweili, A., & Liu, C. (2020, July). Robust online model adaptation by extended kalman filter with exponential moving average and dynamic multi-epoch strategy. In Learning for Dynamics and Control (pp. 65-74). PMLR.
[2] Huang, H., Gu, X., Wang, H., Xiao, C., Liu, H., & Wang, Y. (2022). Extrapolative continuous-time bayesian neural network for fast training-free test-time adaptation. Advances in Neural Information Processing Systems, 35, 36000-36013.

**Questions:**

See weakness.

---

> ### Author Response · Authors · 2023-11-16
> **Response to Reviewer iWLp [1/2]**
>
> ### **Overall**
>
> We have meticulously reviewed the questions raised by the reviewers **iWLp** and have endeavored to enhance our paper through this process. **By incorporating a section on related works in Appendix D.2 and Limitations of Appendix E,** we have addressed the papers mentioned in your queries. This addition helps clarify the distinctions between CMF and other studies that have applied Bayesian filters to different adaptation tasks.
>
> ### **Response**
>
> **Q1. Utilizing the Kalman filter for sequential model parameter inference might not be optimal given its inherent assumptions. The Kalman filter operates under the presumption of linear system dynamics and posits a Gaussian distribution for the posterior. This is often misaligned with real-world scenarios where state posteriors frequently exhibit multimodal distributions.**
>
> **A1.** We are grateful for your insightful observation regarding the use of the Kalman filter for sequential model parameter inference in our work. Your point about the limitations of the Kalman filter, particularly its reliance on linear system dynamics and Gaussian distribution assumptions, is well-taken and pertinent to our discussion.
>
> In response, we acknowledge that real-world scenarios often present complexities that linear systems and Gaussian models may not adequately capture. As you rightly pointed out, state posteriors in practical applications can exhibit non-linear characteristics and multimodal distributions. This discrepancy is a well-known challenge in the application of Bayesian filters to real-world data, as highlighted in studies such as [1], which deal with natural data or labels exhibiting such complex forms.
>
> → **However, in the context of our CMF approach, we treat DNN’s parameters as observations, where the linear Gaussian model assumption remains effective. This treatment is underpinned by theoretical and empirical evidence from studies [a, b, c],** which suggest that applying a linear system, such as a moving average, to DNN’s parameters over time can enhance model performance and generalizability. For instance, study [a] demonstrates **the application of a linear system to parameters obtained via an SGD-based optimizer, employing a methodology similar to CMF's linear Gaussian model and scalar parameter settings (Section 3.4)**.
>
> Moreover, **we have rigorously validated CMF across various datasets and in real-world applications, such as speech recognition tasks, where complex distribution shifts are prevalent**. In these scenarios, CMF has demonstrated superior performance compared to existing OTTA methods. This empirical evidence supports the efficacy of our approach even in the presence of non-linear and multimodal distribution characteristics in real-world data.
>
> → We also recognize the limitations of our approach, particularly when the hyperparameter $\gamma$ in CMF is set to excessively low values.  As shown in Appendix C.5, Table 12 of our paper, CMF's performance can be disproportionately affected by changes in other hyperparameters under such conditions. This phenomenon, as you astutely noted, could be attributed to the non-linearities inherent in the data. **The sensitivity of CMF to non-linearities or multimodality, especially through the direct influence of the hyperparameter $\gamma$ on the target model, underscores the importance of hyperparameter tuning in maintaining the linearity of parameters. This aspect indeed represents a limitation of CMF.** In our experiments, as detailed in the paper, we fixed $\gamma$ at 0.99, which proved effective in most scenarios.
>
> **In conclusion, while we agree with your assessment of the limitations inherent in linear Gaussian models, our findings suggest that in the specific context of CMF and DNN parameter treatment, this approach remains robust and effective. We plan future work to identify applications where such linear CMF systems are challenging and apply simple non-linear functions (for example, confidence-based parameter selection) because it is challenging to guarantee that all applications will depend on this linear system.**
>
> [a] Izmailov, Pavel et al. “Averaging Weights Leads to Wider Optima and Better Generalization.” *Conference on Uncertainty in Artificial Intelligence* (2018).
>
> [b] Garipov, Timur, et al. "Loss surfaces, mode connectivity, and fast ensembling of dnns." *Advances in neural information processing systems* 31 (2018).
>
> [c] Guo, Hao, Jiyong Jin, and Bin Liu. "Stochastic weight averaging revisited." *Applied Sciences* 13.5 (2023): 2935.

---

> > ### Comment · Reviewer_iWLp · 2023-11-23
> > **Thanks**
> >
> > Thanks for carefully handling my concerns.  I still have concerns about the oversimplification of the assumption of dynamics of DNN's parameters with respect to domain shifts. Besides, the oversimplification of the update related to the moment function is indeed remembering the moving average. I hope the author can put the details of such a moment function in the major paper to avoid confusing readers.  My doubt is that updating the final model parameter by balancing between the source model with the test-time tuned model with linear/nonlinear combinations seems equivalent and exhibits a much simpler form. Then the motivation for introducing the Kalman filter seems vague, considering such a model assumes linear dynamics for almost all components in a dynamical system (e.g. transition and emission probabilities ).  Since my major concern remains, I decided to keep my original ratings.

---

> > > ### Author Response · Authors · 2023-11-23
> > > **Response to Reviewer iWLp [-/3]**
> > >
> > > We thank the reviewer for their continued engagement and thoughtful feedback.
> > >
> > > Our approach, as mentioned, interprets the process of retaining past task information and assimilating new information for OTTA from a probabilistic perspective. The adoption of the KF was motivated by its suitability in modeling these dynamics probabilistically. We acknowledge that this choice involves a degree of simplification, particularly in assuming linear dynamics for almost all components of the dynamic system, such as transition and emission probabilities.
> > >
> > > However, this simplification is not merely a matter of convenience but a strategic decision to balance the complexity of the model with its practical applicability and interpretability. The CMF's design, which simplifies the algorithm, aims to strike a balance between the source model and the test-time adjustment model. This balance is crucial in ensuring that the model remains adaptable and responsive to new data, without being overly complex or computationally intensive.
> > >
> > > In conclusion, while we recognize the reviewer's concerns about simplification, we assure you that these decisions were made with careful consideration of the balance between model complexity and practical effectiveness.

---

> ### Author Response · Authors · 2023-11-16
> **Response to Reviewer iWLp [2/2]**
>
> **Q2. While the exploration of the Kalman filter and other Bayesian filtering techniques for model adaptation/TTA is not entirely novel, it is crucial to delineate this work from previous contributions. It's recommended to rigorously compare, both theoretically and empirically, with established works like EKF[1] and PFDE[2]. Such a comparative analysis can better spotlight this paper's unique technical contributions.**
>
> **A2.** We have not found papers applying KF or EKF to OTTA. **[1] focuses on multi-epoch adaptation problems with observations being natural data or labels**. However, CMF focuses on **OTTA problems with DNN’s parameter** $\theta_t$ as the observation.
>
> → [1] designs the emission model $\mathcal{N}(Y|f(\theta_{t-1}, X_{t-1}), I)$ with a non-linear function $f$.
>
> → **EKF requires linear approximation of non-linear models**, which can lead to instability and reduced accuracy, especially in OTTA scenarios where computation cost is a critical factor.
>
> → In contrast, CMF’s emission model $p(\theta^{(t)}|\phi^{(t)})=\mathcal{N} (\theta^{(t)}|H\phi^{(t)}, R)$ targets DNN’s parameter $\theta^{(t)}$. Based on the rationale mentioned in A1, a linear system is applicable, **avoiding the need for linear approximation in Gaussian models** and **enabling exact Bayesian inference.**
>
> → **Thus, CMF is numerically stable, simple, and has a relatively low computation cost, making it suitable for various applications.**
>
> **A2 (cont.).** To our knowledge, there are no instances of particle filters being applied to OTTA methods. This is primarily due to the significant computational demands associated with particle filters, which require a large number of particles to function effectively.
>
> → This characteristic poses a substantial challenge for their use in computation-efficient OTTA tasks, where minimizing computational overhead is crucial.
>
> → [2] attempts to address this by redefining the problem using PFDE, which reduces the required number of particles. However, even with this reduction, PFDE still necessitates the use of 14 particles for relatively simple applications like MNIST and RNN. This requirement becomes even more prohibitive when considering large models such as ViT, Swin, and D2V, which are used in complex real-world scenarios proposed by SAR, or datasets with a high number of classes like ImageNet-C. **Our CMF employs the KF with scalar parameter settings, akin to using only one particle, which significantly reduces computational use and memory consumption compared to particle filters.**
>
> → Despite these challenges, the integration of particle filters with OTTA methods remains a promising avenue for future research, offering potential advancements in the field.
>
> **Q3. In Section 3.4, the paper discusses simplifying Bayesian filtering computations. An analysis evaluating the accuracy of these approximations is warranted.**
>
> **A3.** In the context of EKF, the non-linear function serves as the mean of the Gaussian model and is linearly approximated. This approximation is crucial for maintaining accuracy, as the non-linear function itself represents the ground truth.
>
> → However, the linear Gaussian model employed by CMF does not necessitate additional linear approximation. Furthermore, the parameters of KF in this model are optional, which complicates the definition of accuracy since there is no explicit ground truth against which to measure.
>
> → Moreover, **Section 3.4 of CMF primarily focuses on the settings for KF parameters rather than an approximation of the matrix version.**
>
> → A potential comparison could be made between the performance of CMF in matrix settings versus scalar settings. However, the dimension $d$ of the adaptation target parameter in our experiments ranges from $128$ to $1024$, leading to matrices as large as $1024\times1024$.
>
> → Both EKF and KF require matrix inversion for all target DNN's parameters to calculate the Kalman gain $K$. For example, for the ViT model, the calculation is required for $50$ matrices. Thus, this process lead to computational errors or extremely slow processing speeds in the PyTorch toolkit.
>
> → **Due to these limitations, CMF adopts scalar settings and proposes a simplified inference process in Section 3.4.** This approach is a key contribution of our work, as it allows for exact Bayesian inference within the linear Gaussian model of the scalar setting, while also addressing computational efficiency concerns.

---

> ### Author Response · Authors · 2023-11-22
> **A gentle reminder for discussion**
>
> Dear Reviewer iWLp ,
>
> Thank you again for your insightful review of our submission. We have consistently endeavored to provide detailed responses to address the concerns you have raised along your commnets. Your valuable comments have also helped us to make our study more robust and clear.
>
> There is now one day left in the author-reviewer discussion period, during which we hope to have the reviewer review our response and revised submission. If you have any additional comments, we will do our best to answer any questions.
>
> We sincerely appreciate your efforts and time in reviewing our submission.
>
> Author(s)

---

### Official Review · Reviewer_4Nt5 · 2023-11-05

**Soundness:** 3 good
**Presentation:** 3 good
**Contribution:** 4 excellent
**Rating:** 8
**Confidence:** 4

**Summary:**

The paper proposes parameter averaging/filtering for online test-time adaptation. The idea is to first find the task minimum, and then use the previous task to form a “prior” and average the parameters. The actual algorithm can be thought of as a simplified version of the Kalman filtering algorithm but they do not consider the full covariance because of high dimensionality. The paper shows the algorithm’s effectiveness on online test-time adaptation, where a full network has been pretrained and only normalization parameters are “adapted” to each task, where different tasks exhibit distribution shift on image styles and textures or speech environments.

**Strengths:**

- The proposed methodology is properly derived from a Kalman filter algorithm and has probabilistic interpretations.
- The proposed methodology can be adapted and simplified to a deep network where only normalization parameters are changing.
- Empirically, the proposed method can be applied on various backbone networks and achieve strong results on online test-time adaptation.

**Weaknesses:**

- The proposed method seems to work for online test-time adaptation which requires a well trained network. It would be good to investigate whether this could be a limitation. It would be good to understand whether it relies on a pretrained network or the continual style of training can also extend to a network from scratch. It would also be good to understand the limitation on the number of adaptation parameters and the number of tasks it can continually learn without forgetting. When does the method break down? Since the methodology is a general one, it would be good to understand its general characteristics. To this end, I would appreciate to see some toy experiments on parameter averaging that answers these questions.

- It would be good to study on the sensitivity of each hyperparameter.

**Questions:**

See above comments.

---

> ### Author Response · Authors · 2023-11-16
> **Response to Reviewer 4Nt5**
>
> ### **Response**
>
> **Q1. The proposed method seems to work for online test-time adaptation which requires a well trained network. It would be good to investigate whether this could be a limitation. It would be good to understand whether it relies on a pretrained network or the continual style of training can also extend to a network from scratch. It would also be good to understand the limitation on the number of adaptation parameters and the number of tasks it can continually learn without forgetting. When does the method break down? Since the methodology is a general one, it would be good to understand its general characteristics. To this end, I would appreciate to see some toy experiments on parameter averaging that answers these questions.**
>
> **A1.** Thank you for your insightful comments. Our proposed framework is indeed primarily designed for OTTA scenarios. The methodology is built upon the premise of adapting a well-trained network to new, unseen data distributions encountered during the test phase. This approach is particularly relevant in real-world applications where data distributions can shift unpredictably.
>
> Regarding the reliance on a pretrained network, our method does indeed assume the availability of a well-trained source model as a starting point. This is a common practice in OTTA methodologies, as it leverages the extensive knowledge already encapsulated in the pretrained model. However, the adaptability of our CMF framework to a network trained from scratch is an interesting avenue for future research.
>
> Regarding the limitations concerning the number of adaptation parameters and tasks, our current experiments have not explicitly quantified the upper bounds of these aspects. The CMF framework has demonstrated robustness in various OTTA scenarios, but there is indeed a theoretical limit to the number of tasks and parameters it can handle before performance degradation occurs.
>
> → However, one point we would like to emphasize is **the new experimental results added in Appendix C.4, Figure 5**. These results suggest that **CMF has the potential to scale to a larger number of tasks**. In this experiment, we measured performance on both the source and target datasets at the end of each adaptation for each task.
>
> → As observed in the experiments, CMF appears to retain consistent information across the entire test dataset of the source dataset, ImageNet (Clean), while simultaneously improving performance on the entire ImageNet-C (Corrupt) dataset. **This characteristic is not observed in other recent methodologies** like EATA and ROID, indicating a unique advantage of CMF in **handling multiple tasks**.
>
> → We appreciate the suggestion to conduct toy experiments on parameter averaging to explore these boundaries and will consider incorporating such experiments in our future research.
>
> **Q2. It would be good to study on the sensitivity of each hyperparameter.**
>
> **A2.** Thank you for suggesting sensitivity analyses. In response to your valuable feedback, we have expanded our experiments to include a comprehensive analysis of hyperparameter sensitivity. These additional experiments are detailed in Section 4's ablation study and further elaborated in Appendix C.5.
>
> → In these experiments, conducted in the TC-CS scenario on ImageNet-C, we varied one hyperparameter at a time for the parameters in Algorithm 1 $(q, \alpha, \gamma)$.
>
> **Sensitivity Analysis for $q$:**
>
> | Model | Source | TENT | CMF ($q$) |  |  |  |  |
> | --- | --- | --- | --- | --- | --- | --- | --- |
> |  |  |  | 0.01 | 0.0075 | 0.005 | 0.0025 | 0.0001 |
> | ViT | 60.2 | 54.5±0.04 | 44.9±0.09 | 44.9±0.09 | 44.8±0.12 | 44.8±0.14 | 44.9±0.05 |
> | Swin | 64.0 | 64.0±0.14 | 46.7±0.25 | 46.5±0.05 | 46.6±0.12 | 46.5±0.18 | 46.9±0.18 |
> | D2V | 51.8 | 51.9±0.09 | 43.4±0.08 | 43.4±0.09 | 43.5±0.04 | 43.7±0.15 | 44.5±0.02 |
>
> **Sensitivity Analysis for $\alpha$:**
>
> | Model | Source | TENT | CMF ($\alpha$) |  |  |  |  |
> | --- | --- | --- | --- | --- | --- | --- | --- |
> |  |  |  | 0.999 | 0.99 | 0.95 | 0.9 | 0.8 |
> | ViT | 60.2 | 54.5±0.04 | 45.2±0.03 | 44.8±0.12 | 44.7±0.06 | 44.9±0.09 | 44.9±0.06 |
> | Swin | 64.0 | 64.0±0.14 | 46.9±0.12 | 46.6±0.12 | 46.7±0.13 | 46.8±0.24 | 47.1±0.20 |
> | D2V | 51.8 | 51.9±0.09 | 43.6±0.17 | 43.5±0.04 | 44.2±0.11 | 44.6±0.04 | 44.8±0.04 |
>
> **Sensitivity Analysis for $\gamma$:**
>
> | Model | Source | TENT | CMF ($\gamma$) |  |  |  |  |
> | --- | --- | --- | --- | --- | --- | --- | --- |
> |  |  |  | 0.999 | 0.99 | 0.95 | 0.9 | 0.8 |
> | ViT | 60.2 | 54.5±0.04 | 45.2±0.11 | 44.8±0.12 | 45.5±0.09 | 46.7±0.07 | 49.18±0.04 |
> | Swin | 64.0 | 64.0±0.14 | 47.1±0.21 | 46.6±0.12 | 47.2±0.14 | 48.5±0.20 | 50.68±0.28 |
> | D2V | 51.8 | 51.9±0.09 | 43.2±0.22 | 43.5±0.04 | 45.3±0.04 | 46.8±0.06 | 48.64±0.11 |
>
> → The results from these sensitivity analyses revealed that $\gamma$ had the most significant impact on the performance of our CMF framework. These findings have been incorporated into Appendix C.5 for a more detailed examination.

---

> ### Author Response · Authors · 2023-11-22
> **A gentle reminder for discussion**
>
> Dear Reviewer 4Nt5 ,
>
> Thank you again for your insightful review of our submission. We have consistently endeavored to provide detailed responses to address the concerns you have raised along your commnets. Your valuable comments have also helped us to make our study more robust and clear.
>
> There is now one day left in the author-reviewer discussion period, during which we hope to have the reviewer review our response and revised submission. If you have any additional comments, we will do our best to answer any questions.
>
> We sincerely appreciate your efforts and time in reviewing our submission.
>
> Author(s)

---

### Author Response · Authors · 2023-11-18
**Summary of Revision**

**Acknowledgements:**

We extend our heartfelt gratitude to all the reviewers for their valuable comments and suggestions. We have exerted considerable effort to address the concerns raised. We believe that the quality of the paper has been significantly enhanced with the help of the reviewers.

**Major Revisions:**

After thorough revisions, we have updated the manuscript. The key changes are summarized as follows:

1. **Added Appendix C.3:** This section empirically demonstrates the limitations of existing OTTA methods that use a fixed source model and how CMF can alleviate these limitations. A more comprehensive ablation study has been conducted.
2. **Added Appendix C.4:** This section empirically shows that while existing OTTA methods preserve information from the source data, they fail to retain sufficient information about the target data beyond a certain point. CMF addresses this issue and suggests potential for growth in handling more tasks.
3. **Added Appendix C.5 - “Sensitivity of Hyperparameters in CMF”:** This experiment adds an analysis of the robustness of CMF and its relatively vulnerable hyperparameters.
4. **Added Appendix D.1:** A comparison between existing OTTA methods and CMF has been made, with an effort to provide a high-level explanation through Figure 6.
5. **Added Appendix D.2:** A methodological comparison from a Bayesian inference perspective has been conducted between existing studies applying Bayesian inference for state-space models to adaptation tasks and CMF. This comparison highlights the unique approach and contributions of CMF in addressing the OTTA problem.
6. **Appendix E - "Limitation":**  We discuss the limitations of CMF and future research directions to address these limitations.

For detailed content and complete experimental results, please refer to the revised manuscript. We are aware that our responses to the weaknesses mentioned by the reviewers might be inadequate or potentially misunderstood. If there are any such aspects, please inform us, and we will do our utmost to prepare a thorough response. We are committed to addressing all concerns and questions to the best of our ability and to further improve the quality and clarity of our work. We also hope that the revisions and responses adequately address all the concerns of the reviewers.

---

### Meta-Review · Area_Chair_yiBF · 2023-12-05

**Metareview:**

This paper proposes a continual momentum filtering (CMF) framework for the online test-time adaptation (TTA) problem leveraging the Kalman filter (KF) to balance model's adaptability vs knowledge retention. Online TTA is an important problem (often also referred to as lifelong or continual TTA) and the proposed method solves this problem by alternating between an SGD optimization step for the parameters and a KF step which models how the parameters evolve.

The reviewers appreciate the method for its probabilistic formulation/interpretation (so the method's derivation can be understood easily and its properties can be justified), and for its usefulness for continual/online TTA. The results also look impressive and the method outperforms various SOTA methods for online TTA.

There were some concerns about the method's reliance on the Kalman filter which could potentially imply a simplified evolution model for the parameters. This aspect was discussed during the discussion and it is felt that this concern is not very critical, given the rigorous formulation of the method and the excellent results.

Based on the reviews, the author response, the author-reviewer discussion, and the review-AC discussion, and my own reading of the paper, I believe the paper's strengths outweigh some of the issues such as the KF assumption. I therefore recommend acceptance.

**Justification For Why Not Higher Score:**

Accept as poster is appropriate. The paper isn't studying a new setting as such (continual/online TTA has already been around for some time) and some of the issue such as the KF assumption (although didn't result in rejection) are somewhat restrictive.

**Justification For Why Not Lower Score:**

Reasons for accepting are already summarized in the meta-review.

---

### Decision · Program_Chairs · 2024-01-16

Accept (poster)